# GEOMETRIC ANALYSIS OF NONCONVEX OPTIMIZATION LANDSCAPES FOR OVERCOMPLETE LEARNING

**Qing Qu**[*]
Center for Data Science
New York University
qq213@nyu.edu

**Yuexiang Zhai**
EECS
UC Berkeley
ysz@berkeley.edu

**Xiao Li**
Eletronic Engineering
CUHK
xli@ee.cuhk.edu.hk

**Yuqian Zhang**
Electrical & Computer Engineering
Rutgers University
yqz.zhang@rutgers.edu

**Zhihui Zhu**
Electrical & Computer Engineering
University of Denver
zhihui.zhu@du.edu

## ABSTRACT

Learning overcomplete representations finds many applications in machine learning and data analytics. In the past decade, despite the empirical success of heuristic methods, theoretical understandings and explanations of these algorithms are still far from satisfactory. In this work, we provide new theoretical insights for several important representation learning problems: learning *(i)* sparsely used overcomplete dictionaries and *(ii)* convolutional dictionaries. We formulate these problems as $\ell^4$-norm optimization problems over the sphere, and study the geometric properties of their nonconvex optimization landscapes. For both problems, we show the nonconvex objectives have benign (global) geometric structures, which enable development of efficient optimization methods finding the target solutions. Finally, our theoretical results are justified by numerical simulations.

## 1 INTRODUCTION

High dimensional data often has *low-complexity* structures (e.g., sparsity or low rankness). The performance of modern machine learning and data analytical methods heavily depends on appropriate low-complexity data representations (or features) which capture hidden information underlying the data. While we used to *manually* craft representations in the past, it has been demonstrated that *learned* representations from the data show much superior performance (Elad, 2010). Therefore, (unsupervised) learning of latent representations of high-dimensional data becomes a fundamental problem in signal processing, machine learning, theoretical neuroscience and many other fields (Bengio et al., 2013). Moreover, overcomplete representations for which the number of latent features exceeds the data dimensionality, have shown better representation of the data in various applications compared to complete representations (Lewicki & Sejnowski, 2000; Chen et al., 2001; Rubinstein et al., 2010). In this paper, we study the following overcomplete representation learning problems.

- **Overcomplete dictionary learning (ODL).** One of the most important unsupervised representation learning problems is learning *sparsely* used dictionaries (Olshausen & Field, 1997), which finds many applications in image processing and computer vision (Wright et al., 2010; Mairal et al., 2014). The task is given data

$$\underbrace{\boldsymbol{Y}}_{\text{data}} = \underbrace{\boldsymbol{A}}_{\text{dictionary}} \cdot \underbrace{\boldsymbol{X}}_{\text{sparse code}}, \tag{1.1}$$

we want to learn the compact representation (or dictionary) $\boldsymbol{A} \in \mathbb{R}^{n \times m}$ along with the sparse code $\boldsymbol{X} \in \mathbb{R}^{m \times p}$. For better representation of the data, it is often more desired that the dictionary $\boldsymbol{A}$ is overcomplete $m > n$, where it provides greater flexibility in matching structures in the data.

- **Convolutional dictionary learning (CDL).** Inspired by deconvolutional networks (Zeiler et al., 2010), the convolutional form of sparse representations (Bristow et al., 2013; Garcia-Cardona &

---

[*]The full version of this work can be found at `https://arxiv.org/abs/1912.02427`.

Wohlberg, 2018) replaces the unstructured dictionary $\boldsymbol{A}$ with a set of convolution filters $\{\boldsymbol{a}_{0k}\}_{k=1}^K$. Namely, the problem is that given multiple circulant convolutional measurements

$$\boldsymbol{y}_i \;=\; \sum_{k=1}^K \underbrace{\boldsymbol{a}_{0k}}_{\text{filter}} \;\circledast\; \underbrace{\boldsymbol{x}_{ik}}_{\text{sparse code}}, \qquad 1 \;\leqslant\; i \;\leqslant\; p, \tag{1.2}$$

one wants to learn the filters $\{\boldsymbol{a}_{0k}\}_{k=1}^K$ along with the sparse codes. The problem resembles a lot similarities to classical ODL. Indeed, one can show that Equation (1.2) reduces to Equation (1.1) in overcomplete settings by reformulation (Huang & Anandkumar, 2015). The interest of studying CDL was spurred by its better modeling ability of human visual and cognitive systems and the development of more efficient computational methods (Bristow et al., 2013), and has led to a number of applications in which the convolutional form provides state-of-art performance (Gu et al., 2015; Papyan et al., 2017b; Lau et al., 2019). Recently, the connections between CDL and convolutional neural network have also been extensively studied (Papyan et al., 2017a; 2018).

In addition, variants of finding overcomplete representations appear in many other problems beyond the dictionary learning problems we introduced here, such as overcomplete tensor decomposition (Anandkumar et al., 2017; Ge & Ma, 2017), overcomplete ICA (Lewicki & Sejnowski, 1998; Le et al., 2011), and short-and-sparse blind deconvolution (Zhang et al., 2017; 2018; Kuo et al., 2019).

**Prior arts on dictionary learning (DL).**   In the past decades, numerous heuristic methods have been developed for solving DL (Lee et al., 2007; Aharon et al., 2006; Mairal et al., 2010). Despite their empirical success (Wright et al., 2010; Mairal et al., 2014), theoretical understandings of when and why these methods work are still limited.

When the dictionary $\boldsymbol{A}$ is complete (Spielman et al., 2012) (i.e., square and invertible, $m = n$), by the fact that the row space of $\boldsymbol{Y}$ equals to that of $\boldsymbol{X}$ (i.e., $\mathrm{row}(\boldsymbol{Y}) = \mathrm{row}(\boldsymbol{X})$), Sun et al. (2016a) reduced the problem to finding the sparsest vector in a subspace (Demanet & Hand, 2014; Qu et al., 2016). By considering a (smooth) variant of the following $\ell^1$-minimization problem over the sphere,

$$\min_{\boldsymbol{q}} \; \frac{1}{p} \left\| \boldsymbol{q}^\top \boldsymbol{Y} \right\|_1, \quad \text{s.t.} \quad \boldsymbol{q} \in \mathbb{S}^{n-1}, \tag{1.3}$$

Sun et al. (2016a) showed that the nonconvex problem has no spurious local minima when the sparsity level[1] $\theta \in \mathcal{O}(1)$, and every local minimizer $\boldsymbol{q}_\star$ is a global minimizer with $\boldsymbol{q}_\star^\top \boldsymbol{Y}$ corresponding to one row of $\boldsymbol{X}$. The new discovery has led to efficient, guaranteed optimization methods for complete DL from random initializations (Sun et al., 2016b; Bai et al., 2018; Gilboa et al., 2019).

However, all these methods critically rely on the fact that $\mathrm{row}(\boldsymbol{Y}) = \mathrm{row}(\boldsymbol{X})$ for complete $\boldsymbol{A}$, there is no obvious way to generalize the approach to the overcomplete setting $m > n$. On the other hand, for learning incoherent overcomplete dictionaries, with sparsity $\theta \in \mathcal{O}(1/\sqrt{n})$ and stringent assumptions on $\boldsymbol{X}$, most of the current theoretical analysis results are local (Geng et al., 2011; Arora et al., 2015; Agarwal et al., 2016; Chatterji & Bartlett, 2017), in the sense that they require complicated initializations that could be difficult to implement in practice. Therefore, the legitimate question remains: why do heuristic methods solve ODL with simple initializations?

**Contributions.**   In this work we study the geometry of nonconvex landscapes for overcomplete/convolutional DL, where our result can be simply summarized by the following statement.

> There exists nonconvex formulations for ODL/CDL with benign optimization landscapes, that descent methods can learn overcomplete/convolutional dictionaries with simple[2] initializations.

Our approach follows the spirit of Sun et al. (2016a), while we overcome the aforementioned obstacles for overcomplete dictionaries by *directly finding columns of $\boldsymbol{A}$ instead of recovering sparse rows of $\boldsymbol{X}$*. We achieve this by reducing the problem to maximizing the $\ell^4$-norm[3] of $\boldsymbol{Y}^\top \boldsymbol{q}$ over the sphere,

---

[1]Here, the sparsity level $\theta$ denotes the proportion of nonzero entries in $\boldsymbol{X}$.

[2]Here, for ODL simple means random initializations; for CDL, it means simple data-driven initializations.

[3]The use of $\ell^4$-norm can also be justified from the perspective of sum of squares (SOS) (Barak et al., 2015; Ma et al., 2016; Schramm & Steurer, 2017). One can utilize properties of higher order SOS polynomials (such as 4-th order polynomials) to correctly recover columns of $\boldsymbol{A}$. But the complexity of these methods are quasi-polynomial, and hence much more expensive than the direct optimization approach we consider here.

which is known to promote the *spikiness* of the solution (Zhang et al., 2018; Li & Bresler, 2018; Zhai et al., 2019). In particular, we show the following results for ODL and CDL, respectively.

1. For the ODL problem, when $A$ is unit norm tight frame and incoherent, our nonconvex objective is *strict saddle* (Ge et al., 2015; Sun et al., 2015b) in the sense that any saddle point can be escaped by negative curvature and all local minimizers are globally optimal. Furthermore, every local minimizer is close to a column of $A$.

2. For the CDL problem, when the filters are self and mutual incoherent, a similar nonconvex objective is strict saddle over a sublevel set, within which every local minimizer is close to a target solution. Moreover, we develop a simple data-driven initialization that falls into this sublevel set.

Our analysis on ODL provides the *first global* characterization for nonconvex optimization landscape in the overcomplete regime. On the other hand, our result also gives the *first* provable guarantee for CDL. Indeed, under mild assumptions, our landscape analysis implies that with simple initializations, any descent method with the ability of escaping strict saddle points[4] provably finds global minimizers that are close to our target solutions for both problems. Moreover, our result opens up several interesting directions on nonconvex optimization that are worth of further investigations.

## 2 OVERCOMPLETE DICTIONARY LEARNING

In this section, we start stating our result with ODL. In Section 3, we will show how our geometric analysis here can be extended to CDL in a nontrivial way.

### 2.1 BASIC ASSUMPTIONS

We study the DL problem in Equation (1.1) under the following assumptions for $A \in \mathbb{R}^{n \times m}$ and $X \in \mathbb{R}^{m \times p}$. In particular, our assumption for the dictionary $A$ can be viewed as a generalization of orthogonality in the overcomplete setting (Mixon, 2016).

**Assumption 2.1 (Tight frame and incoherent dictionary $A$)** *We assume that the dictionary $A$ is unit norm tight frame (UNTF) (Mixon, 2016), in the sense that*

$$\frac{n}{m} A A^\top = I, \quad \|a_i\| = 1 \ (1 \leqslant i \leqslant m), \tag{2.1}$$

*and its columns satisfy the $\mu$-incoherence condition. Namely, let $A = [a_1 \quad a_2 \quad \cdots \quad a_m]$,*

$$\mu(A) := \max_{1 \leqslant i \neq j \leqslant m} \left| \left\langle \frac{a_i}{\|a_i\|}, \frac{a_j}{\|a_j\|} \right\rangle \right| \in (0, 1). \tag{2.2}$$

*We assume the coherence of $A$ is small, i.e., $\mu(A) \ll 1$.*

**Assumption 2.2 (Random Bernoulli-Gaussian $X$)** *We assume entries of $X \sim_{i.i.d.} \mathcal{BG}(\theta)$[5], that*

$$X = B \odot G, \quad B_{ij} \sim_{i.i.d.} \text{Ber}(\theta), \quad G_{ij} \sim_{i.i.d.} \mathcal{N}(0,1),$$

*where the Bernoulli parameter $\theta \in (0,1)$ controls the sparsity level of $X$.*

**Remark 1.** The coherence parameter $\mu$ plays an important role in shaping the optimization landscape. A smaller coherence $\mu$ implies that the columns of $A$ are less correlated, and hence easier for optimization. For matrices with $\ell^2$-normalized columns, classical Welch bound (Welch, 1974; Foucart & Rauhut, 2013a) suggests that the coherence $\mu$ is lower bounded by $\mu(A) \geqslant \sqrt{\frac{m-n}{(m-1)n}}$, which is achieved when $A$ is *equiangular tight frame* (Sustik et al., 2007). For a *generic random*[6] matrix $A$, w.h.p. it is approximately UNTF, with coherence $\mu(A) \approx \sqrt{\frac{\log m}{n}}$ roughly achieving the order of Welch bound. For a typical dictionary $A$ under Assumption 2.1, this suggests that the coherence parameter $\mu(A)$ often decreases w.r.t. the feature dimension $n$.

---

[4]Recent results show that methods such as trust-region (Absil et al., 2007; Boumal et al., 2018), cubic-regularization (Nesterov & Polyak, 2006), curvilinear search (Goldfarb et al., 2017), and even gradient descent (Lee et al., 2016) can provably escape strict saddle points.

[5]Here, we use $\mathcal{BG}(\theta)$ for abbreviation of Bernoulli-Gaussian distribution, with sparsity level $\theta \in (0,1)$.

[6]For instance, when $A$ is random Gaussian matrix, with each entry $a_{ij} \sim_{i.i.d.} \mathcal{N}(0, 1/n)$.

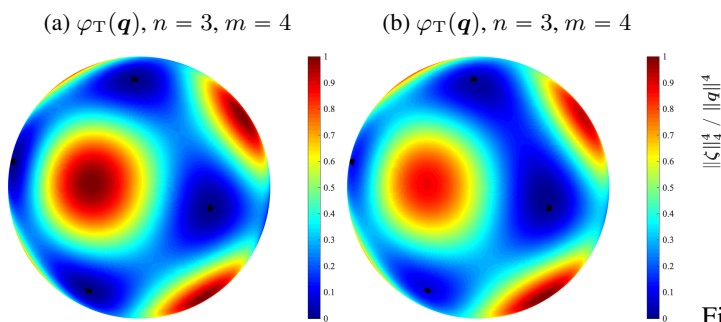

(a) $\varphi_{\mathrm{T}}(\boldsymbol{q})$, $n = 3$, $m = 4$     (b) $\varphi_{\mathrm{T}}(\boldsymbol{q})$, $n = 3$, $m = 4$

Figure 1: **Plots of landscapes** $\varphi_{\mathrm{T}}(\boldsymbol{q})$ and $\varphi_{\mathrm{DL}}(\boldsymbol{q})$ over $\mathbb{S}^2$. Both function values are normalized to $[0, 1]$. The overcomplete dictionary $\boldsymbol{A}$ is generated to be UNTF, with $n = 3$ and $m = 4$. The sparse coefficient $\boldsymbol{X} \sim \mathcal{BG}(\theta)$ with $\theta = 0.1$ and $p = 2 \times 10^4$. Black dots denote columns of $\boldsymbol{A}$ (target).

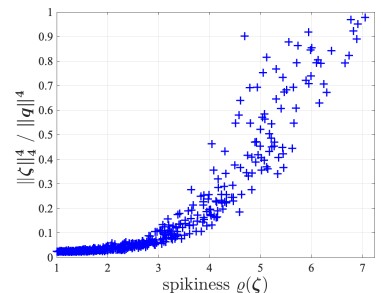

Figure 2: **Spikiness** $\varrho(\boldsymbol{\zeta})$ **vs.** $\|\boldsymbol{\zeta}\|_4^4 / \|\boldsymbol{q}\|^4$. We generate UNTF $\boldsymbol{A}$, randomly draw many points $\boldsymbol{q} \in \mathbb{S}^{n-1}$, and compute $\|\boldsymbol{\zeta}\|_4^4$ and spikiness $\varrho(\boldsymbol{\zeta})$ as in (2.6) with $\boldsymbol{\zeta} = \boldsymbol{A}^\top \boldsymbol{q}$. On the plot, we mark each point $\boldsymbol{q} \in \mathbb{S}^{n-1}$ by "+".

## 2.2 PROBLEM FORMULATION

We solve DL in the overcomplete regime by considering the following problem

$$\min_{\boldsymbol{q}} \ \varphi_{\mathrm{DL}}(\boldsymbol{q}) \ := \ -\frac{c_{\mathrm{DL}}}{p} \left\| \boldsymbol{q}^\top \boldsymbol{Y} \right\|_4^4 \ = \ -\frac{c_{\mathrm{DL}}}{p} \left\| \boldsymbol{q}^\top \boldsymbol{A} \boldsymbol{X} \right\|_4^4, \quad \text{s.t.} \quad \|\boldsymbol{q}\|_2 \ = \ 1, \tag{2.3}$$

where $c_{\mathrm{DL}} > 0$ is a normalizing constant. At the first glance, our objective looks similar to Equation (1.3) in complete DL, but we tackle the problem from a very different aspect – we directly find columns of $\boldsymbol{A}$ instead of recovering sparse rows of $\boldsymbol{X}$. Given UNTF $\boldsymbol{A}$ and random $\boldsymbol{X} \sim \mathcal{BG}(\theta)$, our intuition of solving Equation (2.3) originates from the fact (Lemma D.1)

$$\mathbb{E}_{\boldsymbol{X}} \left[ \varphi_{\mathrm{DL}}(\boldsymbol{q}) \right] \ = \ \varphi_{\mathrm{T}}(\boldsymbol{q}) \ - \ \frac{\theta}{2(1 - \theta)} \left( \frac{m}{n} \right)^2, \qquad \varphi_{\mathrm{T}}(\boldsymbol{q}) \ := \ -\frac{1}{4} \left\| \boldsymbol{A}^\top \boldsymbol{q} \right\|_4^4, \tag{2.4}$$

where $\varphi_{\mathrm{T}}(\boldsymbol{q})$ can be viewed as the objective for 4th order tensor decomposition in Ge & Ma (2017). When $p$ is large, this tells us that optimizing Equation (2.3) is approximately maximizing $\ell^4$-norm of $\boldsymbol{\zeta} = \boldsymbol{A}^\top \boldsymbol{q}$ over the sphere (see Figure 1). If $\boldsymbol{q}$ equals to one of the target solutions (e.g., $\boldsymbol{q} = \boldsymbol{a}_1$),

$$\boldsymbol{\zeta}(\boldsymbol{q}) \ := \ \boldsymbol{A}^\top \boldsymbol{q} \ = \ \left[ \underbrace{\|\boldsymbol{a}_1\|^2}_{=1} \quad \underbrace{\boldsymbol{a}_1^\top \boldsymbol{a}_2}_{|\cdot| < \mu} \quad \cdots \quad \underbrace{\boldsymbol{a}_1^\top \boldsymbol{a}_m}_{|\cdot| < \mu} \right]^\top, \tag{2.5}$$

then $\boldsymbol{\zeta}$ is *spiky* when $\mu$ is *small* (e.g., $\mu \ll 1$). Here, we introduce a notion of *spikiness* $\varrho$ for a vector $\boldsymbol{\zeta} \in \mathbb{R}^m$ by

$$\varrho(\boldsymbol{\zeta}) \ := \ \left| \zeta_{(1)} \right| / \left| \zeta_{(2)} \right|, \qquad \left| \zeta_{(1)} \right| \ \geqslant \ \left| \zeta_{(2)} \right| \ \geqslant \ \cdots \ \geqslant \ \left| \zeta_{(m)} \right|, \tag{2.6}$$

where $\zeta_{(i)}$ denotes the $i$th ordered entry of $\boldsymbol{\zeta}$. Figure 2 shows that larger $\varrho(\boldsymbol{\zeta})$ leads to larger $\|\boldsymbol{\zeta}\|_4^4$ with $\ell^2$-norm fixed. This implies that maximizing $\ell^4$-norm over the sphere *promotes* the spikiness of $\boldsymbol{\zeta}$ (Zhang et al., 2018; Li & Bresler, 2018; Zhai et al., 2019). Thus, from Equation (2.5), we expect the *global* minimizer $\boldsymbol{q}_\star$ of Equation (2.3) is close to one column of $\boldsymbol{A}$. Ge & Ma (2017) proved that for $\varphi_{\mathrm{T}}(\boldsymbol{q})$ there is no spurious local minimizer below a sublevel set whose measure over $\mathbb{S}^{n-1}$ geometrically shrinks w.r.t. the dimension $n$, and without providing valid initialization into the set.

Therefore, the challenge still remains: can simple descent methods solve the nonconvex objective Equation (2.3) to global optimality? In this work, we show that the answer is *affirmative*. Under proper assumptions, we show that our objective actually has benign *global* geometric structure, explaining why descent methods with random initialization solve the problem to the target solutions.

## 2.3 GEOMETRIC ANALYSIS OF NONCONVEX OPTIMIZATION LANDSCAPE

To characterize the landscape of $\varphi_{\mathrm{DL}}(\boldsymbol{q})$ over the sphere $\mathbb{S}^{n-1}$, let us first introduce some basic tools from Riemannian optimization (Absil et al., 2009a). For any function $f : \mathbb{S}^{n-1} \mapsto \mathbb{R}$, we have

$$\mathrm{grad} \, f(\boldsymbol{q}) \ := \ \boldsymbol{P}_{\boldsymbol{q}^\perp} \nabla f(\boldsymbol{q}), \qquad \mathrm{Hess} \, f(\boldsymbol{q}) \ := \ \boldsymbol{P}_{\boldsymbol{q}^\perp} \left( \nabla^2 f(\boldsymbol{q}) - \langle \boldsymbol{q}, \nabla f(\boldsymbol{q}) \rangle \boldsymbol{I} \right) \boldsymbol{P}_{\boldsymbol{q}^\perp}$$

to be the Riemannian gradient and Hessian[7] of $f(\boldsymbol{q})$. In addition, we partition $\mathbb{S}^{n-1}$ into two regions

$$\mathcal{R}_{\mathrm{N}} := \left\{ \boldsymbol{q} \in \mathbb{S}^{n-1} \mid \varphi_{\mathrm{T}}(\boldsymbol{q}) \geqslant -\xi_{\mathrm{DL}}\, \mu^{2/3}\, \|\boldsymbol{\zeta}(\boldsymbol{q})\|_3^2 \right\}, \tag{2.7}$$

$$\mathcal{R}_{\mathrm{C}} := \left\{ \boldsymbol{q} \in \mathbb{S}^{n-1} \mid \varphi_{\mathrm{T}}(\boldsymbol{q}) \leqslant -\xi_{\mathrm{DL}}\, \mu^{2/3}\, \|\boldsymbol{\zeta}(\boldsymbol{q})\|_3^2 \right\}, \tag{2.8}$$

for some fixed numerical constant $\xi_{\mathrm{DL}} > 0$. Unlike the approach in Sun et al. (2016a), our partition and landscape analysis are based on function value $\varphi_{\mathrm{T}}(\boldsymbol{q})$ instead of target solutions. This is because in overcomplete case the optimization landscape is more *irregular* compared to that of the complete/orthogonal case, which introduces extra difficulties for explicit partition of the sphere. In particular, for each region we show the following results.

**Theorem 2.3 (Global geometry of nonconvex landscape for ODL)** *Suppose we have*

$$K := m/n, \quad \theta \in \left( m^{-1}, 3^{-1} \right), \quad \xi_{\mathrm{DL}} > 2^6, \quad \mu \in \left( 0, 40^{-1} \right), \tag{2.9}$$

*and assume $\boldsymbol{Y} = \boldsymbol{A}\boldsymbol{X}$ such that $\boldsymbol{A}$ and $\boldsymbol{X}$ satisfy Assumption 2.1 and Assumption 2.2, respectively.*

*1. (Negative curvature in $\mathcal{R}_{\mathrm{N}}$) W.h.p. over the randomness of $\boldsymbol{X}$, whenever*

$$p \geqslant C\theta K^4 n^6 \log(\theta n/\mu) \quad and \quad K \leqslant 3 \cdot \left( 1 + 6\mu + 6\xi_{\mathrm{DL}}^{3/5}\mu^{2/5} \right)^{-1},$$

*any point $\boldsymbol{q} \in \mathcal{R}_{\mathrm{N}}$ exhibits negative curvature in the sense that*

$$\exists\, \boldsymbol{v} \in \mathbb{S}^{n-1}, \quad s.t. \quad \boldsymbol{v}^\top \operatorname{Hess}\varphi_{\mathrm{DL}}(\boldsymbol{q})\boldsymbol{v} \leqslant -3\, \|\boldsymbol{\zeta}\|_4^4\, \|\boldsymbol{\zeta}\|_\infty^2.$$

*2. (No bad critical points in $\mathcal{R}_{\mathrm{C}}$) W.h.p. over the randomness of $\boldsymbol{X}$, whenever*

$$p \geqslant C\theta K^3 \max\left\{ \mu^{-2}, Kn^2 \right\} n^3 \log(\theta n/\mu) \quad and \quad K \leqslant \xi_{\mathrm{DL}}^{3/2}/8,$$

*every critical point $\boldsymbol{q}_{\mathrm{c}}$ of $\varphi_{\mathrm{DL}}(\boldsymbol{q})$ in $\mathcal{R}_{\mathrm{C}}$ is either a strict saddle point that exhibits negative curvature for descent, or it is near one of the target solutions (e.g. $\boldsymbol{a}_1$) such that*

$$\langle \boldsymbol{a}_1/\|\boldsymbol{a}_1\|, \boldsymbol{q}_{\mathrm{c}} \rangle \geqslant 1 - 5\xi_{\mathrm{DL}}^{-3/2}.$$

*Here $C > 0$ is a universal constant.*

**Remark 2.** A combination of our geometric analysis for both regions provides the first global geometric analysis for ODL with $\theta \in \mathcal{O}(1)$, which implies that $\varphi_{\mathrm{DL}}(\boldsymbol{q})$ has *no* spurious local minimizers over $\mathbb{S}^{n-1}$: any critical point is either a strict saddle point that can be efficiently escaped, or it is near one of the target solutions. Moreover, recent results show that nonconvex problems with this type of optimization landscapes can be solved to optimal solutions by using (noisy) gradient descent methods with random initializations (Lee et al., 2016; Jin et al., 2017; Lee et al.; Criscitiello & Boumal, 2019). In addition, we point out several limitations of our result for future work.

- As we have only characterized properties of critical points, our result does not directly lead to convergence rate for descent methods. To show polynomial-time convergence, as suggested by Sun et al. (2016a; 2018); Li & Bresler (2018); Kuo et al. (2019), we need finer partitions of the sphere and uniform controls of derivatives in each region[8]. We leave this for future work.

- Our analysis in $\mathcal{R}_{\mathrm{N}}$ says that when $\mu$ is sufficiently small[9] the maximum overcompleteness $K$ allowed is roughly $K = 3$, which is smaller than that of $\mathcal{R}_{\mathrm{C}}$ (which could be a large constant). We believe this is mainly due to loose bounds for controlling norms of $\boldsymbol{A}$ in $\mathcal{R}_{\mathrm{C}}$. Moreover, our experiment result in Section 4 suggests that there is a substantial gap of $K$ between our theory and practice: the phase transition in Figure 3a shows that gradient descent with random initialization works even in the regime $m \leqslant n^2$. We leave improvement of our result as an open question.

---

[7]The Riemannian derivatives are similar to ordinary derivatives in Euclidean space, but they are defined in the tangent space of the manifold $\mathcal{M} = \mathbb{S}^{n-1}$. We refer readers to Absil et al. (2009a) for more details.

[8]Our preliminary investigation indicates that our premature analysis is not tight enough to achieve this.

[9]From Remark 1, for a typical $\boldsymbol{A}$, we expect $\mu \in \tilde{\mathcal{O}}((nK)^{-1/2})$ to be diminishing w.r.t. $n$.

**Brief sketch of analysis.** From Equation (2.4), we know that $\varphi_{\mathrm{DL}}(\boldsymbol{q})$ reduces to $\varphi_{\mathrm{T}}(\boldsymbol{q})$ in large sample limit as $p \to \infty$. This suggests an expectation and concentration type of analysis: *(i)* we first characterize critical points and negative curvature for the deterministic function $\varphi_{\mathrm{T}}(\boldsymbol{q})$ in $\mathcal{R}_{\mathrm{C}}$ and $\mathcal{R}_{\mathrm{N}}$ (see Appendix B); *(ii)* for any small $\delta > 0$, we show the measure concentrates in the sense that for a finitely large $p \geqslant \widetilde{\Omega}(\delta^{-2}\mathrm{poly}(n))$,

$$
\sup_{\boldsymbol{q} \in \mathbb{S}^{n-1}} \| \mathrm{grad}\, \varphi_{\mathrm{DL}}(\boldsymbol{q}) \,-\, \mathrm{grad}\, \varphi_{\mathrm{T}}(\boldsymbol{q}) \| \;\leqslant\; \delta, \qquad \sup_{\boldsymbol{q} \in \mathbb{S}^{n-1}} \| \mathrm{Hess}\, \varphi_{\mathrm{DL}}(\boldsymbol{q}) \,-\, \mathrm{Hess}\, \varphi_{\mathrm{T}}(\boldsymbol{q}) \| \;\leqslant\; \delta
$$

holds w.h.p. over the randomness of $\boldsymbol{X}$. Thus we can turn our analysis of $\varphi_{\mathrm{T}}(\boldsymbol{q})$ to that of $\varphi_{\mathrm{DL}}(\boldsymbol{q})$ by a perturbation analysis (see Appendix C & D). Here, it should be noticed that $\mathrm{grad}\, \varphi_{\mathrm{DL}}(\boldsymbol{q})$ and $\mathrm{Hess}\, \varphi_{\mathrm{DL}}(\boldsymbol{q})$ are 4th-order polynomials of $\boldsymbol{X}$, which are *heavy-tailed* empirical processes over $\boldsymbol{q} \in \mathbb{S}^{n-1}$. To control suprema of heavy-tailed processes, we developed a general truncation and concentration type of analysis similar to Zhang et al. (2018); Zhai et al. (2019), so that we can utilize classical bounds for sub-exponential random variables (Boucheron et al., 2013) (see Appendix F).

## 3 CONVOLUTIONAL DICTIONARY LEARNING

### 3.1 PROBLEM FORMULATION

Recall from Section 1, the basic task of CDL is that given convolutional measurements in the form of Equation (1.2), we want to recover kernels $\{\boldsymbol{a}_{0k}\}_{k=1}^{K}$. Here, by reformulating[10] CDL in the form of ODL, we generalize our analysis from Section 2.3 to CDL with a few new ingredients.

**Reduction from CDL to ODL.** For any $\boldsymbol{z} \in \mathbb{R}^n$, let $\boldsymbol{C}_{\boldsymbol{z}} \in \mathbb{R}^{n \times n}$ be the circulant matrix generated from $\boldsymbol{z}$. From Equation (1.2), the properties of circulant matrix imply that

$$
\boldsymbol{C}_{\boldsymbol{y}_i} \;=\; \boldsymbol{C}_{\sum_{k=1}^{K} \boldsymbol{a}_{0k} \circledast \boldsymbol{x}_{ik}} \;=\; \sum_{k=1}^{K} \boldsymbol{C}_{\boldsymbol{a}_{0k}} \boldsymbol{C}_{\boldsymbol{x}_{ik}} \;=\; \boldsymbol{A}_0 \cdot \boldsymbol{X}_i, \qquad 1 \leqslant i \leqslant p,
$$

with $\boldsymbol{A}_0 = \begin{bmatrix} \boldsymbol{C}_{\boldsymbol{a}_{01}} & \boldsymbol{C}_{\boldsymbol{a}_{02}} & \cdots & \boldsymbol{C}_{\boldsymbol{a}_{0K}} \end{bmatrix}$ and $\boldsymbol{X}_i = \begin{bmatrix} \boldsymbol{C}_{\boldsymbol{x}_{i1}}^{\top} & \boldsymbol{C}_{\boldsymbol{x}_{i2}}^{\top} & \cdots & \boldsymbol{C}_{\boldsymbol{x}_{iK}}^{\top} \end{bmatrix}^{\top}$, so that $\boldsymbol{A}_0 \in \mathbb{R}^{n \times nK}$ is *overcomplete* and structured. Thus, contencating all $\boldsymbol{C}_{\boldsymbol{y}_i}$, we have

$$
\underbrace{\begin{bmatrix} \boldsymbol{C}_{\boldsymbol{y}_1} & \boldsymbol{C}_{\boldsymbol{y}_2} & \cdots & \boldsymbol{C}_{\boldsymbol{y}_p} \end{bmatrix}}_{\boldsymbol{Y} \in \mathbb{R}^{n \times np}} = \boldsymbol{A}_0 \cdot \underbrace{\begin{bmatrix} \boldsymbol{X}_1 & \boldsymbol{X}_2 & \cdots & \boldsymbol{X}_p \end{bmatrix}}_{\boldsymbol{X} \in \mathbb{R}^{nK \times np}} \;\implies\; \boldsymbol{Y} = \boldsymbol{A}_0 \cdot \boldsymbol{X}.
$$

This suggests that we can view the CDL problem as ODL: if we can recover a column of the overcomplete dictionary $\boldsymbol{A}_0$, we find one of the filters $\boldsymbol{a}_{0k}$ $(1 \leqslant k \leqslant K)$ up to a *circulant shift*[11].

**Nonconvex problem formulation and preconditioning.** To solve CDL, one may consider the same objective Equation (2.3) as ODL. However, for many applications our structured dictionary $\boldsymbol{A}_0$ could be badly conditioned and *not* tight frame, which results in bad optimization landscape and even spurious local minimizers. To deal with this issue, we *whiten* our data $\boldsymbol{Y}$ by preconditioning[12]

$$
\boldsymbol{P}\boldsymbol{Y} = \boldsymbol{P}\boldsymbol{A}_0\boldsymbol{X}, \qquad \boldsymbol{P} = \left[ \left(\theta K^2 np\right)^{-1} \boldsymbol{Y}\boldsymbol{Y}^{\top} \right]^{-1/2}. \tag{3.1}
$$

For large $p$, we approximately have $\boldsymbol{P} \approx \left( K^{-1} \boldsymbol{A}_0 \boldsymbol{A}_0^{\top} \right)^{-1/2}$ (see Appendix E.5), so that

$$
\boldsymbol{P}\boldsymbol{Y} \approx \left( K^{-1} \boldsymbol{A}_0 \boldsymbol{A}_0^{\top} \right)^{-1/2} \boldsymbol{A}_0 \cdot \boldsymbol{X} = \boldsymbol{A} \cdot \boldsymbol{X}, \qquad \boldsymbol{A} := \left( K^{-1} \boldsymbol{A}_0 \boldsymbol{A}_0^{\top} \right)^{-1/2} \boldsymbol{A}_0,
$$

where $\boldsymbol{A}$ is automatically tight frame with $K^{-1}\boldsymbol{A}\boldsymbol{A}^{\top} = \boldsymbol{I}$. This suggests to consider

$$
\boxed{\min_{\boldsymbol{q}} \;\; \varphi_{\mathrm{CDL}}(\boldsymbol{q}) := -\frac{c_{\mathrm{CDL}}}{np} \left\| \boldsymbol{q}^{\top} \left( \boldsymbol{P}\boldsymbol{Y} \right) \right\|_4^4, \quad \text{s.t.} \quad \|\boldsymbol{q}\|_2 = 1,} \tag{3.2}
$$

---

[10]Similar formulation ideas also appeared in (Huang & Anandkumar, 2015) with no theoretical guarantees.

[11]The CDL problem exhibits shift symmetry in the sense that $\boldsymbol{a}_{0k} \circledast \boldsymbol{x}_{ik} = \mathrm{s}_\ell\left[\boldsymbol{a}_{0k}\right] \circledast \mathrm{s}_{-\ell}\left[\boldsymbol{x}_{ik}\right]$, where $\mathrm{s}_\ell\left[\cdot\right]$ is a circulant shift operator by length $\ell$. This implies we can only hope to solve CDL up to a shift ambiguity.

[12]Again, the $\theta$ here is only for normalization purpose, which does not affect optimization landscape. Similar $\boldsymbol{P}$ is also considered in Sun et al. (2016a); Zhang et al. (2018); Qu et al. (2019).

---

**Algorithm 1** Finding one filter with data-driven initialization

---

**Input:**   data $\boldsymbol{Y} \in \mathbb{R}^{n \times p}$
**Output:**   an esimated filter $\boldsymbol{a}_\star$
  1: **preconditioning.** Cook up the preconditioning matrix $\boldsymbol{P}$ in Equation (3.1).
  2: **initialization.** Initialize $\boldsymbol{q}_{\text{init}} = \mathcal{P}_{\mathbb{S}^{n-1}} (\boldsymbol{P} \boldsymbol{y}_\ell)$ with a random sample $\boldsymbol{y}_\ell$, $1 \leqslant \ell \leqslant p$.
  3: **optimization with escaping saddle points.** Optimize Equation (3.2) to a local minimizer $\boldsymbol{q}_\star$, by using a descent method such as Goldfarb et al. (2017) that escapes strict saddle points.
  4: **return** an estimated filter $\boldsymbol{a}_\star = \mathcal{P}_{\mathbb{S}^{n-1}} (\boldsymbol{P}^{-1} \boldsymbol{q}_\star)$.

---

for some normalizing constant $c_{\text{CDL}} > 0$, so that is *close* to optimizing

$$\widehat{\varphi}_{\text{CDL}}(\boldsymbol{q}) := -\frac{c_{\text{CDL}}}{np} \left\| \boldsymbol{q}^\top \boldsymbol{A} \boldsymbol{X} \right\|_4^4 \approx \varphi_{\text{CDL}}(\boldsymbol{q}),$$

for a tight frame dictionary $\boldsymbol{A}$ (we make this rigorous in Appendix E.4). To study the problem, we make assumptions on the sparse signals $\boldsymbol{x}_{ik} \sim_{i.i.d.} \mathcal{BG}(\theta)$ similar to Assumption 2.2. Furthermore, we assume $\boldsymbol{A}_0$ and $\boldsymbol{A}$ satisfy the following properties which serve as counterparts to Assumption 2.1.

**Assumption 3.1 (Properties of $\boldsymbol{A}_0$ and $\boldsymbol{A}$)** *We assume the filter matrix $\boldsymbol{A}_0$ has minimum singular value $\sigma_{\min}(\boldsymbol{A}_0) > 0$ with bounded condition number $\kappa(\boldsymbol{A}_0) := \sigma_{\max}(\boldsymbol{A}_0)/\sigma_{\min}(\boldsymbol{A}_0)$. In addition, we assume the columns of $\boldsymbol{A}$ are mutually incoherent:* $\max_{i \neq j} \left| \left\langle \frac{\boldsymbol{a}_i}{\|\boldsymbol{a}_i\|}, \frac{\boldsymbol{a}_j}{\|\boldsymbol{a}_j\|} \right\rangle \right| \leqslant \mu$.

### 3.2   GEOMETRIC ANALYSIS AND NONCONVEX OPTIMIZATION

**Optimization landscape for CDL.**   We characterize the geometric structure of $\varphi_{\text{CDL}}(\boldsymbol{q})$ over

$$\mathcal{R}_{\text{CDL}} := \left\{ \boldsymbol{q} \in \mathbb{S}^{n-1} \,\middle|\, \varphi_{\text{T}}(\boldsymbol{q}) \leqslant -\xi_{\text{CDL}} \, \mu^{2/3} \kappa^{4/3}(\boldsymbol{A}_0) \, \|\boldsymbol{\zeta}(\boldsymbol{q})\|_3^2 \right\}, \tag{3.3}$$

for some fixed numerical constant $\xi_{\text{CDL}} > 0$, where $\zeta(\boldsymbol{q}) = \boldsymbol{A}^\top \boldsymbol{q}$ and $\varphi_{\text{T}}(\boldsymbol{q}) = -4^{-1} \|\boldsymbol{\zeta}(\boldsymbol{q})\|_4^4$ as introduced in Equation (2.4). We show $\varphi_{\text{CDL}}(\boldsymbol{q})$ satisfies the following properties.

**Theorem 3.2 (Local geometry of nonconvex landscape for CDL)** *Let us denote $m := Kn$, and let $C_0 > 5$ and $\eta < 2^{-6}$ be some positive constants. Suppose we have*

$$\theta \in \left( m^{-1}, 3^{-1} \right), \qquad \xi_{\text{CDL}} = C_0 \cdot \eta^{-2/3}, \quad \mu \in \left( 0, 40^{-1} \right),$$

*and assume that Assumption 3.1 and $\boldsymbol{x}_{ik} \sim_{i.i.d.} \mathcal{BG}(\theta)$ hold. There exists some constant $C > 0$, w.h.p. over the randomness of $\boldsymbol{x}_{ik}s$, whenever*

$$p \geqslant C\theta K^2 \mu^{-2} n^4 \max \left\{ \frac{K^6 \kappa^6(\boldsymbol{A}_0)}{\sigma_{\min}^2(\boldsymbol{A}_0)}, \, n \right\} \log^6(n/\mu) \quad \text{and} \quad K < C_0,$$

*every critical point $\boldsymbol{q}_c$ in $\mathcal{R}_{\text{CDL}}$ is either a strict saddle point that exhibits negative curvature for descent, or it is near one of the target solutions (e.g. $\boldsymbol{a}_1$) such that $\langle \boldsymbol{a}_1/\|\boldsymbol{a}_1\|, \boldsymbol{q}_c \rangle \geqslant 1 - 5\kappa^{-2}\eta$.*

**Remark 3.**   The analysis is similar to that of ODL in $\mathcal{R}_C$ (see Appendix D). In contrast, our sample complexity $p$ and $\mathcal{R}_{\text{CDL}}$ have extra dependence on $\kappa(\boldsymbol{A}_0)$ due to preconditioning in Equation (3.1). On the other hand, because our preconditioned dictionary $\boldsymbol{A}$ is tight frame but not necessarily UNTF, in the worst case we *cannot* exclude existence of spurious local minima in $\mathcal{R}_{\text{CDL}}^c \bigcap \mathbb{S}^{n-1}$ for CDL.

**From geometry to optimization.**   Nonetheless, in Algorithm 1 we propose a simple data-driven initialization $\boldsymbol{q}_{\text{init}}$ such that $\boldsymbol{q}_{\text{init}} \in \mathcal{R}_{\text{CDL}}$. Since $\mathcal{R}_{\text{CDL}}$ does not have bad local minimizers, by proving that all iterates stay within $\mathcal{R}_{\text{CDL}}$, it suffices to show global convergence of Algorithm 1. We initialize $\boldsymbol{q}$ by randomly picking a preconditioned data sample $\boldsymbol{P}\boldsymbol{y}_\ell$ with $\ell \in [p]$, and set

$$\boldsymbol{q}_{\text{init}} = \mathcal{P}_{\mathbb{S}^{n-1}} (\boldsymbol{P}\boldsymbol{y}_\ell), \qquad \text{s.t.} \quad \boldsymbol{\zeta}_{\text{init}} = \boldsymbol{A}^\top \boldsymbol{q}_{\text{init}} \approx \sqrt{K} \mathcal{P}_{\mathbb{S}^{nK-1}} (\boldsymbol{A}^\top \boldsymbol{A} \boldsymbol{x}_\ell). \tag{3.4}$$

For generic $\boldsymbol{A}$, small $\mu(\boldsymbol{A})$ implies that $\boldsymbol{A}^\top \boldsymbol{A}$ is close to a diagonal matrix[13], so that $\boldsymbol{\zeta}_{\text{init}}$ is *spiky* for sparse $\boldsymbol{x}_\ell$. Therefore, we expect large $\|\boldsymbol{\zeta}_{\text{init}}\|_4^4$ and $\boldsymbol{q}_{\text{init}} \in \mathcal{R}_{\text{CDL}}$ by leveraging sparsity of $\boldsymbol{x}_\ell$.

---

[13]This is because the off diagonal entries are bounded roughly by $\sqrt{K}\mu$, which are tiny when $\mu$ is small.

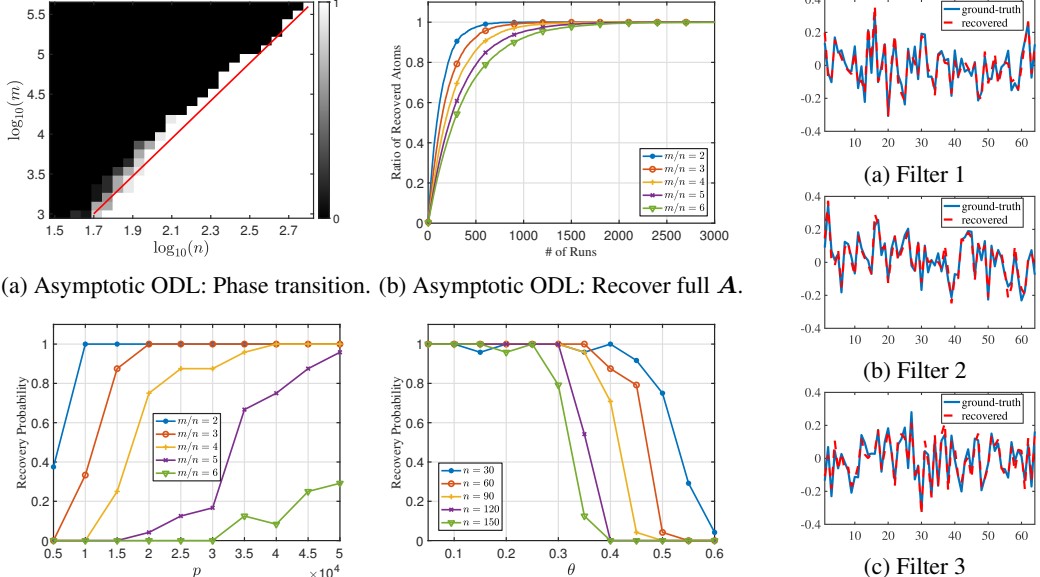

(a) Asymptotic ODL: Phase transition. (b) Asymptotic ODL: Recover full $\boldsymbol{A}$.

(a) Filter 1

(b) Filter 2

(c) ODL: Recovery probability vs. $p$. (d) ODL: Recovery probability vs. $\theta$.

(c) Filter 3

Figure 3: **Simulations for ODL.** (a) $\theta = 0.1$; (b) $n = 64$; (c) $n = 64, \theta = 0.1$; (d) $m = 3n, p = 5 \times 10^4$.

Figure 4: **CDL Simulation.** Parameters: $n = 64$, $\theta = 0.1$, $K = 3$, $p = 1 \times 10^4$.

**Proposition 3.3 (Convergence of Algorithm 1 to target solutions)** *With $m = Kn$, suppose*

$$c_1 \frac{\log m}{m} \leqslant \theta \leqslant c_2 \frac{\mu^{-2/3}}{\kappa^{4/3} m \log m} \cdot \min \left\{ \frac{\kappa^{4/3}}{\mu^{4/3}}, \frac{K\mu^{-4}}{m^2 \log m} \right\}. \tag{3.5}$$

*W.h.p. over the randomness of $\boldsymbol{x}_{ik}$s, whenever*

$$p \geqslant C\theta K^2 \mu^{-2} \max \left\{ K^6 \kappa^6(\boldsymbol{A}_0)/\sigma_{\min}^2(\boldsymbol{A}_0), \ n \right\} n^4 \log^6(m/\mu),$$

*we have $\boldsymbol{q}_{\text{init}} \in \mathcal{R}_{\text{CDL}}$, and all future iterates of Algorithm 1 stay within $\mathcal{R}_{\text{CDL}}$ and converge to an approximate solution (e.g., some circulant shift $\mathrm{s}_\ell [\boldsymbol{a}_{01}]$ of $\boldsymbol{a}_{0k}$ with $1 \leqslant \ell \leqslant n$) in the sense that*

$$\left\| \mathcal{P}_{\mathbb{S}^{n-1}} \left( \boldsymbol{P}^{-1} \boldsymbol{q}_\star \right) - \mathrm{s}_\ell [\boldsymbol{a}_{01}] \right\| \leqslant \epsilon,$$

*where $\epsilon$ is a small numerical constant. Here, $c_1, \ , c_2, \ C > 0$ are some numerical constants.*

**Remark 4.** Our result (Equation (3.5)) suggests that there is a *tradeoff* between $\mu$ and $\theta$ for optimization. For generic filters (e.g. drawn uniformly from the sphere), we approximately have[14] $\mu \in \widetilde{\mathcal{O}}(m^{-1/2})$ and $\kappa \in \mathcal{O}(1)$, so that our theory suggests the maximum sparsity allowed is $\theta \in \widetilde{\mathcal{O}}(m^{-2/3})$. For other smoother filters which may have larger $\mu$ and $\kappa$, the sparsity $\theta$ allowed tends to be smaller. Improving Equation (3.5) is the subject of future work. On the other hand, our result guarantees convergence to an approximate solution of constant error. We left exact recovery for future work. Finally, although we write CDL in the matrix-vector form, the optimization could be implemented very efficiently using fast Fourier transform (FFT) (see Appendix G).

## 4 EXPERIMENTS

In this section, we experimentally demonstrate our proposed formulation and approach for ODL and CDL. We solve our nonconvex problems in Equation (2.3) and Equation (3.2) using optimization methods[15] with random initializations introduced in Appendix G.

---

[14]See Figure 3 of Zhang et al. (2018) for an illustration of these estimations.

[15]For simplicity, we use power method (see Algorithm 3) for optimizing without tuning step sizes. In practice, we find both power method and Riemannian gradient descent have similar performance.

**Experiments on ODL.**  We generate data $\boldsymbol{Y} = \boldsymbol{AX}$, with dictionary $\boldsymbol{A} \in \mathbb{R}^{n \times m}$ being UNTF[16], and sparse code $\boldsymbol{X} \in \mathbb{R}^{m \times p} \sim_{i.i.d.} \mathcal{BG}(\theta)$. To judge the success recovery of one column of $\boldsymbol{A}$, let

$$\rho_e = \min_{1 \leqslant i \leqslant m} \left( 1 - |\langle \boldsymbol{q}_\star, \boldsymbol{a}_i / \|\boldsymbol{a}_i\| \rangle| \right).$$

We have $\rho_e = 0$ when $\boldsymbol{q}_\star = \mathcal{P}_{\mathbb{S}^{n-1}}(\boldsymbol{a}_i)$, thus we assume a recovery is successful if $\rho_e < 5 \times 10^{-2}$.

- **Overcompleteness.** First, we fix $\theta = 0.1$, and test the limit of the overcompleteness $K = m/n$ we can achieve by plotting the phase transition on $(m, n)$ in log scale. To get rid of the influence of sample complexity $p$, we run our algorithm on $\varphi_{\mathrm{T}}(\boldsymbol{q})$ which is the sample limit of $\varphi_{\mathrm{DL}}(\boldsymbol{q})$. For each pair of $(m, n)$, we repeat the experiment for 12 times. As shown in Figure 3a, it suggests that the limit of overcompleteness is roughly $m \approx n^2$, which is much larger than our theory predicts.
- **Recovering full matrix $\boldsymbol{A}$.** Second, although our theory only guarantees recovery of one column of $\boldsymbol{A}$, Figure 3b suggests that we can recover the full dictionary $\boldsymbol{A}$ by repetitive independent trials. As the result shows, $\mathcal{O}(m \log m)$ independent runs suffice to recover the full $\boldsymbol{A}$.
- **Recovery with varying $\theta$ and $p$.** Our simulation in Figure 3c implies that we need more samples $p$ when the overcompleteness $K$ increases. Meanwhile, Figure 3d shows successful recovery even when sparsity $\theta \approx 0.3$. The maximum $\theta$ seems to remain as a constant when $n$ increases.

**Experiments on CDL.**  Finally, for CDL, we generate measurement according to Equation (1.2) with $K = 3$, where the filters $\{\boldsymbol{a}_{0k}\}_{k=1}^K$ are drawn uniformly from the sphere $\mathbb{S}^{n-1}$, and $\boldsymbol{x}_{ik} \sim_{i.i.d.} \mathcal{BG}(\theta)$. Figure 4 shows that our method can approximately recover all the filters by running a few number of repetitive independent trials.

## 5  CONCLUSION AND DISCUSSION

In this work, we showed that nonconvex landscapes of overcomplete representation learning also possess benign geometric structures. In particular, by reducing the problem to an $\ell^4$ optimization problem over the sphere, we proved that ODL has no spurious local minimizers globally: every critical point is either an approximate solution or a saddle point can be efficiently escaped. Moreover, we showed that this type of analysis can be carried over to CDL with a few new ingredients, leading to the first provable method for solving CDL globally. Our results have opened several interesting questions that are worth of further investigations, that we discuss as follows.

**Tighter bound on overcompleteness for ODL.**  As shown in Theorem 2.3, our bound on the overcompleteness $K = m/n$ is an absolute constant, which we believe is far from tight (see experiments in Figure 3a). In the high overcompleteness regime (e.g., $n \ll m \leqslant n^2$), one conjecture is that spurious local minimizer does exist but descent methods with random initializations implicitly regularizes itself such that bad regions are automatically avoided Ma et al. (2017); another conjecture is that there is actually no spurious local minimizers. We tend to believe the latter conjecture is true. Indeed, the looseness of our analysis appears in the region $\mathcal{R}_{\mathrm{N}}$ (see Appendix B.2), for controlling the norms of $\boldsymbol{A}$.

One idea might be to consider i.i.d. Gaussian dictionary instead of the deterministic incoherent dictionary $\boldsymbol{A}$, and use probabilistic analysis instead of the worst-case deterministic analysis. However, our preliminary analysis suggests that elementary concentration tools for Gaussian empirical processes are not sufficient to achieve this goal. More advanced probabilistic tools might be needed here.

Another idea that might be promising is to leverage more advanced tools such as the sum of squares (SoS) techniques Lasserre (2001); Blekherman et al. (2012). Previous results Barak et al. (2015); Ma et al. (2016); Hopkins et al. (2015) used SoS as a computational tool for solving this type of problems, while the computational complexity is often quasi-polynomial and hence cannot handle problems of large-scale. In contrast, our idea here is to use SoS to verify the geometric structure of the optimizing landscape instead of computation, to have a better uniform control of the negative curvature in $\mathcal{R}_{\mathrm{N}}$. If we succeeded, this might lead to a tighter bound on the overcompleteness. Moreover, analogous to building dual certificates for convex relaxations such as compressive sensing Candès & Wakin (2008); Candes & Plan (2011) and matrix completion Candès & Recht (2009); Candès et al. (2011), it could potentially lead to a more general approach for verifying benign geometry structures for nonconvex optimization.

---

[16]The UNTF dictionary is generated by Tropp et al. (2005): (i) generate a standard Gaussian matrix $\boldsymbol{A}_0$, (ii) from $\boldsymbol{A}_0$ alternate between preconditioning the matrix and normalize the columns until convergence.

**Composition rules for nonconvex optimization?** Another interesting phenomenon we found through understanding ODL is that under certain scenarios the benign nonconvex geometry can be preserved under nonnegative addition. Indeed, if we separate our dictionary $\boldsymbol{A}$ into several subdictionaries as $\boldsymbol{A} = [\boldsymbol{A}_1 \quad \cdots \quad \boldsymbol{A}_N]$, then the asymptotic version of nonconvex objective for ODL (Equation (2.3)) can be rewritten as

$$\varphi_{\mathrm{T}}(\boldsymbol{q}) \;=\; -\frac{1}{4}\left\|\boldsymbol{A}^\top \boldsymbol{q}\right\|_4^4 \;=\; \sum_{k=1}^{N}\varphi_{\mathrm{T}}^k(\boldsymbol{q}), \quad \varphi_{\mathrm{T}}^k(\boldsymbol{q}) \;:=\; -\frac{1}{4}\left\|\boldsymbol{A}_k^\top \boldsymbol{q}\right\|_4^4 \quad (1 \leqslant k \leqslant N). \tag{5.1}$$

Presumably, every function $\varphi_{\mathrm{T}}^k(\boldsymbol{q})$ also possess benign geometry for each submatrix $\boldsymbol{A}_k$. This discovery might suggest more general properties in nonconvex optimization – benign geometry structures can be preserved under certain composition rules. Analogous to the study of convex functions Boyd & Vandenberghe (2004), discovering composition rules can potentially lead to simpler analytical tools for studying nonconvex optimization problems and hence have broad impacts.

**Finding all components over Stiefel or Oblique manifolds.** The nonconvex formulations considered in this work is only guaranteed to recover one column/filter at a time for ODL/CDL. Although our experimental results in Section 4 implies that the full dictionary or all the filters can be recovered by using repetitive independent trials, it is more desirable to have a formulation that can recover the whole dictionary/filters in one shot. This requires us to consider optimization problems constraint over more complicated manifolds rather than the sphere, such as Stiefel and Oblique manifolds Absil et al. (2009a). Despite of recent empirical evidences Lau et al. (2019); Li et al. (2019) and study of local geometry Zhai et al. (2019); Zhu et al. (2019), more technical tools need to be developed towards better understandings for nonconvex problems constraint over these more complicated manifolds.

**Miscellaneous.** Finally, we summarize several small questions that might be also worth of pursuing.

- **Exact recovery.** Our results only lead to approximate recovery of the target solutions. To obtain exact solutions, one might need to consider similar rounding steps as introduced in Qu et al. (2016); Sun et al. (2016b); Qu et al. (2019).

- **Designing better loss functions.** The $\ell^4$ objective we considered here for ODL and CDL is heavy-tailed for sub-Gaussian random variables, resulting in bad sample complexity and large approximation error. It would be nice to design better loss functions that also promotes spikiness of the solutions.

- **Non-asymptotic convergence for descent methods.** Unlike the results in Sun et al. (2016a;c); Kuo et al. (2019), our geometric analysis here does not directly lead to non-asymptotic convergence guarantees of any descent methods to global minimizers. This is because we only characterized the geometric properties of critical points on the function landscape. To show non-asymptotic convergence of methods introduced in Appendix G, we need to uniformly characterize the geometric properties over the sphere.

- **Finer models for CDL.** Finally, for CDL, it is worth noting that in many cases the length of the filters $\{\boldsymbol{a}_{0k}\}_{k=1}^{K}$ is often much shorter than the observations $\{\boldsymbol{y}_i\}_{i=1}^{p}$ Zhang et al. (2017); Kuo et al. (2019); Zhang et al. (2018); Lau et al. (2019), which has not been considered in this work. The extra structure leads to the so-called *short*-and-*sparse* CDL Lau et al. (2019), where the lower dimensional model can lead to fewer samples for recovery. Based on our results, we believe the short structure can be dealt with by developing finer analysis such as that in Kuo et al. (2019).

## ACKNOWLEDGEMENT

Part of this work was done when QQ and YXZ were at Columbia University. QQ thanks the generous support of the Microsoft graduate research fellowship and Moore-Sloan fellowship. XL would like to acknowledge the support by Grant CUHK14210617 from the Hong Kong Research Grants Council. YQZ is grateful to be supported by NSF award 1740822. ZZ was partly supported by NSF Grant 1704458. The authors would like to thank Joan Bruna (NYU Courant), Yuxin Chen (Princeton University), Lijun Ding (Cornell University), Han-wen Kuo (Columbia University), Shuyang Ling (NYU Shanghai), Yi Ma (UC Berkeley), Ju Sun (University of Minnesota, Twin Cities), René Vidal (Johns Hopkins University), and John Wright (Columbia University) for helpful discussions and inputs regarding this work.

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

APPENDIX

The Appendix is organized as follows. In Appendix A, we introduce the basic notations and technical tools for analysis. Appendix B provides a determinsitic characterization of the optimization landscape in population. In Appendix C, we turn our analysis of Appendix B into finite sample version. Appendix D and Appendix E provide the detailed proof for ODL and CDL, respectively. The detailed concentration bounds are postponed to Appendix F. Finally, Appendix G provides some ideas of optimization methods.

## A NOTATIONS AND BASIC TOOLS

### A.1 BASIC NOTATIONS

Throughout this paper, all vectors/matrices are written in bold font $a/A$; indexed values are written as $a_i, A_{ij}$. We use $\mathbb{S}^{n-1}$ to denote an $n$-dimensional unit sphere in the Euclidean space $\mathbb{R}^n$. We let $[m] = \{1, 2, \cdots, m\}$. We use $\odot$ to denote Hadamard product between two vectors/matrices. For $v \in \mathbb{R}^n$, we use $v^{\odot r}$ to denote entry-wise power of order $m$, i.e., $v^{\odot r} = [v_1^r, \cdots, v_n^r]^\top$. Let $F_n \in \mathbb{C}^{n \times n}$ denote a unnormalized $n \times n$ DFT matrix, with $\|F_n\| = \sqrt{n}$, and $F_n^{-1} = n^{-1} F_n^*$. In many cases, we just use $F$ to denote the DFT matrix.

**Some basic operators.** We use $\mathcal{P}_v$ and $\mathcal{P}_{v^\perp}$ to denote projections onto $v$ and its orthogonal complement, respectively. We let $\mathcal{P}_{\mathbb{S}^{n-1}}$ to be the $\ell^2$-normalization operator. To sum up, we have

$$\mathcal{P}_{v^\perp} u = u - \frac{vv^\top}{\|v\|^2} v, \quad \mathcal{P}_v u = \frac{vv^\top}{\|v\|^2} u, \quad \mathcal{P}_{\mathbb{S}^{n-1}} v = \frac{v}{\|v\|}.$$

**Circular convolution and circulant matrices.** The convolution operator $\circledast$ is *circular* with modulo-$m$: $(a \circledast x)_i = \sum_{j=0}^{m-1} a_j x_{i-j}$. For $v \in \mathbb{R}^m$, let $s_\ell[v]$ denote the cyclic shift of $v$ with length $\ell$. Thus, we can introduce the circulant matrix $C_v \in \mathbb{R}^{m \times m}$ generated through $v \in \mathbb{R}^m$,

$$C_v = \begin{bmatrix} v_1 & v_m & \cdots & v_3 & v_2 \\ v_2 & v_1 & v_m & & v_3 \\ \vdots & v_2 & v_1 & \ddots & \vdots \\ v_{m-1} & & \ddots & \ddots & v_m \\ v_m & v_{m-1} & \cdots & v_2 & v_1 \end{bmatrix} = \begin{bmatrix} s_0[v] & s_1[v] & \cdots & s_{m-1}[v] \end{bmatrix}. \tag{A.1}$$

Now the circulant convolution can also be written in a simpler matrix-vector product form. For instance, for any $u \in \mathbb{R}^m$ and $v \in \mathbb{R}^m$,

$$u \circledast v = C_u \cdot v = C_v \cdot u, \qquad C_{u \circledast v} = C_u C_v.$$

In addition, the correlation between $u$ and $v$ can be also written in a similar form of convolution operator which reverses one vector before convolution.

**Basics of Riemannian derivatives.** Here, we give a brief introduction to manifold optimization over the sphere, and the forms of Riemannian gradient and Hessian. We refer the readers to the book (Absil et al., 2009b) for more backgrounds. Given a point $q \in S^{n-1}$, the tangent space $T_q \mathbb{S}^{n-1}$ is defined as $T_q \mathbb{S}^{n-1} \doteq \{v \mid v^\top q = 0\}$. Therefore, we have the projection onto $T_q \mathbb{S}^{n-1}$ equal to $P_{q^\perp}$. For a function $f(q)$ defined over $\mathbb{S}^{n-1}$, we use $\operatorname{grad} f$ and $\operatorname{Hess} f$ to denote the Riemannian gradient and the Hessian of $f$, then we have

$$\operatorname{grad} f(q) \doteq P_{q^\perp} \nabla f(q), \qquad \operatorname{Hess} f(q) \doteq P_{q^\perp} \left( \nabla^2 f(q) - \langle q, \nabla f(q) \rangle I \right) P_{q^\perp},$$

where $\nabla f(q)$ and $\nabla^2 f(q)$ are the normal first and second derivatives in Euclidean space. For example, for the function $\varphi_{\mathrm{T}}(q)$ defined in Equation (2.4), direct calculations give that

$$\operatorname{grad} \varphi_{\mathrm{T}}(q) = -P_{q^\perp} A \left( A^\top q \right)^{\odot 3} = -P_{q^\perp} \sum_{k=1}^m \left( a_k^\top q \right)^3 a_k,$$

$$\operatorname{Hess} \varphi_{\mathrm{T}}(q) = -P_{q^\perp} \left[ 3A \operatorname{diag} \left( \left( A^\top q \right)^{\odot 2} \right) A^\top - \left\| A^\top q \right\|_4^4 I \right] P_{q^\perp}.$$

## A.2 BASIC TOOLS

**Lemma A.1 (Norm Inequality)** *If $p > r > 0$, then for $\boldsymbol{x} \in \mathbb{R}^n$, we have*

$$\|\boldsymbol{x}\|_p \leqslant \|\boldsymbol{x}\|_r \leqslant n^{1/r - 1/p} \|\boldsymbol{x}\|_p.$$

**Lemma A.2** *Let $z, \, r \in \mathbb{R}$. We have*

$$
\begin{aligned}
(1 + z)^r &\leqslant 1 + (2^r - 1)z, \quad \forall \, z \, \in \, [0, 1], \quad r \, \in \, \mathbb{R} \backslash (0, 1), \\
(1 + z)^r &\leqslant 1 + rz, \qquad \forall \, z \, \in \, [-1, +\infty), \, r \, \in \, [0, 1],
\end{aligned}
$$

*where the second inequality reverse when $r \in \mathbb{R} \backslash (0, 1)$.*

**Lemma A.3 (Moments of the Gaussian Random Variable)** *If $X \sim \mathcal{N}\left(0, \sigma_X^2\right)$, then it holds for all integer $m \geqslant 1$ that*

$$\mathbb{E}\left[|X|^m\right] \leqslant \sigma_X^m (m - 1)!!, \; k = \lfloor m/2 \rfloor.$$

**Lemma A.4 (Noncentral moments of the $\chi$ Random Variable)** *If $Z \sim \chi(m)$, then it holds for all integer $p \geqslant 1$ that*

$$\mathbb{E}\left[Z^p\right] = 2^{p/2} \frac{\Gamma\left(p/2 + m/2\right)}{\Gamma\left(m/2\right)} \leqslant p!! \, m^{p/2}.$$

**Lemma A.5 (Bernstein's Inequality for R.V.s (Foucart & Rauhut, 2013b))** *Let $X_1, \ldots, X_p$ be i.i.d. real-valued random variables. Suppose that there exist some positive numbers $R$ and $\sigma_X^2$ such that*

$$\mathbb{E}\left[|X_k|^m\right] \leqslant \frac{m!}{2} \sigma_X^2 R^{m-2}, \; \text{for all integers } m \geqslant 2.$$

*Let $S \doteq \frac{1}{p} \sum_{k=1}^p X_k$, then for all $t > 0$, it holds that*

$$\mathbb{P}\left[|S - \mathbb{E}\left[S\right]| \geqslant t\right] \leqslant 2 \exp\left(-\frac{pt^2}{2\sigma_X^2 + 2Rt}\right).$$

**Lemma A.6 (Bernstein's Inequality for Random Vectors (Sun et al., 2015a))** *Let $\boldsymbol{x}_1, \ldots, \boldsymbol{x}_p \in \mathbb{R}^d$ be i.i.d. random vectors. Suppose there exist some positive number $R$ and $\sigma_X^2$ such that*

$$\mathbb{E}\left[\|\boldsymbol{x}_k\|^m\right] \leqslant \frac{m!}{2} \sigma_X^2 R^{m-2}, \quad \text{for all integers } m \geqslant 2.$$

*Let $\boldsymbol{s} = \frac{1}{p} \sum_{k=1}^p \boldsymbol{x}_k$, then for any $t > 0$, it holds that*

$$\mathbb{P}\left[\|\boldsymbol{s} - \mathbb{E}\left[\boldsymbol{s}\right]\| \geqslant t\right] \leqslant 2(d + 1) \exp\left(-\frac{pt^2}{2\sigma_X^2 + 2Rt}\right).$$

**Lemma A.7 (Bernstein's Inequality for Bounded R.M.s, Theorem 1.6.2 of Tropp et al. (2015))** *Let $\boldsymbol{X}_1, \boldsymbol{X}_2, \cdots, \boldsymbol{X}_p \in \mathbb{R}^{d_1 \times d_2}$ be i.i.d. random matrices. Suppose we have*

$$\|\boldsymbol{X}_i\| \leqslant R \text{ almost surely}, \qquad \max\left\{\left\|\mathbb{E}\left[\boldsymbol{X}_i \boldsymbol{X}_i^\top\right]\right\|, \left\|\mathbb{E}\left[\boldsymbol{X}_i^\top \boldsymbol{X}_i\right]\right\|\right\} \leqslant \sigma_X^2, \quad 1 \leqslant i \leqslant p.$$

*Let $\boldsymbol{S} = \frac{1}{p} \sum_{i=1}^p \boldsymbol{X}_i$, then we have*

$$\mathbb{P}\left(\|\boldsymbol{S} - \mathbb{E}\left[\boldsymbol{S}\right]\| \geqslant t\right) \leqslant (d_1 + d_2) \exp\left(-\frac{pt^2}{2\sigma_X^2 + 4Rt/3}\right).$$

**Lemma A.8 (Bernstein's Inequality for Bounded Random Vectors)** *Let $\boldsymbol{x}_1, \boldsymbol{x}_2, \cdots, \boldsymbol{x}_p \in \mathbb{R}^d$ be i.i.d. random vectors. Suppose we have*

$$\|\boldsymbol{x}_i\| \leqslant R \text{ almost surely}, \qquad \mathbb{E}\left[\|\boldsymbol{x}_i\|^2\right] \leqslant \sigma_X^2, \quad 1 \leqslant i \leqslant p.$$

*Let $\boldsymbol{s} = \frac{1}{p} \sum_{i=1}^p \boldsymbol{x}_i$, then we have*

$$\mathbb{P}\left(\|\boldsymbol{s} - \mathbb{E}\left[\boldsymbol{s}\right]\| \geqslant t\right) \leqslant d \exp\left(-\frac{pt^2}{2\sigma_X^2 + 4Rt/3}\right).$$

**Lemma A.9 (Lemma A.4 of (Zhang et al., 2018))** *Let $v \in \mathbb{R}^d$ with each entry following i.i.d. $\mathrm{Ber}(\theta)$ distribution, then*

$$\mathbb{P}\left(\left|\|v\|_0 - \theta d\right| \geqslant t\theta d\right) \leqslant 2\exp\left(-\frac{3t^2}{2t+6}\theta d\right).$$

**Lemma A.10 (Matrix Perturbation Bound, Lemma B.12 of (Qu et al., 2019))** *Suppose $B > 0$ is a positive definite matrix. For any symmetric perturbation matrix $\Delta$ with $\|\Delta\| \leqslant \frac{1}{2}\sigma_{\min}(B)$, it holds that*

$$\left\|(B+\Delta)^{-1/2} - B^{-1/2}\right\| \leqslant \frac{4\|\Delta\|}{\sigma_{\min}^2(B)},$$

$$\left\|(B+\Delta)^{1/2}B^{-1/2} - I\right\| \leqslant \frac{4\|\Delta\|}{\sigma_{\min}^{3/2}(B)},$$

*where $\sigma_{\min}(B)$ denotes the minimum singular value of $B$.*

**Lemma A.11** *For any $q$, $q_1$, $q_2 \in \mathbb{S}^{n-1}$, we have*

$$\left\|P_{q^\perp}\right\| \leqslant 1, \quad \left\|P_{q_1} - P_{q_2}\right\| \leqslant 2\|q_1 - q_2\|.$$

**Proof** The first is obvious, and for the second inequality we have

$$\left\|P_{q_1^\perp} - P_{q_2^\perp}\right\| = \left\|q_1 q_1^\top - q_2 q_2^\top\right\| \leqslant \left\|q_1 q_1^\top - q_1 q_2^\top\right\| + \left\|q_1 q_2^\top - q_2 q_2^\top\right\| \leqslant 2\|q_1 - q_2\|,$$

as desired. ∎

**Lemma A.12** *For any nonzero vectors $u$ and $v$, we have*

$$\left\|\frac{u}{\|u\|} - \frac{v}{\|v\|}\right\| \leqslant \frac{2}{\|v\|}\|u - v\|.$$

**Proof** We have

$$\left\|\frac{u}{\|u\|} - \frac{v}{\|v\|}\right\| = \frac{1}{\|u\|\|v\|}\left\|\|v\|u - \|u\|v\right\|$$

$$= \frac{1}{\|u\|\|v\|}\left\|\|v\|u - \|v\|v + \|v\|v - \|u\|v\right\|$$

$$\leqslant \frac{1}{\|u\|\|v\|}\left(\|v\|\|u - v\| + \|v\|\left|\|u\| - \|v\|\right|\right) \leqslant \frac{2}{\|u\|}\|u - v\|,$$

as desired. ∎

## B    Analysis of Asymptotic Optimization Landscape

In this part of the appendix, we present the detailed analysis of the optimization landscape of the asymptotic objective

$$\min_q \varphi_{\mathrm{T}}(q) = -\frac{1}{4}\left\|A^\top q\right\|_4^4, \quad \text{s.t.} \quad q \in \mathbb{S}^{n-1}$$

over the sphere. We denote the overcompleteness of the dictionary $A \in \mathbb{R}^{n \times m}$ and the correlation of columns of $A$ with $q$ by

$$K := \frac{m}{n}, \quad \zeta(q) := A^\top q = \begin{bmatrix} \zeta_1 & \cdots & \zeta_m \end{bmatrix}^\top.$$

Without loss of generality, for a given $q \in \mathbb{S}^{n-1}$, we assume that

$$|\zeta_1| \geqslant |\zeta_2| \geqslant \cdots \geqslant |\zeta_m|.$$

**Assumption.** We assume that the dictionary $A$ is tight frame with $\ell^2$-norm bounded columns

$$\frac{1}{K} A A^\top = I, \quad \|a_i\| \leqslant M \ (1 \leqslant i \leqslant m). \tag{B.1}$$

We also assume that the columns of $A$ satisfy the $\mu$-*incoherence* condition. Namely, we have

$$\mu(A) := \max_{1 \leqslant i \neq j \leqslant m} \left| \left\langle \frac{a_i}{\|a_i\|}, \frac{a_j}{\|a_j\|} \right\rangle \right| \in (0,1), \tag{B.2}$$

such that $\mu$ is sufficiently small. Based on the function value of the objective $\varphi_{\mathrm{T}}(q)$, we partition the sphere into two regions

$$\mathcal{R}_{\mathrm{C}}(q; \xi) = \left\{ q \in \mathbb{S}^{n-1} \mid \|\zeta\|_4^4 \geqslant \xi \mu^{2/3} \|\zeta\|_3^2 \right\}, \tag{B.3}$$

$$\mathcal{R}_{\mathrm{N}}(q; \xi) = \left\{ q \in \mathbb{S}^{n-1} \mid \|\zeta\|_4^4 \leqslant \xi \mu^{2/3} \|\zeta\|_3^2 \right\}, \tag{B.4}$$

where $\xi > 0$ is some scalar. In the following, for appropriate choices of $K$, $\mu$, and $\xi$, we first show that $\mathcal{R}_{\mathrm{C}}$ does not have any spurious local minimizers by characterizing all the *critical* points within the region. Second, under more stringent condition that $A$ is $\ell^2$ column normalized, for the region $\mathcal{R}_{\mathrm{N}}$ we show that there exhibits large negative curvature throughout the region, such that there is no local/global minimizer within $\mathcal{R}_{\mathrm{N}}$.

## B.1 GEOMETRIC ANALYSIS OF CRITICAL POINTS IN $\mathcal{R}_{\mathrm{C}}$

In this subsection, we show that all the critical points of $\varphi_{\mathrm{T}}(q)$ in $\mathcal{R}_{\mathrm{C}}$ are either ridable saddle points, or satisfy second-order optimality condition and are close to the target solutions.

**Proposition B.1** *Suppose we have*

$$KM < 4^{-1} \cdot \xi^{3/2}, \quad M^3 < \eta \cdot \xi^{3/2}, \quad \mu < \frac{1}{20} \tag{B.5}$$

*for some constant $\eta < 2^{-6}$. Then any critical point $q \in \mathcal{R}_{\mathrm{C}}$, with $\operatorname{grad} \varphi_{\mathrm{T}}(q) = 0$, either is a ridable (strict) saddle point, or it satisfies second-order optimality condition and is near one of the components e.g., $a_1$ in the sense that*

$$\left\langle \frac{a_1}{\|a_1\|}, q \right\rangle \geqslant 1 - 5\xi^{-3/2} M^3 \geqslant 1 - 5\eta.$$

First, in Appendix B.1.1 we characterize some basic properties of critical points of $\varphi_{\mathrm{T}}(q)$. Based on this, we prove Proposition B.1 in Appendix B.1.2.

### B.1.1 BASIC PROPERTIES OF CRITICAL POINTS

**Lemma B.2 (Properties of critical points)** *For any point $q \in \mathbb{S}^{n-1}$, if $q$ is a critical point of $\varphi_{\mathrm{T}}(q)$ over the sphere, then it satisfies*

$$f(\zeta_i) = \zeta_i^3 - \alpha_i \zeta_i + \beta_i = 0 \tag{B.6}$$

*for all $i \in [m]$ with $\zeta(q) = A^\top q$, where*

$$\alpha_i := \frac{\|\zeta\|_4^4}{\|a_i\|^2}, \qquad \beta_i := \frac{\sum_{j \neq i} \langle a_i, a_j \rangle \zeta_j^3}{\|a_i\|^2}. \tag{B.7}$$

**Proof** For any point $q \in \mathbb{S}^{n-1}$, if $q$ is a critical point of $\varphi_{\mathrm{T}}(q)$ over the sphere, then its Riemannian gradient satisfies

$$\operatorname{grad} \varphi_{\mathrm{T}}(q) = P_{q^\perp} A \zeta^{\odot 3} = 0 \implies A \zeta^{\odot 3} - \|\zeta\|_4^4 q = 0.$$

Multiple $a_i^\top$ $(1 \leqslant i \leqslant m)$ on both sides of the equality, we obtain

$$\|a_i\|^2 \zeta_i^3 - \|\zeta\|_4^4 \zeta_i + \sum_{j \neq i} \langle a_i, a_j \rangle \zeta_j^3 = 0.$$

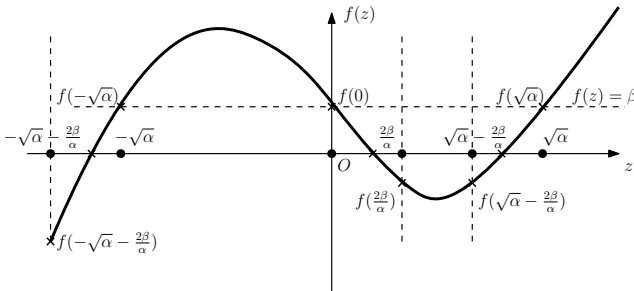

Figure 5: Illustration of $f(z)$ in Equation (B.8) when $\beta > 0$.

By replacing $\alpha_i$ and $\beta_i$ defined in Equation (B.7) into the equation above, we obtain the necessary condition in Equation (B.6) as desired. ∎

Since the roots of $f(z)$ correspond to the critical points of $\varphi_{\mathrm{T}}(\boldsymbol{q})$, we characterize the properties of the roots as follows.

**Lemma B.3** *Consider the following cubic polynomial*

$$f(z) = z^3 - \alpha z + \beta \tag{B.8}$$

*with*

$$0 < |\beta| \leqslant \frac{1}{4}\alpha^{3/2}, \qquad \alpha > 0. \tag{B.9}$$

*Then the roots of the function $f(z)$ is contained in one of the nonoverlapping intervals:*

$$\mathcal{I}_1 := \left\{ z \in \mathbb{R} \;\middle|\; |z| \leqslant \frac{2|\beta|}{\alpha} \right\}, \; \mathcal{I}_2 := \left\{ z \in \mathbb{R} \;\middle|\; |z - \sqrt{\alpha}| \leqslant \frac{2|\beta|}{\alpha} \right\},$$

$$\mathcal{I}_3 := \left\{ z \in \mathbb{R} \;\middle|\; |z + \sqrt{\alpha}| \leqslant \frac{2|\beta|}{\alpha} \right\}.$$

**Proof** By our construction $|\beta| \leqslant \frac{1}{4}\alpha^{3/2}$ and $\alpha > 0$ in Equation (B.9), it is obvious that the intervals $\mathcal{I}_1, \mathcal{I}_2$, and $\mathcal{I}_3$ are nonoverlapping. Without loss of generality, let us assume that $\beta$ is positive. We have

$$f(\sqrt{\alpha}) = f(-\sqrt{\alpha}) = f(0) = \beta > 0. \tag{B.10}$$

Thus, as illustrated in Figure 5, if we can show that

$$f\left(\frac{2\beta}{\alpha}\right) < 0, \quad f\left(-\sqrt{\alpha} - \frac{2\beta}{\alpha}\right) < 0, \quad f\left(\sqrt{\alpha} - \frac{2\beta}{\alpha}\right) < 0, \tag{B.11}$$

then this together with Equation (B.10) suffices to show that there exists at least one root in each of the three intervals $\mathcal{I}_1$, $\mathcal{I}_2$, and $\mathcal{I}_3$. Next, we show Equation (B.11) by direct calculations. First, notice that

$$f\left(\frac{2\beta}{\alpha}\right) = \left(\frac{2\beta}{\alpha}\right)^3 - \beta = \frac{\beta}{\alpha^3}\left(8\beta^2 - \alpha^3\right) = \frac{\beta}{\alpha^3}\left(\frac{1}{2}\alpha^3 - \alpha^3\right) \leqslant -\frac{1}{2}\beta < 0,$$

Second, we have

$$f\left(-\sqrt{\alpha} - \frac{2\beta}{\alpha}\right) = \left(-\sqrt{\alpha} - \frac{2\beta}{\alpha}\right)^3 - \alpha\left(-\sqrt{\alpha} - \frac{2\beta}{\alpha}\right) + \beta$$

$$= -8\frac{\beta^3}{\alpha^3} - \alpha^{3/2} - 6\beta - \frac{12\beta^2}{\alpha^{3/2}} + \alpha^{3/2} + 3\beta = -\frac{8\beta^3}{\alpha^3} - \frac{12\beta^2}{\alpha^{3/2}} - 3\beta < 0.$$

Similarly, we have

$$f\left(\sqrt{\alpha} - \frac{2\beta}{\alpha}\right) = -\frac{8\beta^3}{\alpha^3} + \frac{12\beta^2}{\alpha^{3/2}} - 3\beta = \beta\left(-\frac{8\beta^2}{\alpha^3} + \frac{12\beta}{\alpha^{3/2}} - 3\right) < -\frac{8\beta^3}{\alpha^3} < 0.$$

This proves Equation (B.11). Similar argument also holds for $\beta < 0$. Thus, we obtain the desired results. ∎

### B.1.2 Geometric Characterizations of Critical Points in $\mathcal{R}_C$

Based on the results in Appendix B.1.1, we prove Proposition B.1, showing that there is no spurious local minimizers in $\mathcal{R}_C$.

**Proof** [Proof of Proposition B.1] First recall from Lemma B.2, we defined

$$\alpha_i \;=\; \frac{\|\boldsymbol{\zeta}\|_4^4}{\|\boldsymbol{a}_i\|^2} \;>\; 0, \qquad \beta_i \;=\; \frac{\sum_{j\neq i}\langle \boldsymbol{a}_i, \boldsymbol{a}_j\rangle \zeta_j^3}{\|\boldsymbol{a}_i\|^2}.$$

Then for any $\boldsymbol{q} \in \mathcal{R}_C$, we have

$$\frac{|\beta_i|}{\alpha_i^{3/2}} \;=\; \frac{\left|\sum_{j\neq i}\langle \boldsymbol{a}_i, \boldsymbol{a}_j\rangle \zeta_j^3\right| \|\boldsymbol{a}_i\|}{\|\boldsymbol{\zeta}\|_4^6} \;\leqslant\; \frac{\mu M^3 \|\boldsymbol{\zeta}\|_3^3}{\|\boldsymbol{\zeta}\|_4^6} \;\leqslant\; M^3 \xi^{-3/2}, \tag{B.12}$$

where for the first inequality we used the fact that for any $i \in [m]$, $\|\boldsymbol{a}_i\| \leqslant M$ and

$$\left|\sum_{j\neq i}\langle \boldsymbol{a}_i, \boldsymbol{a}_j\rangle \zeta_j^3\right| \;\leqslant\; \left|\sum_{j\neq i}\left\langle \frac{\boldsymbol{a}_i}{\|\boldsymbol{a}_i\|}, \frac{\boldsymbol{a}_j}{\|\boldsymbol{a}_j\|}\right\rangle \zeta_j^3\right| \|\boldsymbol{a}_i\| \max_{1\leqslant j\leqslant m}\|\boldsymbol{a}_j\| \;\leqslant\; \mu M^2 \sum_{i=1}^m |\zeta_i|^3 \;=\; \mu M^2 \|\boldsymbol{\zeta}\|_3^3,$$

and the last inequality derives from the fact that $\boldsymbol{q} \in \mathcal{R}_C$. Thus, by Equation (B.5) and Equation (B.12), we obtain

$$M^3 \xi^{-3/2} \;\leqslant\; \frac{1}{4} \quad\implies\quad \frac{|\beta_i|}{\alpha_i^{3/2}} \;\leqslant\; \frac{1}{4}.$$

This implies that the condition in Equation (B.9) holds, so that we can apply Lemma B.3 to characterize the critical points. Based on Lemma B.3, we classify critical points $\boldsymbol{q} \in \mathcal{R}_C$ into three categories

1. All $|\zeta_i|$ $(1 \leqslant i \leqslant m)$ are smaller than $\frac{2|\beta_i|}{\alpha_i}$;
2. Only $|\zeta_1|$ is larger than $\frac{2|\beta_1|}{\alpha_1}$;
3. At least $|\zeta_1|$ and $|\zeta_2|$ are larger than $\frac{2|\beta_1|}{\alpha_1}$ and $\frac{2|\beta_2|}{\alpha_2}$, respectively.

For Case 1, Lemma B.4 shows that this type of critical point does not exist under the assumption in Equation (B.5). For Case 2, under the same assumption, Lemma B.5 implies that such a critical point $\boldsymbol{q} \in \mathcal{R}_C$ satisfies the second-order optimality condition, and it is near one of the target solution with

$$\left\langle \frac{\boldsymbol{a}_1}{\|\boldsymbol{a}_1\|}, \boldsymbol{q} \right\rangle \;\geqslant\; 1 - 5\xi^{-3/2}M^3 \;\geqslant\; 1 - 5\eta.$$

for some $\eta < 2^{-6}$. Finally, for Case 3, Lemma B.6 proves that this type of critical points $\boldsymbol{q} \in \mathcal{R}_C$ is ridable saddle, for which the Riemannian Hessian exhibits negative eigenvalue. Therefore, the critical points in $\mathcal{R}_C$ are either ridable saddle or near target solutions, so that there is no spurious local minimizer in $\mathcal{R}_C$. $\blacksquare$

In the following, we provided more detailed analysis for each case.

CASE 1: NO CRITICAL POINTS WITH SMALL ENTRIES.

First, we show by contradiction that if $\boldsymbol{q} \in \mathcal{R}_C$ and is a critical points, then there is at least one coordinate, e.g., $|\zeta_1| \geqslant \frac{2|\beta_1|}{\alpha_1}$. This implies that Case 1 (i.e., all $|\zeta_i|$ $(1 \leqslant i \leqslant m)$ are smaller than $\frac{2|\beta_i|}{\alpha_i}$) is impossible to happen. In other words, this means that any critical point $\boldsymbol{q} \in \mathcal{R}_C$ should be close to superpositions of columns of $\boldsymbol{A}$.

**Lemma B.4** *Suppose we have*

$$M^{4/3}K^{1/3} \;<\; 4^{-1/3}\xi.$$

*If $\boldsymbol{q} \in \mathcal{R}_C$ is a critical point, then there exists at least one $i \in [m]$ such that the entry $\zeta_i$ of $\boldsymbol{\zeta}(\boldsymbol{q})$ satisfies*

$$|\zeta_i| \;\geqslant\; \frac{2|\beta_i|}{\alpha_i}.$$

**Proof** Suppose there exists a $\boldsymbol{q} \in \mathcal{R}_{\mathrm{C}}$ such that all entries $\zeta_i$ satisfying $|\zeta_i| < \frac{2|\beta_i|}{\alpha_i}$. Then we have

$$\max_{1 \leqslant i \leqslant m} |\zeta_i| \;=\; \|\boldsymbol{\zeta}\|_\infty \;\leqslant\; \frac{2\left|\sum_{k=2}^m \langle \boldsymbol{a}_1, \boldsymbol{a}_k \rangle \zeta_k^3 \right|}{\|\boldsymbol{\zeta}\|_4^4} \;\leqslant\; \frac{2M^2 \mu \|\boldsymbol{\zeta}\|_3^3}{\|\boldsymbol{\zeta}\|_4^4}.$$

This implies that

$$\|\boldsymbol{\zeta}\|_4^4 \;\leqslant\; \|\boldsymbol{\zeta}\|_\infty^2 \|\boldsymbol{\zeta}\|^2 \;\leqslant\; \frac{4M^4 \mu^2 \|\boldsymbol{\zeta}\|_3^6}{\|\boldsymbol{\zeta}\|_4^8} \|\boldsymbol{\zeta}\|^2 \quad \Longrightarrow \quad \|\boldsymbol{\zeta}\|_4^{12} \;\leqslant\; 4M^4 \mu^2 \|\boldsymbol{\zeta}\|_3^6 \|\boldsymbol{\zeta}\|^2$$

$$\Longrightarrow \quad \|\boldsymbol{\zeta}\|_4^4 \;\leqslant\; 4^{1/3} M^{4/3} K^{1/3} \mu^{2/3} \|\boldsymbol{\zeta}\|_3^2,$$

where we used the fact that $\|\boldsymbol{\zeta}\|^2 = K$ according to [Equation (B.1)](#). Thus, by our assumption, we have

$$M^{4/3} K^{1/3} \;<\; \xi/4^{1/3} \quad \Longrightarrow \quad \|\boldsymbol{\zeta}\|_4^4 \;<\; \xi \mu^{2/3} \|\boldsymbol{\zeta}\|_3^2.$$

This contradicts with the fact that $\boldsymbol{q} \in \mathcal{R}_{\mathrm{C}}$. ∎

CASE 2: CRITICAL POINTS NEAR GLOBAL MINIMIZERS

Second, we consider the case that there exists only one big $\zeta_1$, for which the critical point satisfies second-order optimality and is near a true component.

**Lemma B.5** *Suppose $\xi$ is sufficiently large such that*

$$M^3 \;<\; \eta \cdot \xi^{3/2}, \qquad KM \;<\; 4^{-1} \cdot \xi^{3/2}, \tag{B.13}$$

*for some constant $\eta < 2^{-6}$. For any critical point $\boldsymbol{q} \in \mathcal{R}_{\mathrm{C}}$, if there is only one entry in $\boldsymbol{\zeta}$ such that $\zeta_1 \geqslant \frac{2|\beta_1|}{\alpha_1}$,*

$$\left\langle \frac{\boldsymbol{a}_1}{\|\boldsymbol{a}_1\|}, \boldsymbol{q} \right\rangle \;\geqslant\; 1 - 5\xi^{-3/2} M^3 \;\geqslant\; 1 - 5\eta.$$

*Moreover, such a critical point $\boldsymbol{q} \in \mathcal{R}_{\mathrm{C}}$ satisfies the second-order optimality condition: for any $\boldsymbol{v} \in \mathbb{S}^{n-1}$ with $\boldsymbol{v} \perp \boldsymbol{q}$,*

$$\boldsymbol{v}^\top \mathrm{Hess}\, \varphi_{\mathrm{T}}(\boldsymbol{q}) \boldsymbol{v} \;\geqslant\; \frac{1}{20} \|\boldsymbol{\zeta}\|_4^4.$$

**Proof** We first show that under our assumptions the critical point $\boldsymbol{q} \in \mathcal{R}_{\mathrm{C}}$ is near a target solution. Following this, we prove that $\boldsymbol{q}$ also satisfies second-order optimality condition.

**Closeness to target solutions.** First, if $\boldsymbol{q}$ is a critical point such that there is only one $\zeta_1 \geqslant \frac{2|\beta_1|}{\alpha_1}$, we show that such $\boldsymbol{q}$ is very close to a true component. By Lemma [B.2](#) and Lemma [B.3](#), we know that $\zeta_1$ needs to be upper bounded by

$$\zeta_1^2 \;\leqslant\; \left( \sqrt{\alpha_1} + \frac{2|\beta_1|}{\alpha_1} \right)^2 \;=\; \left( \frac{\|\boldsymbol{\zeta}\|_4^2}{\|\boldsymbol{a}_1\|} + \frac{2\left|\sum_{k=2}^m \langle \boldsymbol{a}_1, \boldsymbol{a}_k \rangle \zeta_k^3\right|}{\|\boldsymbol{\zeta}\|_4^4} \right)^2$$

$$\leqslant\; \frac{\|\boldsymbol{\zeta}\|_4^4}{\|\boldsymbol{a}_1\|^2} \left( 1 + \frac{2\mu \|\boldsymbol{\zeta}\|_3^3 \|\boldsymbol{a}_1\|^2 \max_{1 \leqslant j \leqslant m} \|\boldsymbol{a}_j\|}{\|\boldsymbol{\zeta}\|_4^6} \right)^2.$$

By using the fact that $\boldsymbol{q} \in \mathcal{R}_{\mathrm{C}}$ and $\|\boldsymbol{a}_j\| \leqslant M$ $(1 \leqslant j \leqslant m)$, we have

$$\|\boldsymbol{a}_1\|^2 \zeta_1^2 \;\leqslant\; \left( 1 + \frac{2\mu \|\boldsymbol{\zeta}\|_3^3 \|\boldsymbol{a}_1\|^2 \max_{1 \leqslant j \leqslant m} \|\boldsymbol{a}_j\|}{\|\boldsymbol{\zeta}\|_4^6} \right)^2 \|\boldsymbol{\zeta}\|_4^4 \;\leqslant\; \left( 1 + 2\xi^{-3/2} M^3 \right)^2 \|\boldsymbol{\zeta}\|_4^4. \tag{B.14}$$

On the other hand, by using the fact that $|\zeta_k| \leqslant \frac{2|\beta_k|}{\alpha_k}$ for all $k \geqslant 2$, we have

$$
\begin{aligned}
\zeta_1^4 \; \geqslant \; \|\boldsymbol{\zeta}\|_4^4 - \zeta_2^2 \sum_{k=2}^m \zeta_k^2 \; &\geqslant \; \|\boldsymbol{\zeta}\|_4^4 - \frac{4|\beta_2|^2}{\alpha_2^2} K \; \geqslant \; \|\boldsymbol{\zeta}\|_4^4 \left(1 - \frac{4\mu^2 \|\boldsymbol{\zeta}\|_3^6}{\|\boldsymbol{\zeta}\|_4^{12}} KM^4 \right) \\
&\geqslant \; \|\boldsymbol{\zeta}\|_4^4 \left(1 - 4\xi^{-3} KM^4 \right). \quad \text{(B.15)}
\end{aligned}
$$

Combining the lower and upper bounds in Equation (B.14) and Equation (B.15), we obtain

$$
\begin{aligned}
\left\langle \frac{\boldsymbol{a}_1}{\|\boldsymbol{a}_1\|}, \boldsymbol{q} \right\rangle^2 \; = \; \frac{\zeta_1^2}{\|\boldsymbol{a}_1\|^2} \; &\geqslant \; \frac{1 - 4\xi^{-3}KM^4}{\left(1 + 2\xi^{-3/2}M^3\right)^2} \; \geqslant \; \frac{\left(1 - 4\xi^{-3}KM^4\right)}{1 + 6\xi^{-3/2}M^3} \\
&= \; 1 - 2\xi^{-3}M^3 \left(3\xi^{3/2} + 2KM\right) \\
&\geqslant \; 1 - 8\xi^{-3/2}M^3 \; \geqslant \; 1 - 8\eta,
\end{aligned}
$$

where the second inequality follows by Lemma A.2, and the last inequality follows from Equation (B.13). This further gives

$$
\left\langle \frac{\boldsymbol{a}_1}{\|\boldsymbol{a}_1\|}, \boldsymbol{q} \right\rangle \; \geqslant \; \frac{1 - 8\xi^{-3/2}M^3}{\left(1 - 8\xi^{-3/2}M^3\right)^{1/2}} \; \geqslant \; \frac{1 - 8\xi^{-3/2}M^3}{1 - 4\xi^{-3/2}M^3} \; = \; 1 - 5\xi^{-3/2}M^3 \; \geqslant \; 1 - 5\eta. \quad \text{(B.16)}
$$

**Second-order optimality condition.** Second, we check the second order optimality condition for the critical point. Let $\boldsymbol{v} \in \mathbb{S}^{n-1}$ be any vector such that $\boldsymbol{v} \perp \boldsymbol{q}$, then

$$
\begin{aligned}
\boldsymbol{v}^\top \operatorname{Hess} \varphi_{\mathrm{T}}(\boldsymbol{q}) \boldsymbol{v} \; &= \; -3\boldsymbol{v}^\top \boldsymbol{A} \operatorname{diag}\left(\boldsymbol{\zeta}^{\odot 2}\right) \boldsymbol{A}^\top \boldsymbol{v} + \|\boldsymbol{\zeta}\|_4^4 \\
&= \; -3\langle \boldsymbol{a}_1, \boldsymbol{v} \rangle^2 \zeta_1^2 - 3\sum_{k=2}^m \langle \boldsymbol{a}_k, \boldsymbol{v} \rangle^2 \zeta_k^2 + \|\boldsymbol{\zeta}\|_4^4 \\
&\geqslant \; -3\langle \boldsymbol{a}_1, \boldsymbol{v} \rangle^2 \zeta_1^2 - 3\zeta_2^2 \left\|\boldsymbol{A}^\top \boldsymbol{v}\right\|^2 + \|\boldsymbol{\zeta}\|_4^4 \\
&= \; -3\langle \boldsymbol{a}_1, \boldsymbol{v} \rangle^2 \zeta_1^2 - 3K\zeta_2^2 + \|\boldsymbol{\zeta}\|_4^4 \quad \text{(B.17)}
\end{aligned}
$$

Next, we control $\langle \boldsymbol{a}_1, \boldsymbol{v} \rangle^2 \zeta_1^2$ and $K\zeta_2^2$ in terms of $\|\boldsymbol{\zeta}\|_4^4$, respectively. By Equation (B.14) and $\langle \boldsymbol{q}, \boldsymbol{v} \rangle = 0$,

$$
\begin{aligned}
\langle \boldsymbol{a}_1, \boldsymbol{v} \rangle^2 \cdot \zeta_1^2 \; &= \; \left\langle \frac{\boldsymbol{a}_1}{\|\boldsymbol{a}_1\|} - \boldsymbol{q}, \boldsymbol{v} \right\rangle^2 \left(\|\boldsymbol{a}_1\|^2 \zeta_1^2\right) \\
&\leqslant \; \left\|\frac{\boldsymbol{a}_1}{\|\boldsymbol{a}_1\|} - \boldsymbol{q}\right\|^2 \left(1 + 2\xi^{-3/2}M^3\right)^2 \|\boldsymbol{\zeta}\|_4^4 \\
&= \; 2\left(1 - \left\langle \frac{\boldsymbol{a}_1}{\|\boldsymbol{a}_1\|}, \boldsymbol{q} \right\rangle\right) \left(1 + 2\xi^{-3/2}M^3\right)^2 \|\boldsymbol{\zeta}\|_4^4 \\
&\leqslant \; 10\xi^{-3/2}M^3 \left(1 + 2\xi^{-3/2}M^3\right)^2 \|\boldsymbol{\zeta}\|_4^4 \; \leqslant \; \frac{1}{4} \|\boldsymbol{\zeta}\|_4^4. \quad \text{(B.18)}
\end{aligned}
$$

On the other hand, for $\boldsymbol{q} \in \mathcal{R}_{\mathrm{C}}$, using Equation (B.13) we have

$$
K\zeta_2^2 \; \leqslant \; K\frac{4|\beta_2|^2}{\alpha_2^2} \; \leqslant \; 4KM^4 \frac{\mu^2 \|\boldsymbol{\zeta}\|_3^6}{\|\boldsymbol{\zeta}\|_4^{12}} \cdot \|\boldsymbol{\zeta}\|_4^4 \; \leqslant \; 4KM^4 \xi^{-3} \|\boldsymbol{\zeta}\|_4^4 \; \leqslant \; \frac{1}{15} \|\boldsymbol{\zeta}\|_4^4. \quad \text{(B.19)}
$$

Thus, combining the results in Equation (B.17), Equation (B.18), and Equation (B.19), we obtain

$$
\boldsymbol{v}^\top \operatorname{Hess} \varphi_{\mathrm{T}}(\boldsymbol{q}) \boldsymbol{v} \; \geqslant \; \left(1 - \frac{3}{4} - \frac{1}{5}\right) \|\boldsymbol{\zeta}\|_4^4 \; \geqslant \; \frac{1}{20} \|\boldsymbol{\zeta}\|_4^4.
$$

This completes our proof. ∎

CASE 3: CRITICAL POINTS ARE RIDABLE SADDLES.

Finally, we consider the critical points $q \in \mathcal{R}_C$ that at least two entries $|\zeta_1|$ and $|\zeta_2|$ are larger than $\frac{2|\beta_1|}{\alpha_1}$ and $\frac{2|\beta_2|}{\alpha_2}$, respectively. For this type of critical points in $\mathcal{R}_C$, we show that they are ridable saddle points: the Hessian is nondegenerate and exhibits negative eigenvalues.

**Lemma B.6** *Suppose we have*

$$M^3 \; < \; \eta \cdot \xi^{3/2}, \qquad \mu \; < \; \frac{1}{20}, \tag{B.20}$$

*for some constant $\eta < 2^{-6}$ For any critical point $q \in \mathcal{R}_C$, if there are at least two entries in $\zeta(q)$ such that $|\zeta_i| > \frac{2|\beta_i|}{\alpha_i}$ ($i \in [m]$), then $q$ is a strict saddle point: there exists some $v \in \mathbb{S}^{n-1}$ with $v \perp q$, such that*

$$v^\top \operatorname{Hess} \varphi_{\mathrm{T}}(q) v \; \leqslant \; - \|\zeta\|_4^4.$$

**Proof** Without loss of generality, for any critical point $q \in \mathcal{R}_C$, we assume that $\zeta_1 = a_1^\top q$ and $\zeta_2 = a_2^\top q$ are the two largest entries in $\zeta(q)$. We pick a vector $v \in \operatorname{span}\left\{ \frac{a_1}{\|a_1\|}, \frac{a_2}{\|a_2\|} \right\}$ such that $v \perp q$ with $v \in \mathbb{S}^{n-1}$. Thus,

$$\begin{aligned}
v^\top \operatorname{Hess} \varphi_{\mathrm{T}}(q) v &= -3 v^\top A \operatorname{diag}\left( \zeta^{\odot 2} \right) A^\top v + \|\zeta\|_4^4 \\
&\leqslant -3 \|a_1\|^2 \zeta_1^2 \left\langle \frac{a_1}{\|a_1\|}, v \right\rangle^2 - 3 \|a_2\|^2 \zeta_2^2 \left\langle \frac{a_2}{\|a_2\|}, v \right\rangle^2 + \|\zeta\|_4^4 .
\end{aligned}$$

Since $|\zeta_1| \geqslant \frac{2|\beta_1|}{\alpha_1}$ and $|\zeta_2| \geqslant \frac{2|\beta_2|}{\alpha_2}$, by Lemma B.2, Lemma B.3, and the fact that $q \in \mathcal{R}_C$, we have

$$\begin{aligned}
\|a_1\|^2 \zeta_1^2 \;\geqslant\; \|a_1\|^2 \left( \sqrt{\alpha_1} - \frac{2|\beta_1|}{\alpha_1} \right)^2 &\geqslant\; \left( 1 - \frac{2\mu M^2 \|\zeta\|_3^3 \|a_1\|}{\|\zeta\|_4^6} \right)^2 \|\zeta\|_4^4 \\
&\geqslant\; \left( 1 - 2\xi^{-3/2} M^3 \right)^2 \|\zeta\|_4^4 .
\end{aligned}$$

In the same vein, we can also show that

$$\|a_2\|^2 \zeta_2^2 \;\geqslant\; \left( 1 - 2\xi^{-3/2} M^3 \right)^2 \|\zeta\|_4^4 .$$

Therefore, combining the results above, we obtain

$$v^\top \operatorname{Hess} \varphi_{\mathrm{T}}(q) v \;\leqslant\; \|\zeta\|_4^4 \left[ 1 - 3 \left( 1 - 2\xi^{-3/2} M^3 \right)^2 \left( \left\langle \frac{a_1}{\|a_1\|}, v \right\rangle^2 + \left\langle \frac{a_2}{\|a_2\|}, v \right\rangle^2 \right) \right] .$$

As $v \in \operatorname{span}\left\{ \frac{a_1}{\|a_1\|}, \frac{a_2}{\|a_2\|} \right\}$, we can write

$$v \;=\; c_1 \frac{a_1}{\|a_1\|} \;+\; c_2 \frac{a_2}{\|a_2\|}$$

for some coefficients $c_1, c_2 \in \mathbb{R}$. As $v \in \mathbb{S}^{n-1}$, we observe

$$\|v\|^2 \;=\; c_1^2 + c_2^2 + 2 c_1 c_2 \left\langle \frac{a_1}{\|a_1\|}, \frac{a_2}{\|a_2\|} \right\rangle \;=\; 1 \quad\Longrightarrow\quad c_1^2 + c_2^2 \;\geqslant\; 1 - 2|c_1 c_2| \mu \;\geqslant\; 1 - 4\mu,$$

where the last inequality follows from Lemma B.7. Thus, we observe

$$\begin{aligned}
\left\langle \frac{a_1}{\|a_1\|}, v \right\rangle^2 + \left\langle \frac{a_2}{\|a_2\|}, v \right\rangle^2 &= \left( c_1 + c_2 \left\langle \frac{a_1}{\|a_1\|}, \frac{a_2}{\|a_2\|} \right\rangle \right)^2 + \left( c_2 + c_1 \left\langle \frac{a_1}{\|a_1\|}, \frac{a_2}{\|a_2\|} \right\rangle \right)^2 \\
&= \left( c_1^2 + c_2^2 \right) + \left( c_1^2 + c_2^2 \right) \left\langle \frac{a_1}{\|a_1\|}, \frac{a_2}{\|a_2\|} \right\rangle^2 + 4 c_1 c_2 \left\langle \frac{a_1}{\|a_1\|}, \frac{a_2}{\|a_2\|} \right\rangle \\
&\geqslant\; 1 - 4\mu - (1 - 4\mu)\mu^2 - 4 \frac{1+\mu}{1-\mu^2} \mu \\
&\geqslant\; 1 - 10\mu
\end{aligned}$$

By the fact in Equation (B.20) and combining all the bounds above we obtain

$$\boldsymbol{v}^\top \operatorname{Hess} \varphi_{\mathrm{T}}(\boldsymbol{q})\boldsymbol{v} \;\leqslant\; \left[1 - 3\left(1 - 2\xi^{-3/2}M^3\right)^2 (1 - 10\mu)\right] \|\boldsymbol{\zeta}\|_4^4 \;\leqslant\; -\frac{1}{4}\|\boldsymbol{\zeta}\|_4^4.$$

This completes the proof. ∎

**Lemma B.7** *Suppose* $\left|\left\langle \frac{\boldsymbol{a}_1}{\|\boldsymbol{a}_1\|}, \frac{\boldsymbol{a}_1}{\|\boldsymbol{a}_1\|}\right\rangle\right| \leqslant \mu$ *with* $\mu < 1/2$. *Let* $\boldsymbol{v} \in \operatorname{span}\left\{\frac{\boldsymbol{a}_1}{\|\boldsymbol{a}_1\|}, \frac{\boldsymbol{a}_2}{\|\boldsymbol{a}_2\|}\right\}$ *such that* $\|\boldsymbol{v}\| = 1$ *and* $\boldsymbol{v} = c_1 \frac{\boldsymbol{a}_1}{\|\boldsymbol{a}_1\|} + c_2 \frac{\boldsymbol{a}_2}{\|\boldsymbol{a}_2\|}$, *then we have*

$$|c_1 c_2| \;\leqslant\; \frac{1+\mu}{1-\mu^2},$$

**Proof** By the fact that $\left|\left\langle \boldsymbol{v}, \frac{\boldsymbol{a}_1}{\|\boldsymbol{a}_1\|}\right\rangle \left\langle \boldsymbol{v}, \frac{\boldsymbol{a}_2}{\|\boldsymbol{a}_2\|}\right\rangle\right| \leqslant 1$, we have

$$\left|\left(c_1 + c_2\left\langle \frac{\boldsymbol{a}_1}{\|\boldsymbol{a}_1\|}, \frac{\boldsymbol{a}_2}{\|\boldsymbol{a}_2\|}\right\rangle\right)\left(c_2 + c_1\left\langle \frac{\boldsymbol{a}_1}{\|\boldsymbol{a}_1\|}, \frac{\boldsymbol{a}_2}{\|\boldsymbol{a}_2\|}\right\rangle\right)\right| \;\leqslant\; 1,$$

which further implies that

$$\left|c_1 c_2 + \left(c_1^2 + c_2^2\right)\left\langle \frac{\boldsymbol{a}_1}{\|\boldsymbol{a}_1\|}, \frac{\boldsymbol{a}_2}{\|\boldsymbol{a}_2\|}\right\rangle + c_1 c_2 \left\langle \frac{\boldsymbol{a}_1}{\|\boldsymbol{a}_1\|}, \frac{\boldsymbol{a}_2}{\|\boldsymbol{a}_2\|}\right\rangle^2\right| \;\leqslant\; 1.$$

Since $\|\boldsymbol{v}\| = 1$, we also have

$$c_1^2 + c_2^2 \;=\; 1 - 2c_1 c_2 \left\langle \frac{\boldsymbol{a}_1}{\|\boldsymbol{a}_1\|}, \frac{\boldsymbol{a}_2}{\|\boldsymbol{a}_2\|}\right\rangle.$$

Combining the two (in)equalities above, we obtain

$$\begin{aligned}
1 &\geqslant \left|c_1 c_2 + \left\langle \frac{\boldsymbol{a}_1}{\|\boldsymbol{a}_1\|}, \frac{\boldsymbol{a}_2}{\|\boldsymbol{a}_2\|}\right\rangle - c_1 c_2 \left\langle \frac{\boldsymbol{a}_1}{\|\boldsymbol{a}_1\|}, \frac{\boldsymbol{a}_2}{\|\boldsymbol{a}_2\|}\right\rangle^2\right| \\
&\geqslant |c_1 c_2|\left(1 - \left\langle \frac{\boldsymbol{a}_1}{\|\boldsymbol{a}_1\|}, \frac{\boldsymbol{a}_2}{\|\boldsymbol{a}_2\|}\right\rangle^2\right) - \left|\left\langle \frac{\boldsymbol{a}_1}{\|\boldsymbol{a}_1\|}, \frac{\boldsymbol{a}_2}{\|\boldsymbol{a}_2\|}\right\rangle\right| \geqslant |c_1 c_2|\left(1 - \mu^2\right) - \mu.
\end{aligned}$$

Thus, we obtain the desired result. ∎

## B.2 Negative Curvature in $\mathcal{R}_{\mathrm{N}}$

Finally, we make more stringent assumption on $\boldsymbol{A}$ that each column of $\boldsymbol{A}$ is $\ell^2$ normalized, i.e.,

$$\|\boldsymbol{a}_i\| \;=\; 1, \quad 1 \leqslant i \leqslant m.$$

We show that the function $\varphi_{\mathrm{T}}(\boldsymbol{q})$ exhibits negative curvature in the region $\mathcal{R}_{\mathrm{N}}$. Namely, the Riemannian Hessian for any points $\boldsymbol{q} \in \mathcal{R}_{\mathrm{N}}$ has a negative eigenvalue, such that the Hessian is negative in a certain direction.

**Lemma B.8** *Suppose each column of* $\boldsymbol{A}$ *is* $\ell^2$ *normalized and*

$$K \;\leqslant\; 3\left(1 + 6\mu + 6\xi^{3/5}\mu^{2/5}\right)^{-1}.$$

*For any point* $\boldsymbol{q} \in \mathcal{R}_{\mathrm{N}}$, *there exists some direction* $\boldsymbol{d} \in \mathbb{S}^{n-1}$, *such that*

$$\boldsymbol{d}^\top \operatorname{Hess} \varphi_{\mathrm{T}}(\boldsymbol{q})\boldsymbol{d} \;<\; -4\|\boldsymbol{\zeta}\|_4^4 \|\boldsymbol{\zeta}\|_\infty^2.$$

**Proof** By definition, we have

$$
\begin{aligned}
& \boldsymbol{a}_1^\top \operatorname{Hess} \varphi_{\mathrm{T}}(\boldsymbol{q}) \boldsymbol{a}_1 \\
={} & -3 \boldsymbol{a}_1^\top \boldsymbol{P}_{\boldsymbol{q}^\perp} \boldsymbol{A} \operatorname{diag}\left(\boldsymbol{\zeta}^{\odot 2}\right) \boldsymbol{A}^* \boldsymbol{P}_{\boldsymbol{q}^\perp} \boldsymbol{a}_1 + \|\boldsymbol{\zeta}\|_4^4 \left\|\boldsymbol{P}_{\boldsymbol{q}^\perp} \boldsymbol{a}_1\right\|^2 \\
={} & -3 \boldsymbol{a}_1^\top \boldsymbol{A} \operatorname{diag}\left(\boldsymbol{\zeta}^{\odot 2}\right) \boldsymbol{A}^\top \boldsymbol{a}_1 + 6 \|\boldsymbol{\zeta}\|_\infty \boldsymbol{\zeta}^\top \operatorname{diag}\left(\boldsymbol{\zeta}^{\odot 2}\right) \boldsymbol{A}^\top \boldsymbol{a}_1 - 3 \|\boldsymbol{\zeta}\|_\infty^2 \|\boldsymbol{\zeta}\|_4^4 + \|\boldsymbol{\zeta}\|_4^4 \left(\|\boldsymbol{a}_1\|^2 - \|\boldsymbol{\zeta}\|_\infty^2\right) \\
\leqslant{} & -3 \|\boldsymbol{\zeta}\|_\infty^2 \|\boldsymbol{a}_1\|^4 + 6 \|\boldsymbol{\zeta}\|_\infty^4 \|\boldsymbol{a}_1\|^2 + 6\mu \|\boldsymbol{\zeta}\|_\infty \|\boldsymbol{\zeta}\|_3^3 - 3 \|\boldsymbol{\zeta}\|_\infty^2 \|\boldsymbol{\zeta}\|_4^4 + \|\boldsymbol{a}_1\|^2 \|\boldsymbol{\zeta}\|_4^4 - \|\boldsymbol{\zeta}\|_\infty^2 \|\boldsymbol{\zeta}\|_4^4 \\
={} & -3 \|\boldsymbol{\zeta}\|_\infty^2 + 6 \|\boldsymbol{\zeta}\|_\infty^4 + 6\mu \|\boldsymbol{\zeta}\|_\infty \|\boldsymbol{\zeta}\|_3^3 - 4 \|\boldsymbol{\zeta}\|_\infty^2 \|\boldsymbol{\zeta}\|_4^4 + \|\boldsymbol{\zeta}\|_4^4 \\
\leqslant{} & \|\boldsymbol{\zeta}\|_\infty^2 \left(-3 + 6 \|\boldsymbol{\zeta}\|_\infty^2 + 6\mu \|\boldsymbol{\zeta}\|^2 - 4 \|\boldsymbol{\zeta}\|_4^4 + \|\boldsymbol{\zeta}\|^2\right) \\
={} & \|\boldsymbol{\zeta}\|_\infty^2 \left(-3 + 6 \|\boldsymbol{\zeta}\|_\infty^2 + 6\mu K - 4 \|\boldsymbol{\zeta}\|_4^4 + K\right)
\end{aligned}
$$

where for the second inequality we used the fact that $\|\boldsymbol{\zeta}\|_4^4 \leqslant \|\boldsymbol{\zeta}\|_\infty^2 \|\boldsymbol{\zeta}\|^2$, and for the last equality we applied that $\|\boldsymbol{\zeta}\|^2 = \boldsymbol{q}^\top \boldsymbol{A} \boldsymbol{A}^\top \boldsymbol{q} = K$. Moreover, as $\boldsymbol{q} \in \mathcal{R}_{\mathrm{N}}$, we have

$$
\|\boldsymbol{\zeta}\|_\infty^2 \ \leqslant\ \|\boldsymbol{\zeta}\|_4^2 \ \leqslant\ \xi^{1/2} \mu^{1/3} \|\boldsymbol{\zeta}\|_3
$$

$$
\|\boldsymbol{\zeta}\|_3 \ =\ \left(\sum_{k=1}^m |\zeta_k|^3\right)^{1/3} \ \leqslant\ \|\boldsymbol{\zeta}\|_\infty^{1/3} K^{1/3}.
$$

Thus, we obtain

$$
\|\boldsymbol{\zeta}\|_\infty^2 \ \leqslant\ \xi^{1/2} \mu^{1/3} \|\boldsymbol{\zeta}\|_\infty^{1/3} K^{1/3} \quad\Longrightarrow\quad \|\boldsymbol{\zeta}\|_\infty^2 \ \leqslant\ \xi^{3/5} (\mu K)^{2/5}.
$$

Hence, we have

$$
\boldsymbol{a}_1^\top \operatorname{Hess} \varphi_{\mathrm{T}}(\boldsymbol{q}) \boldsymbol{a}_1 \ \leqslant\ \|\boldsymbol{\zeta}\|_\infty^2 \left(-3 + 6\xi^{3/5} (\mu K)^{2/5} + 6\mu K - 4 \|\boldsymbol{\zeta}\|_4^4 + K\right) \ \leqslant\ -4 \|\boldsymbol{\zeta}\|_4^4 \|\boldsymbol{\zeta}\|_\infty^2,
$$

whenever

$$
K \ \leqslant\ 3 \left(1 + 6\mu + 6\xi^{3/5} \mu^{2/5}\right)^{-1}.
$$

Thus, we obtain the desired result. ∎

## C   OPTIMIZATION LANDSCAPE IN FINITE SAMPLE

In this section, we will show that the finite sample objective functions in the overcomplete dictionary learning and convolutional dictionary learning have similar geometric properties as $\varphi_{\mathrm{T}}(\boldsymbol{q}) = -\frac{1}{4} \|\boldsymbol{A}^\top \boldsymbol{q}\|_4^4$ analyzed in Appendix B. Specifically, we will analyze the geometric properties of objective function $\varphi(\boldsymbol{q})$ (which could be $\varphi_{\mathrm{DL}}(\boldsymbol{q})$ and $\varphi_{\mathrm{CDL}}(\boldsymbol{q})$) whose gradient and Hessian are close to $\varphi_{\mathrm{T}}(\boldsymbol{q})$. We denote by

$$
\begin{aligned}
\boldsymbol{\delta}_g(\boldsymbol{q}) &:= \operatorname{grad} \varphi(\boldsymbol{q}) - \operatorname{grad} \varphi_{\mathrm{T}}(\boldsymbol{q}), \\
\boldsymbol{\Delta}_H(\boldsymbol{q}) &:= \operatorname{Hess} \varphi(\boldsymbol{q}) - \operatorname{Hess} \varphi_{\mathrm{T}}(\boldsymbol{q}),
\end{aligned} \tag{C.1}
$$

both of which will be proved to be small for overcomplete dictionary learning and convolutional dictionary learning in Appendix F.

### C.1   GEOMETRIC ANALYSIS OF CRITICAL POINTS IN $\mathcal{R}_{\mathrm{C}}$

**Proposition C.1** *Assume*

$$
\|\boldsymbol{\delta}_g(\boldsymbol{q})\| \leqslant \mu M \|\boldsymbol{\zeta}\|_3^3 \quad\text{and}\quad \|\boldsymbol{\Delta}_H(\boldsymbol{q})\| < \frac{1}{20} \|\boldsymbol{\zeta}\|_4^4.
$$

*Also suppose we have*

$$
KM \ <\ 8^{-1} \cdot \xi^{3/2}, \quad M^3 \ <\ 2\eta \cdot \xi^{3/2}, \quad \mu \ <\ \frac{1}{20} \tag{C.2}
$$

*for some constant $\eta < 2^{-6}$. Then any critical point $\boldsymbol{q} \in \mathcal{R}_C$, with $\operatorname{grad} \varphi(\boldsymbol{q}) = 0$, either is a ridable (strict) saddle point, or it satisfies second-order optimality condition and is near one of the components e.g., $\boldsymbol{a}_1$ in the sense that*

$$\left\langle \frac{\boldsymbol{a}_1}{\|\boldsymbol{a}_1\|}, \boldsymbol{q} \right\rangle \geqslant 1 - 5\xi^{-3/2}M^3 \geqslant 1 - 5\eta. \tag{C.3}$$

**Proof** [Proof of Proposition C.1] With the same argument in Lemma B.2, we have that any critical point $\boldsymbol{q} \in \mathbb{S}^{n-1}$ satisfies

$$f(\zeta_i) = \zeta_i^3 - \alpha_i \zeta_i + \beta_i' = 0,$$

for all $i \in [m]$ with $\boldsymbol{\zeta} = \boldsymbol{A}^\top \boldsymbol{q}$, where

$$\alpha_i = \frac{\|\boldsymbol{\zeta}\|_4^4}{\|\boldsymbol{a}_i\|^2}, \qquad \beta_i' = \frac{\langle \boldsymbol{\delta}_g(\boldsymbol{q}), \boldsymbol{a}_i \rangle + \sum_{j \neq i} \langle \boldsymbol{a}_i, \boldsymbol{a}_j \rangle \zeta_j^3}{\|\boldsymbol{a}_i\|^2} = \beta_i + \frac{\langle \boldsymbol{\delta}_g(\boldsymbol{q}), \boldsymbol{a}_i \rangle}{\|\boldsymbol{a}_i\|^2}, \tag{C.4}$$

with $\beta_i = \frac{\sum_{j \neq i} \langle \boldsymbol{a}_i, \boldsymbol{a}_j \rangle \zeta_j^3}{\|\boldsymbol{a}_i\|^2}$ which is defined in equation B.7.

Recall that a widely used upper bound for $\beta_i$ in Appendix B.1 is:

$$|\beta_i| = \frac{\left| \sum_{j \neq i} \langle \boldsymbol{a}_i, \boldsymbol{a}_j \rangle \zeta_j^3 \right|}{\|\boldsymbol{a}_i\|^2} \leqslant \frac{\mu M \|\boldsymbol{\zeta}\|_3^3}{\|\boldsymbol{a}_i\|},$$

which together with $\|\boldsymbol{\delta}_g(\boldsymbol{q})\| \leqslant \mu M \|\boldsymbol{\zeta}\|_3^3$ gives

$$\beta' = \beta_i + \frac{\langle \boldsymbol{\delta}_g(\boldsymbol{q}), \boldsymbol{a}_i \rangle}{\|\boldsymbol{a}_i\|^2} \leqslant 2 \frac{\mu M \|\boldsymbol{\zeta}\|_3^3}{\|\boldsymbol{a}_i\|}. \tag{C.5}$$

To easily utilize the proofs in Appendix B.1, we define $\xi' = 2^{-2/3}\xi$ such that $\xi'^{-3/2} = 2\xi^{-3/2}$. Plugging the assumption $M^3 \xi'^{-3/2} \leqslant \frac{1}{4}$ into equation C.5, we have

$$\frac{|\beta_i'|}{\alpha_i^{3/2}} \leqslant 2 \frac{\mu M \|\boldsymbol{\zeta}\|_3^3 \|\boldsymbol{a}_i\|^2}{\|\boldsymbol{\zeta}\|_4^6} \leqslant 2 \frac{\mu M^3 \|\boldsymbol{\zeta}\|_3^3}{\|\boldsymbol{\zeta}\|_4^6} \leqslant 2 M^3 \xi^{-3/2} \leqslant 2 M^3 \xi'^{-3/2} \leqslant \frac{1}{4}.$$

This implies that the condition in equation B.9 holds, so that we can apply Lemma B.3 based on which we classify critical points $\boldsymbol{q} \in \mathcal{R}_C$ into three categories

1. All $|\zeta_i|$ $(1 \leqslant i \leqslant m)$ are smaller than $\frac{2|\beta_i'|}{\alpha_i}$;

2. Only $|\zeta_1|$ is larger than $\frac{2|\beta_1'|}{\alpha_1}$;

3. At least $|\zeta_1|$ and $|\zeta_2|$ are larger than $\frac{2|\beta_1'|}{\alpha_1}$ and $\frac{2|\beta_2'|}{\alpha_2}$, respectively.

For Case 1, using the same argument as in Lemma B.4 we can easily show that this type of critical point does not exist. For Case 2, with the same argument as in Lemma B.5, we obtain that such a critical point is near one of the target solution with

$$\left\langle \frac{\boldsymbol{a}_1}{\|\boldsymbol{a}_1\|}, \boldsymbol{q} \right\rangle \geqslant 1 - 5\xi'^{-3/2}M^3 \geqslant 1 - 5\eta,$$

and satisfies the second-order optimality condition, i.e., for any $\boldsymbol{v} \in \mathbb{S}^{n-1}$ with $\boldsymbol{v} \perp \boldsymbol{q}$, we have

$$\boldsymbol{v}^\top \operatorname{Hess} \varphi(\boldsymbol{q}) \boldsymbol{v} \geqslant \boldsymbol{v}^\top \operatorname{Hess} \varphi_T(\boldsymbol{q}) \boldsymbol{v} - \|\boldsymbol{\Delta}_H(\boldsymbol{q})\| \geqslant \frac{1}{20} \|\boldsymbol{\zeta}\|_4^4 - \|\boldsymbol{\Delta}_H(\boldsymbol{q})\|.$$

Finally, for Case 3, with the same $\boldsymbol{v}$ constructed in Lemma B.6 and using the assumption $\|\boldsymbol{\Delta}_H(\boldsymbol{q})\| < \frac{1}{20} \|\boldsymbol{\zeta}\|_4^4$, we have

$$\boldsymbol{v}^\top \operatorname{Hess} \varphi(\boldsymbol{q}) \boldsymbol{v} \leqslant \boldsymbol{v}^\top \operatorname{Hess} \varphi_T(\boldsymbol{q}) \boldsymbol{v} + \|\boldsymbol{\Delta}_H(\boldsymbol{q})\| \leqslant -\|\boldsymbol{\zeta}\|_4^4 + \|\boldsymbol{\Delta}_H(\boldsymbol{q})\| < 0,$$

indicating that this type of critical points $\boldsymbol{q} \in \mathcal{R}_C$ is ridable saddle, for which the Riemannian Hessian exhibits negative eigenvalue. Therefore, the critical points in $\mathcal{R}_C$ are either ridable saddle or near target solutions, so that there is no spurious local minimizer in $\mathcal{R}_C$. ∎

## C.2 NEGATIVE CURVATURE IN $\mathcal{R}_N$

By directly using Lemma B.8, we obtain the negative curvature of $\varphi(\boldsymbol{q})$ in $\mathcal{R}_N$.

**Lemma C.2** *Assume*

$$\|\boldsymbol{\Delta}_H(\boldsymbol{q})\| < \|\boldsymbol{\zeta}\|_4^4 \|\boldsymbol{\zeta}\|_\infty^2 .$$

*Also suppose each column of $\boldsymbol{A}$ is $\ell^2$ normalized and*

$$K \leqslant 3 \left( 1 + 6\mu + 6\xi^{3/5}\mu^{2/5} \right)^{-1} .$$

*For any point $\boldsymbol{q} \in \mathcal{R}_N$, there exists some direction $\boldsymbol{d} \in \mathbb{S}^{n-1}$, such that*

$$\boldsymbol{d}^\top \operatorname{Hess} \varphi(\boldsymbol{q})\boldsymbol{d} < -3 \|\boldsymbol{\zeta}\|_4^4 \|\boldsymbol{\zeta}\|_\infty^2 .$$

**Proof** First, it follows Lemma B.8 that for any point $\boldsymbol{q} \in \mathcal{R}_N$, there exists some direction $\boldsymbol{d} \in \mathbb{S}^{n-1}$, such that

$$\boldsymbol{d}^\top \operatorname{Hess} \varphi_T(\boldsymbol{q})\boldsymbol{d} < -4 \|\boldsymbol{\zeta}\|_4^4 \|\boldsymbol{\zeta}\|_\infty^2 ,$$

which together with the assumption $\|\boldsymbol{\Delta}_H(\boldsymbol{q})\| < \|\boldsymbol{\zeta}\|_4^4 \|\boldsymbol{\zeta}\|_\infty^2$ and the fact $\boldsymbol{d}^\top \operatorname{Hess} \varphi(\boldsymbol{q})\boldsymbol{d} = \boldsymbol{d}^\top \operatorname{Hess} \varphi_T(\boldsymbol{q})\boldsymbol{d} + \boldsymbol{d}^\top \boldsymbol{\Delta}_H(\boldsymbol{q})\boldsymbol{d} \leqslant \boldsymbol{d}^\top \operatorname{Hess} \varphi_T(\boldsymbol{q})\boldsymbol{d} + \|\boldsymbol{\Delta}_H(\boldsymbol{q})\|$ completes the proof. ∎

## D OVERCOMPLETE DICTIONARY LEARNING

In this section, we consider the nonconvex problem of

$$\min_{\boldsymbol{q}} \ \varphi_{\mathrm{DL}}(\boldsymbol{q}) = -\frac{1}{12\theta(1-\theta)p} \left\| \boldsymbol{q}^\top \boldsymbol{Y} \right\|_4^4 = -\frac{1}{12\theta(1-\theta)p} \left\| \boldsymbol{q}^\top \boldsymbol{A}\boldsymbol{X} \right\|_4^4, \quad \text{s.t.} \quad \|\boldsymbol{q}\| = 1.$$

We characterize its expectation and optimization landscape as follows.

### D.1 EXPECTATION CASE: OVERCOMPLETE TENSOR DECOMPOSITION

First, we show that $\varphi_{\mathrm{DL}}(\boldsymbol{q})$ reduces to $\varphi_T(\boldsymbol{q})$ in expectation w.r.t. $\boldsymbol{X}$.

**Lemma D.1** *When $\boldsymbol{X}$ is i.i.d. drawn from Bernoulli Gaussian distribution as in Assumption 2.2, then we have*

$$\mathbb{E}_{\boldsymbol{X}} \left[ \varphi_{\mathrm{DL}}(\boldsymbol{q}) \right] = \varphi_T(\boldsymbol{q}) - \frac{\theta}{2(1-\theta)} \left( \frac{m}{n} \right)^2 .$$

**Proof** Let $\boldsymbol{\zeta} = \boldsymbol{A}^\top \boldsymbol{q} \in \mathbb{R}^m$ with $\|\boldsymbol{\zeta}\|^2 = \frac{m}{n}$. By using the fact that

$$\boldsymbol{X} = \begin{bmatrix} \boldsymbol{x}_1 & \boldsymbol{x}_2 & \cdots & \boldsymbol{x}_p \end{bmatrix}, \quad \boldsymbol{x}_k = \boldsymbol{b}_k \odot \boldsymbol{g}_k, \ \boldsymbol{b}_k \sim \operatorname{Ber}(\theta), \ \boldsymbol{g}_k \sim \mathcal{N}(\boldsymbol{0}, \boldsymbol{I}),$$

we observe

$$\mathbb{E}_{\boldsymbol{X}} \left[ \varphi_{\mathrm{DL}}(\boldsymbol{q}) \right] = -\frac{1}{12(1-\theta)\theta p} \mathbb{E}_{\boldsymbol{X}} \left[ \left\| \boldsymbol{\zeta}^\top \boldsymbol{X} \right\|_4^4 \right] = -\frac{1}{12(1-\theta)\theta p} \sum_{k=1}^p \mathbb{E}_{\boldsymbol{x}_k} \left[ \left( \boldsymbol{\zeta}^\top \boldsymbol{x}_k \right)^4 \right]$$

$$= -\frac{1}{12(1-\theta)\theta} \mathbb{E}_{\boldsymbol{b},\boldsymbol{g}} \left[ \langle \boldsymbol{\zeta} \odot \boldsymbol{b}, \boldsymbol{g} \rangle^4 \right]$$

$$= -\frac{1}{4(1-\theta)\theta} \mathbb{E}_{\boldsymbol{b}} \left[ \|\boldsymbol{\zeta} \odot \boldsymbol{b}\|^4 \right].$$

Write $\|\boldsymbol{z} \odot \boldsymbol{b}\|^2 = \sum_{k=1}^m (z_k b_k)^2$, we obtain

$$\mathbb{E}_{\boldsymbol{X}} \left[ \varphi_{\mathrm{DL}}(\boldsymbol{q}) \right] = -\frac{1}{4(1-\theta)\theta} \mathbb{E}_{\boldsymbol{b}} \left[ \left( \sum_{k=1}^m (z_k b_k)^2 \right)^2 \right] = -\frac{1}{4(1-\theta)} \sum_{k=1}^m z_k^4 - \frac{\theta}{2(1-\theta)} \sum_{i \neq j} \zeta_i^2 z_j^2$$

$$= -\frac{1}{4} \|\boldsymbol{z}\|_4^4 - \frac{\theta}{2(1-\theta)} \|\boldsymbol{z}\|^4$$

$$= \varphi_T(\boldsymbol{q}) - \frac{\theta}{2(1-\theta)} \left( \frac{m}{n} \right)^2 ,$$

as desired. ∎

## D.2 MAIN GEOMETRIC RESULT

Combining Proposition C.1 and Lemma C.2 together with the concentration results of the gradient and Hessian in Proposition F.3 and Proposition F.6, we obtain the following geometry results of overcomplete dictionary learning.

**Theorem D.2** *Suppose $\boldsymbol{A}$ satisfies Equation (2.2) and $\boldsymbol{X} \in \mathbb{R}^{m \times p}$ follows $\mathcal{BG}(\theta)$ with $\theta \in \left(\frac{1}{m}, \frac{1}{2}\right)$. Also suppose we have*

$$K \; < \; \max\left\{8^{-1} \cdot \xi^{3/2}, 3\left(1 + 6\mu + 6\xi^{3/5}\mu^{2/5}\right)^{-1}\right\}, \quad 1 \; < \; 2\eta \cdot \xi^{3/2}, \quad \mu \; < \; \frac{1}{20}$$

*for some constant $\eta < 2^{-6}$.*

- *If $p \geqslant C\theta K^3 n^3 \max\left\{\frac{\log(\theta n^{7/2}/\mu)}{\mu^2}, Kn^2 \log(\theta n^2)\right\}$, then with probability at least $1 - cp^{-2}$, any critical point $\boldsymbol{q} \in \mathcal{R}_{\mathrm{C}}$ of $\varphi_{\mathrm{DL}}(\boldsymbol{q})$ either is a ridable (strict) saddle point, or it satisfies second-order optimality condition and is near one of the components e.g., $\boldsymbol{a}_1$ in the sense that*

$$\left\langle \frac{\boldsymbol{a}_1}{\|\boldsymbol{a}_1\|}, \boldsymbol{q} \right\rangle \; \geqslant \; 1 - 5\xi^{-3/2}M^3 \; \geqslant \; 1 - 5\eta.$$

- *If $p \geqslant C\theta K^4 n^6 \log(\theta n^5)$, then with probability at least $1 - cp^{-2}$, any critical point $\boldsymbol{q} \in \mathcal{R}_{\mathrm{N}}$ of $\varphi_{\mathrm{DL}}(\boldsymbol{q})$ is a ridable (strict) saddle point.*

*Here, $c, C > 0$ are some numerical constants.*

**Proof** First note that for overcomplete dictionary $\boldsymbol{A}$ in Equation (2.2), it satisfies Equation (F.9) with $M = 1$. Now it follows from Proposition F.3 and Proposition F.6 that when

$$p \geqslant C\theta K^5 n^2 \max\left\{\frac{\log(\theta Kn/\mu \|\boldsymbol{\zeta}\|_3^3)}{\mu^2 \|\boldsymbol{\zeta}\|_3^6}, \frac{Kn \log(\theta Kn/ \|\boldsymbol{\zeta}\|_4^4)}{\|\boldsymbol{\zeta}\|_4^8}\right\}, \tag{D.1}$$

then with probability at least $1 - cp^{-2}$,

$$\sup_{\boldsymbol{q} \in \mathbb{S}^{n-1}} \|\mathrm{grad}\, \varphi_{\mathrm{DL}}(\boldsymbol{q}) - \mathrm{grad}\, \varphi_{\mathrm{T}}(\boldsymbol{q})\| \leqslant \mu M \|\boldsymbol{\zeta}\|_3^3,$$

$$\sup_{\boldsymbol{q} \in \mathbb{S}^{n-1}} \|\mathrm{Hess}\, \varphi_{\mathrm{DL}}(\boldsymbol{q}) - \mathrm{Hess}\, \varphi_{\mathrm{T}}(\boldsymbol{q})\| < \frac{1}{20} \|\boldsymbol{\zeta}\|_4^4,$$

which together with Proposition C.1 implies that any critical point $\boldsymbol{q} \in \mathcal{R}_{\mathrm{C}}$ of $\varphi_{\mathrm{DL}}(\boldsymbol{q})$ either is a ridable (strict) saddle point, or it satisfies second-order optimality condition and is near one of the components e.g., $\boldsymbol{a}_1$ in the sense that

$$\left\langle \frac{\boldsymbol{a}_1}{\|\boldsymbol{a}_1\|}, \boldsymbol{q} \right\rangle \; \geqslant \; 1 - 5\eta.$$

We complete the proof for $\boldsymbol{q} \in \mathcal{R}_{\mathrm{C}}$ by plugging inequalities $\|\boldsymbol{\zeta}\|_3 \geqslant m^{-1/6} \|\boldsymbol{\zeta}\|_2 = K^{1/3}n^{-1/6}$ and $\|\boldsymbol{\zeta}\|_4 \geqslant m^{-1/4} \|\boldsymbol{\zeta}\|_2 = K^{1/4}n^{-1/4}$ into Equation (D.1).

Similarly, by Proposition F.6, when

$$p \geqslant C\theta K^6 n^3 \frac{\log(\theta Kn/ \|\boldsymbol{\zeta}\|_4^4 \|\boldsymbol{\zeta}\|_\infty^2)}{\|\boldsymbol{\zeta}\|_4^8 \|\boldsymbol{\zeta}\|_\infty^4}, \tag{D.2}$$

then with probability at least $1 - cp^{-2}$,

$$\sup_{\boldsymbol{q} \in \mathbb{S}^{n-1}} \|\mathrm{Hess}\, \varphi_{\mathrm{DL}}(\boldsymbol{q}) - \mathrm{Hess}\, \varphi_{\mathrm{T}}(\boldsymbol{q})\| < \max \|\boldsymbol{\zeta}\|_\infty^2 \|\boldsymbol{\zeta}\|_4^4,$$

which together with Lemma C.2 implies that any critical point $\boldsymbol{q} \in \mathcal{R}_{\mathrm{N}}$ of $\varphi_{\mathrm{DL}}(\boldsymbol{q})$ either is a ridable (strict) saddle point. The proof is completed by plugging $\|\boldsymbol{\zeta}\|_\infty \geqslant n^{-1/2}$ into Equation (D.2). ∎

# E  CONVOLUTIONAL DICTIONARY LEARNING

In this part of appendix, we provide the detailed analysis for CDL. Recall from Section 3, we denote

$$\boldsymbol{Y} = \begin{bmatrix} \boldsymbol{C_{y_1}} & \boldsymbol{C_{y_2}} & \cdots & \boldsymbol{C_{y_p}} \end{bmatrix} \in \mathbb{R}^{n \times p}, \qquad \boldsymbol{A_0} = \begin{bmatrix} \boldsymbol{C_{a_1}} & \boldsymbol{C_{a_2}} & \cdots & \boldsymbol{C_{a_K}} \end{bmatrix} \in \mathbb{R}^{n \times m},$$

$$\boldsymbol{x}_i = \begin{bmatrix} \boldsymbol{x}_{i1} \\ \boldsymbol{x}_{i2} \\ \vdots \\ \boldsymbol{x}_{iK} \end{bmatrix} \in \mathbb{R}^m, \quad \boldsymbol{X}_i = \begin{bmatrix} \boldsymbol{C_{x_{i1}}} \\ \boldsymbol{C_{x_{i2}}} \\ \vdots \\ \boldsymbol{C_{x_{iK}}} \end{bmatrix} \in \mathbb{R}^{m \times n}, \quad \boldsymbol{X} = \begin{bmatrix} \boldsymbol{X}_1 & \boldsymbol{X}_2 & \cdots & \boldsymbol{X}_p \end{bmatrix} \in \mathbb{R}^{n \times np},$$

For simplicity we let

$$\boldsymbol{A} = \left( K^{-1} \boldsymbol{A}_0 \boldsymbol{A}_0^\top \right)^{-1/2} \boldsymbol{A}_0, \quad m = nK.$$

Recall from Section 3, for CDL we make the following assumptions on $\boldsymbol{A}_0$, $\boldsymbol{A}$ and $\boldsymbol{X}$.

**Assumption E.1 (Properties of $\boldsymbol{A}_0$ and $\boldsymbol{A}$)** *We assume the matrix $\boldsymbol{A}_0$ has full row rank with*

*minimum singular value:*  $\sigma_{\min}(\boldsymbol{A}_0) > 0,$ *condition number:*  $\kappa(\boldsymbol{A}_0) := \dfrac{\sigma_{\max}(\boldsymbol{A}_0)}{\sigma_{\min}(\boldsymbol{A}_0)}.$

*In addition, we assume the columns of $\boldsymbol{A}$ are mutually incoherent in the sense that*

$$\max_{i \neq j} \left| \left\langle \frac{\boldsymbol{a}_i}{\|\boldsymbol{a}_i\|}, \frac{\boldsymbol{a}_j}{\|\boldsymbol{a}_j\|} \right\rangle \right| \leqslant \mu.$$

**Assumption E.2 (Bernoulli-Gaussian $\boldsymbol{x}_{ik}$)** *We assume entries of $\boldsymbol{x}_{ik} \sim_{i.i.d.} \mathcal{BG}(\theta)$ that*

$$\boldsymbol{x}_{ik} = \boldsymbol{b}_{ik} \odot \boldsymbol{g}_{ik}, \quad \boldsymbol{b}_{ik} \sim_{i.i.d.} \mathrm{Ber}(\theta), \quad \boldsymbol{g}_{ik} \sim_{i.i.d.} \mathcal{N}(\boldsymbol{0}, \boldsymbol{I}), \quad 1 \leqslant i \leqslant p, \, 1 \leqslant k \leqslant K.$$

In comparison with Assumption 2.1, it should be noted that the preconditioning does not necessarily result in $\ell^2$-normalized columns of $\boldsymbol{A}$. But their norms are still bounded in the sense that

$$\|\boldsymbol{a}_k\|^2 \leqslant \|\boldsymbol{A}^\top \boldsymbol{a}_k\| \leqslant \sqrt{K} \|\boldsymbol{a}_k\| \implies \|\boldsymbol{a}_k\| \leqslant \sqrt{K}, \qquad 1 \leqslant k \leqslant nK. \tag{E.1}$$

Because of the unbalanced columns of $\boldsymbol{A}$, unlike the ODL problem, the CDL problem

$$\min_{\boldsymbol{q} \in \mathbb{S}^{n-1}} \varphi_{\mathrm{CDL}}(\boldsymbol{q}) = -\frac{1}{12\theta(1-\theta)np} \left\| \boldsymbol{q}^\top \boldsymbol{P} \boldsymbol{Y} \right\|_4^4 = -\frac{1}{12\theta(1-\theta)p} \left\| \boldsymbol{q}^\top \boldsymbol{P} \boldsymbol{A}_0 \boldsymbol{X} \right\|_4^4$$

does not have global geometric structures in the worst case. But still we can show that the problem is benign in local regions in the following. Moreover, we also show that we can cook up data driven initialization which falls into the local region.

## E.1  MAIN RESULT OF OPTIMIZATION LANDSCAPE

In this part, we show our main result for optimization landscape for CDL. Namely, consider the region introduced in Equation (3.3) as

$$\mathcal{R}_{\mathrm{CDL}} := \left\{ \boldsymbol{q} \in \mathbb{S}^{n-1} \mid \varphi_{\mathrm{T}}(\boldsymbol{q}) \leqslant -\xi_{\mathrm{CDL}} \, \kappa^{4/3} \mu^{2/3} \|\boldsymbol{\zeta}(\boldsymbol{q})\|_3^2 \right\},$$

where $\xi_{\mathrm{CDL}} > 0$ is a fixed numerical constant. We show the following result.

**Theorem E.3 (Local geometry of nonconvex landscape for CDL)** *Let $C_0 > 5$ be some constant and $\eta < 2^{-6}$. Suppose we have*

$$\theta \in \left( \frac{1}{nK}, \frac{1}{3} \right), \qquad \xi_{\mathrm{CDL}} = C_0 \cdot \eta^{-2/3} K, \quad \mu < \frac{1}{40}, \quad K < C_0,$$

*and we assume Assumption E.1 and Assumption E.2 hold. There exists some constant $C > 0$, with probability at least $1 - c_1(nK)^{-c_2}$ over the randomness of $\boldsymbol{x}_{ik}$s, whenever*

$$p \geqslant C\theta K^2 \mu^{-2} n^4 \max \left\{ \frac{K^6 \kappa^6(\boldsymbol{A}_0)}{\sigma_{\min}^2(\boldsymbol{A}_0)}, \, n \right\} \log^6(m/\mu),$$

*every critical point $\boldsymbol{q}_c$ of $\varphi_{\mathrm{CDL}}(\boldsymbol{q})$ in $\mathcal{R}_{\mathrm{CDL}}$ is either a strict saddle point that exhibits negative curvature for descent, or it is near one of the target solutions (e.g. $\boldsymbol{a}_1$) such that*

$$\left\langle \frac{\boldsymbol{a}_1}{\|\boldsymbol{a}_1\|}, \boldsymbol{q}_c \right\rangle \geqslant 1 - 5\kappa^{-2}\eta.$$

**Proof** Noting Equation (E.1), we set $M = \sqrt{K}$ in Proposition C.1. It follows from Proposition E.11 that when

$$p \;\geqslant\; C\theta K^4 n^2 \log^5(mK) \max\left\{ \frac{K^6 \kappa^6(\boldsymbol{A}_0)}{\sigma_{\min}^2(\boldsymbol{A}_0)},\; n \right\} \cdot \max\left\{ \frac{\log(\theta K n/\mu K^{1/2} \|\boldsymbol{\zeta}\|_3^3)}{\mu^2 K \|\boldsymbol{\zeta}\|_3^6},\; \frac{\log(\theta K n/\|\boldsymbol{\zeta}\|_4^4)}{\|\boldsymbol{\zeta}\|_4^8} \right\},$$
$$\tag{E.2}$$

then with probability at least $1 - c_1(nK)^{-c_2}$,

$$\sup_{\boldsymbol{q}\in\mathbb{S}^{n-1}} \|\text{grad}\,\varphi_{\text{CDL}}(\boldsymbol{q}) - \text{grad}\,\varphi_{\text{T}}(\boldsymbol{q})\| \;\leqslant\; \mu\sqrt{K} \, \|\boldsymbol{\zeta}\|_3^3,$$

$$\sup_{\boldsymbol{q}\in\mathbb{S}^{n-1}} \|\text{Hess}\,\varphi_{\text{CDL}}(\boldsymbol{q}) - \text{Hess}\,\varphi_{\text{T}}(\boldsymbol{q})\| \;<\; \frac{1}{20} \, \|\boldsymbol{\zeta}\|_4^4.$$

Thus, by using Proposition C.1, we have that any critical point $\boldsymbol{q}_c \in \mathcal{R}_{\text{CDL}}$ of $\varphi_{\text{CDL}}(\boldsymbol{q})$ either is a ridable (strict) saddle point, or it satisfies second-order optimality condition and is near one of the components, e.g., $\boldsymbol{a}_1$ in the sense that

$$\left\langle \frac{\boldsymbol{a}_1}{\|\boldsymbol{a}_1\|}, \boldsymbol{q}_c \right\rangle \;\geqslant\; 1 - 5\xi_{\text{CDL}}^{-3/2} K^{3/2} \kappa^{-2} \;\geqslant\; 1 - 5\eta\kappa^{-2},$$

where we have plugged $M = \sqrt{K}$ and $\xi = \xi_{\text{CDL}}\kappa^{4/3}$ in Equation (C.3). Finally, we complete the proof by using inequalities $\|\boldsymbol{\zeta}\|_3 \geqslant m^{-1/6} \|\boldsymbol{\zeta}\|_2 = K^{1/3} n^{-1/6}$ and $\|\boldsymbol{\zeta}\|_4 \geqslant m^{-1/4} \|\boldsymbol{\zeta}\|_2 = K^{1/4} n^{-1/4}$ in Equation (E.2).

■

### E.2 Proof of Optimization

In the following, we show that with high probability Algorithm 1 with initialization returns an approximate solution of one of the kernels up to a shift.

**Proposition E.4 (Global convergence of Algorithm 1)** *With $m = nK$, suppose*

$$c_1 \frac{\log m}{m} \;\leqslant\; \theta \;\leqslant\; c_2 \frac{\mu^{-2/3}}{\kappa^{4/3} m \log m} \cdot \min\left\{ \frac{\kappa^{4/3}}{\mu^{4/3}},\; \frac{K\mu^{-4}}{m^2 \log m} \right\}. \tag{E.3}$$

*Whenever*

$$p \;\geqslant\; C\theta K^2 \mu^{-2} \max\left\{ \frac{K^6 \kappa^6(\boldsymbol{A}_0)}{\sigma_{\min}^2(\boldsymbol{A}_0)},\; n \right\} n^4 \log^6(m/\mu),$$

*our initialization in Algorithm 1 satisfies*

$$\boldsymbol{q}_{\text{init}} \;\in\; \overline{\mathcal{R}}_{\text{CDL}} \;:=\; \left\{ \boldsymbol{q} \in \mathbb{S}^{n-1} \;\mid\; \varphi_{\text{T}}(\boldsymbol{q}) \leqslant -\xi_{\text{CDL}}\,\mu^{2/3}\kappa^{4/3}K \right\} \;\subset\; \mathcal{R}_{\text{CDL}}, \tag{E.4}$$

*such that all future iterates of Algorithm 1 stays within $\mathcal{R}_{\text{CDL}}$ and converge to an approximate solution (e.g., a circulant shift $\mathsf{s}_\ell[\boldsymbol{a}_{01}]$ of $\boldsymbol{a}_{01}$) in the sense that*

$$\left\| \mathcal{P}_{\mathbb{S}^{n-1}}\left( \boldsymbol{P}^{-1}\boldsymbol{q}_\star \right) - \mathsf{s}_\ell[\boldsymbol{a}_{01}] \right\| \;\leqslant\; \epsilon,$$

*where $\epsilon$ is a small numerical constant.*

**Proof** Note that $\overline{\mathcal{R}}_{\text{CDL}} \subseteq \mathcal{R}_{\text{CDL}}$ is due to the fact that

$$\left\| \boldsymbol{A}^\top \boldsymbol{q} \right\|_3^2 \;\leqslant\; \left\| \boldsymbol{A}^\top \boldsymbol{q} \right\|^2 \;=\; K.$$

We show that the iterates of Algorithm 1 converge to one of the target solutions by the following.

**Initialization falls into $\overline{\mathcal{R}}_{\text{CDL}}$.** From Proposition E.5, taking $\xi = \xi_{\text{CDL}}\kappa^{4/3}$, with $\theta$ satisfies Equation (E.3), whenever

$$p \;\geqslant\; C_1 \frac{K^2}{\mu^{4/3}\theta} \frac{\kappa^{10/3}(\boldsymbol{A}_0)}{\sigma_{\min}^2(\boldsymbol{A}_0)} \log(m),$$

w.h.p. our initialization $\boldsymbol{q}_{\text{init}}$ satisfies $\varphi_{\text{T}}(\boldsymbol{q}_{\text{init}}) \;\leqslant\; -2\xi_{\text{CDL}}\,\mu^{2/3}\kappa^{4/3}K$.

**Iterate stays within the region.** Let $\{q^{(k)}\}$ be the sequence generated by Algorithm 1 with $q^{(0)} = q_{\text{init}}$. From Proposition E.12, we know that whenever

$$p \geqslant C_2 \frac{\theta K^2}{\mu^{4/3}\kappa^{8/3}} \max\left\{\frac{K^6\kappa^6(\boldsymbol{A}_0)}{\sigma_{\min}^2(\boldsymbol{A}_0)}, \ n\right\} n^2 \log\left(\theta n \mu^{-2/3}\kappa^{-4/3}\right) \log^5(mK),$$

we have

$$\sup_{\boldsymbol{q}\in\mathbb{S}^{n-1}} \left|\varphi_{\text{CDL}}(\boldsymbol{q}) - \left(\varphi_{\text{T}}(\boldsymbol{q}) - \frac{\theta}{2(1-\theta)}K^2\right)\right| \leqslant \frac{1}{2}\xi_{\text{CDL}}\,\mu^{2/3}\kappa^{4/3}K,$$

which together with the fact that the sequence $\{q^{(k)}\}$ satisfies $\varphi_{\text{CDL}}(q^{(k)}) \leqslant \varphi_{\text{CDL}}(q^{(0)})$ implies

$$\varphi_{\text{T}}(\boldsymbol{q}^{(k)}) \leqslant \varphi_{\text{CDL}}(\boldsymbol{q}^{(k)}) + \frac{\theta}{2(1-\theta)}K^2 + \frac{1}{2}\xi_{\text{CDL}}\,\mu^{2/3}\kappa^{4/3}K$$

$$\leqslant \varphi_{\text{CDL}}(\boldsymbol{q}^{(0)}) + \frac{\theta}{2(1-\theta)}K^2 + \frac{1}{2}\xi_{\text{CDL}}\,\mu^{2/3}\kappa^{4/3}K$$

$$\leqslant \varphi_{\text{T}}(\boldsymbol{q}^{(0)}) + \xi_{\text{CDL}}\,\mu^{2/3}\kappa^{4/3}K \ \leqslant\ -\xi_{\text{CDL}}\,\mu^{2/3}\kappa^{4/3}K.$$

**Closeness to the target solution.** From Theorem E.3, we know that whenever

$$p \geqslant C\theta K^2\mu^{-2}n^4 \max\left\{\frac{K^6\kappa^6(\boldsymbol{A}_0)}{\sigma_{\min}^2(\boldsymbol{A}_0)}, \ n\right\} \log^6(m/\mu),$$

the function $\varphi_{\text{CDL}}(\boldsymbol{q})$ has benign optimization landscape, that whenever our method can efficient escape strict saddle points, Algorithm 1 produces a solution $\boldsymbol{q}_\star$ that is close to one of the target solutions (e.g. $\boldsymbol{a}_1$, the first column of $\boldsymbol{A}$) in the sense that

$$\left\langle \frac{\boldsymbol{a}_1}{\|\boldsymbol{a}_1\|}, \boldsymbol{q}_\star \right\rangle \ \geqslant\ 1 - \varepsilon,$$

with $\varepsilon = \kappa^{-2}\eta$. In the following, we show that our final output $\boldsymbol{a}_\star = \mathcal{P}_{\mathbb{S}^{n-1}}\left(\boldsymbol{P}^{-1}\boldsymbol{q}_\star\right)$ should be correspondingly close to a circulant shift of one of the kernels $\{\boldsymbol{a}_{0k}\}_{k=1}^K$. Without loss of generality, suppose $\boldsymbol{q}_\star = \boldsymbol{a}_1$, then the corresponding solution should be $\boldsymbol{a}_{01}$ with zero shift (or in other words, the first column $\boldsymbol{a}_{01}$ of $\boldsymbol{A}_0$). In the following, we make this rigorous. Notice that

$$\left\|\mathcal{P}_{\mathbb{S}^{n-1}}\left(\boldsymbol{P}^{-1}\boldsymbol{q}_\star\right) - \boldsymbol{a}_{01}\right\| \ =\ \left\|\mathcal{P}_{\mathbb{S}^{n-1}}\left(\boldsymbol{P}^{-1}\boldsymbol{q}_\star\right) - \mathcal{P}_{\mathbb{S}^{n-1}}\left(\frac{\boldsymbol{a}_{01}}{\|\boldsymbol{a}_1\|}\right)\right\| \ \leqslant\ 2\|\boldsymbol{a}_1\| \left\|\boldsymbol{P}^{-1}\boldsymbol{q}_\star - \frac{\boldsymbol{a}_{01}}{\|\boldsymbol{a}_1\|}\right\|,$$

where for the last inequality we used Lemma A.12. Next, by triangle inequality, we have

$$\left\|\mathcal{P}_{\mathbb{S}^{n-1}}\left(\boldsymbol{P}^{-1}\boldsymbol{q}_\star\right) - \boldsymbol{a}_{01}\right\|$$

$$\leqslant 2\|\boldsymbol{a}_1\| \left\|\boldsymbol{P}^{-1}\frac{\boldsymbol{a}_1}{\|\boldsymbol{a}_1\|} - \frac{\boldsymbol{a}_{01}}{\|\boldsymbol{a}_1\|}\right\| + 2\|\boldsymbol{a}_1\| \left\|\boldsymbol{P}^{-1}\left(\frac{\boldsymbol{a}_1}{\|\boldsymbol{a}_1\|} - \boldsymbol{q}_\star\right)\right\|$$

$$= 2\left\|\left(\boldsymbol{P}^{-1}\left(K^{-1}\boldsymbol{A}_0\boldsymbol{A}_0^\top\right)^{-1/2} - \boldsymbol{I}\right)\boldsymbol{a}_{01}\right\| + 2\|\boldsymbol{a}_1\| \left\|\boldsymbol{P}^{-1}\left(\frac{\boldsymbol{a}_1}{\|\boldsymbol{a}_1\|} - \boldsymbol{q}_\star\right)\right\|$$

$$\leqslant 2\left\|\left(\frac{1}{\theta mp}\boldsymbol{Y}\boldsymbol{Y}^\top\right)^{1/2}\left(\boldsymbol{A}_0\boldsymbol{A}_0^\top\right)^{-1/2} - \boldsymbol{I}\right\| + 2\sqrt{2}\|\boldsymbol{a}_1\|\,\|\boldsymbol{P}^{-1}\| \sqrt{1 - \left\langle\frac{\boldsymbol{a}_1}{\|\boldsymbol{a}_1\|}, \boldsymbol{q}_\star\right\rangle}.$$

Let $\delta \in (0,1)$ be a small constant. From Lemma E.18 and Corollary E.19, we know that whenever

$$p \geqslant C\theta^{-1}K^3 \frac{\kappa^6(\boldsymbol{A}_0)}{\sigma_{\min}^2(\boldsymbol{A}_0)}\delta^{-2}\log(m),$$

we have

$$\left\|\left(\frac{1}{\theta mp}\boldsymbol{Y}\boldsymbol{Y}^\top\right)^{1/2}\left(\boldsymbol{A}_0\boldsymbol{A}_0^\top\right)^{-1/2} - \boldsymbol{I}\right\| \leqslant \delta, \qquad \|\boldsymbol{P}^{-1}\| \leqslant 2K^{-1/2}\|\boldsymbol{A}_0\|.$$

Therefore, we obtain

$$\left\|\mathcal{P}_{\mathbb{S}^{n-1}}\left(\boldsymbol{P}^{-1}\boldsymbol{q}_\star\right) - \boldsymbol{a}_{01}\right\| \leqslant 2\delta + 4\sqrt{2}\|\boldsymbol{A}_0\|\sqrt{\varepsilon}$$

$$\leqslant 2\delta + 4\sqrt{2}\sqrt{\eta}\sigma_{\max}(\boldsymbol{A}_0)\kappa^{-1} \ \leqslant\ 2\delta + 4\sqrt{2}\sqrt{\eta} \ \leqslant\ \epsilon$$

when $\eta$ is sufficiently small. Here, $\epsilon$ is a small numerical constant. ∎

### E.3 PROOF OF INITIALIZATION

In this subsection, we show that we can cook up a good data-driven initialization. We initialize the problem by using a random sample ($1 \leqslant \ell \leqslant p$)

$$\boldsymbol{q}_{\text{init}} = \mathcal{P}_{\mathbb{S}^{n-1}} \left( \boldsymbol{P} \boldsymbol{y}_\ell \right), \quad 1 \leqslant \ell \leqslant p,$$

which roughly equals to

$$\boldsymbol{q}_{\text{init}} \approx \mathcal{P}_{\mathbb{S}^{n-1}} \left( \boldsymbol{A} \boldsymbol{x}_\ell \right), \quad \boldsymbol{A}^\top \boldsymbol{q}_{\text{init}} \approx \sqrt{K} \mathcal{P}_{\mathbb{S}^{m-1}} \left( \boldsymbol{A}^\top \boldsymbol{A} \boldsymbol{x}_\ell \right).$$

For generic kernels, $\boldsymbol{A}^\top \boldsymbol{A}$ is a close to a diagonal matrix, as the magnitudes of off-diagonal entries are bounded by column mutual incoherence. Hence, the sparse property of $\boldsymbol{x}_\ell$ should be approximately preserved, so that $\boldsymbol{A}^\top \boldsymbol{q}_{\text{init}}$ is spiky with large $\left\| \boldsymbol{A}^\top \boldsymbol{q}_{\text{init}} \right\|_4^4$. We define

$$\boldsymbol{\zeta}_{\text{init}} = \boldsymbol{A}^\top \boldsymbol{q}_{\text{init}}, \qquad \widehat{\boldsymbol{\zeta}}_{\text{init}} = \sqrt{K} \mathcal{P}_{\mathbb{S}^{m-1}} \left( \boldsymbol{A}^\top \boldsymbol{A} \boldsymbol{x}_\ell \right).$$

By leveraging the sparsity level $\theta$, one can make sure that such an initialization $\boldsymbol{q}_{\text{init}}$ suffices.

**Proposition E.5** *Let $m = nK$. Suppose the sparsity level $\theta$ satisfies*

$$c_1 \frac{\log m}{m} \leqslant \theta \leqslant c_2 \frac{K \mu^{-2/3}}{\xi m \log m} \cdot \min \left\{ \frac{\xi}{K \mu^{4/3}}, \frac{\mu^{-4}}{m^2 \log m} \right\}.$$

*Whenever*

$$p \geqslant C \frac{K^2}{\mu^{4/3} \xi^2 \theta} \frac{\kappa^6(\boldsymbol{A}_0)}{\sigma_{\min}^2(\boldsymbol{A}_0)} \log(m),$$

*for some $\xi > 0$ we have*

$$\left\| \boldsymbol{\zeta}_{\text{init}} \right\|_4^4 \geqslant \xi K \mu^{2/3}$$

*holds with probability at least $1 - cm^{-c'}$. Here, $c_1$, $c_2$, $c$, $c'$, $C > 0$ are some numerical constants.*

**Proof** By using the convexity of $\ell^4$-loss, we can show that the values of $\left\| \boldsymbol{\zeta}_{\text{init}} \right\|_4^4$ and $\left\| \widehat{\boldsymbol{\zeta}}_{\text{init}} \right\|_4^4$ are close,

$$\left\| \boldsymbol{\zeta}_{\text{init}} \right\|_4^4 \geqslant \left\| \widehat{\boldsymbol{\zeta}}_{\text{init}} \right\|_4^4 + 4 \left\langle \widehat{\boldsymbol{\zeta}}_{\text{init}}^{\odot 3}, \boldsymbol{\zeta}_{\text{init}} - \widehat{\boldsymbol{\zeta}}_{\text{init}} \right\rangle \geqslant \left\| \widehat{\boldsymbol{\zeta}}_{\text{init}} \right\|_4^4 - 4 \left\| \widehat{\boldsymbol{\zeta}}_{\text{init}}^{\odot 3} \right\| \left\| \boldsymbol{\zeta}_{\text{init}} - \widehat{\boldsymbol{\zeta}}_{\text{init}} \right\|$$

$$\geqslant \left\| \widehat{\boldsymbol{\zeta}}_{\text{init}} \right\|_4^4 - 4 K^{3/2} \underbrace{\left\| \boldsymbol{\zeta}_{\text{init}} - \widehat{\boldsymbol{\zeta}}_{\text{init}} \right\|}_{\text{small}}. \tag{E.5}$$

Thus, it is enough to lower bound $\left\| \widehat{\boldsymbol{\zeta}}_{\text{init}} \right\|_4^4$. Let $\mathcal{I} = \text{supp}(\boldsymbol{x}_\ell)$, and let $\mathcal{P}_{\mathcal{I}} : \mathbb{R}^m \mapsto \mathbb{R}^m$ that maps all off support entries to zero and all on support entries to themselves. Thus, we have

$$\left\| \widehat{\boldsymbol{\zeta}}_{\text{init}} \right\|_4^4 = K^2 \left\| \boldsymbol{A}^\top \boldsymbol{A} \boldsymbol{x}_\ell \right\|^{-4} \left\| \boldsymbol{A}^\top \boldsymbol{A} \boldsymbol{x}_\ell \right\|_4^4$$

$$\geqslant K^2 \left( \left\| \mathcal{P}_{\mathcal{I}} \left( \boldsymbol{A}^\top \boldsymbol{A} \boldsymbol{x}_\ell \right) \right\|^2 + \left\| \mathcal{P}_{\mathcal{I}^c} \left( \boldsymbol{A}^\top \boldsymbol{A} \boldsymbol{x}_\ell \right) \right\|^2 \right)^{-2} \left\| \mathcal{P}_{\mathcal{I}} \left( \boldsymbol{A}^\top \boldsymbol{A} \boldsymbol{x}_\ell \right) \right\|_4^4$$

$$= \frac{K^2}{(1 + \rho)^2} \left\| \mathcal{P}_{\mathbb{S}^{n-1}} \left( \mathcal{P}_{\mathcal{I}} \left( \boldsymbol{A}^\top \boldsymbol{A} \boldsymbol{x}_\ell \right) \right) \right\|_4^4,$$

with $\rho := \left( \frac{\left\| \mathcal{P}_{\mathcal{I}^c} \left( \boldsymbol{A}^\top \boldsymbol{A} \boldsymbol{x}_\ell \right) \right\|}{\left\| \mathcal{P}_{\mathcal{I}} \left( \boldsymbol{A}^\top \boldsymbol{A} \boldsymbol{x}_\ell \right) \right\|} \right)^2$. By Lemma E.7 and Lemma E.9, whenever

$$c_1 \frac{\log m}{m} \leqslant \theta \leqslant c_2 \frac{\mu^{-2}}{m \log m},$$

we have

$$\left\| \mathcal{P}_{\mathcal{I}^c} \left( \boldsymbol{A}^\top \boldsymbol{A} \boldsymbol{x}_\ell \right) \right\| \leqslant C_1 K \mu m \sqrt{\theta \log m}, \qquad \left\| \mathcal{P}_{\mathcal{I}} \left( \boldsymbol{A}^\top \boldsymbol{A} \boldsymbol{x}_\ell \right) \right\| \geqslant \frac{1}{\sqrt{2}} K \sqrt{\theta m}$$

holding with probability at least $1 - c_3 m^{-c_4}$, so that

$$\rho = \left( \frac{\left\| \mathcal{P}_{\mathcal{I}^c} \left( \boldsymbol{A}^\top \boldsymbol{A} \boldsymbol{x}_\ell \right) \right\|}{\left\| \mathcal{P}_{\mathcal{I}} \left( \boldsymbol{A}^\top \boldsymbol{A} \boldsymbol{x}_\ell \right) \right\|} \right)^2 \leqslant C_2 \mu^2 m \log m.$$

Thus, we have

$$\left\| \widehat{\boldsymbol{\zeta}}_{\text{init}} \right\|_4^4 \geqslant K^2 (1 + \rho)^{-2} \left\| \mathcal{P}_{\mathbb{S}^{m-1}} \left( \mathcal{P}_{\mathcal{I}} \boldsymbol{A}^\top \boldsymbol{A} \boldsymbol{x}_\ell \right) \right\|_4^4 \geqslant \frac{C_3 K^2}{\mu^4 m^2 \log^2 m} \left\| \mathcal{P}_{\mathbb{S}^{m-1}} \left( \mathcal{P}_{\mathcal{I}} \boldsymbol{A}^\top \boldsymbol{A} \boldsymbol{x}_\ell \right) \right\|_4^4.$$

By Lemma E.10, we have

$$\left\| \mathcal{P}_{\mathbb{S}^{m-1}} \left( \mathcal{P}_{\mathcal{I}} \boldsymbol{A}^\top \boldsymbol{A} \boldsymbol{x}_\ell \right) \right\|_4^4 \geqslant \frac{1}{2\theta m}$$

with probability at least $1 - c_5 m^{-c_6}$. Thus, with high probability, we have

$$\left\| \widehat{\boldsymbol{\zeta}}_{\text{init}} \right\|_4^4 \geqslant \frac{C_3 K^2}{\mu^4 m^2 \log^2 m} \cdot \frac{1}{2\theta m} \geqslant 2\xi K \mu^{2/3}, \tag{E.6}$$

whenever

$$\theta \leqslant C_4 \frac{K \mu^{-2/3}}{\xi m} \cdot \frac{1}{\mu^4 m^2 \log^2 m}.$$

Finally, Lemma E.6 implies that for any $\delta \in (0, 1)$, whenever

$$p \geqslant C_5 \theta^{-1} K^3 \frac{\kappa^6(\boldsymbol{A}_0)}{\sigma_{\min}^2(\boldsymbol{A}_0)} \delta^{-2} \log(m),$$

it holds that

$$\left\| \boldsymbol{\zeta}_{\text{init}} - \widehat{\boldsymbol{\zeta}}_{\text{init}} \right\| \leqslant \delta,$$

with probability at least $1 - c_7(m)^{-c_8}$. Choose $\delta$ such that

$$4K^{3/2} \left\| \boldsymbol{\zeta}_{\text{init}} - \widehat{\boldsymbol{\zeta}}_{\text{init}} \right\| \leqslant 4K^{3/2} \delta \leqslant \xi K \mu^{2/3} \implies \delta \leqslant C_6 \xi K^{-1/2} \mu^{2/3}, \tag{E.7}$$

then by Equations (E.5) to (E.7) we have

$$\left\| \boldsymbol{\zeta}_{\text{init}} \right\|_4^4 \geqslant \left\| \widehat{\boldsymbol{\zeta}}_{\text{init}} \right\|_4^4 - 4K^{3/2} \left\| \boldsymbol{\zeta}_{\text{init}} - \widehat{\boldsymbol{\zeta}}_{\text{init}} \right\| \geqslant \xi K \mu^{2/3}.$$

Summarizing all the result above, we obtain the desired result. ∎

**Lemma E.6** *Let $\delta \in (0, 1)$. Whenever*

$$p \geqslant C \theta^{-1} K^3 \frac{\kappa^6(\boldsymbol{A}_0)}{\sigma_{\min}^2(\boldsymbol{A}_0)} \delta^{-2} \log(m),$$

*we have*

$$\left\| \boldsymbol{\zeta}_{\text{init}} - \widehat{\boldsymbol{\zeta}}_{\text{init}} \right\| \leqslant \delta$$

*with probability at least $1 - c_1(Kn)^{-c_2}$. Here, $c_1$, $c_2$, $C > 0$ are some numerical constants.*

**Proof** By definition, we observe

$$
\begin{aligned}
\left\| \boldsymbol{\zeta}_{\text{init}} - \widehat{\boldsymbol{\zeta}}_{\text{init}} \right\| &= \left\| \boldsymbol{A}^\top \mathbb{P}_{\mathbb{S}^{n-1}} \left( \boldsymbol{P} \boldsymbol{y}_\ell \right) - \sqrt{K} \mathcal{P}_{\mathbb{S}^{n-1}} \left( \boldsymbol{A}^\top \boldsymbol{A} \boldsymbol{x}_\ell \right) \right\| \\
&= \left\| \boldsymbol{A}^\top \mathbb{P}_{\mathbb{S}^{n-1}} \left( \left( \frac{1}{\theta K m p} \boldsymbol{Y} \boldsymbol{Y}^\top \right)^{-1/2} \boldsymbol{A}_0 \boldsymbol{x}_\ell \right) - \sqrt{K} \mathcal{P}_{\mathbb{S}^{n-1}} \left( \boldsymbol{A}^\top \boldsymbol{A} \boldsymbol{x}_\ell \right) \right\| \\
&= \left\| \frac{\boldsymbol{A}^\top \left( \frac{1}{\theta m p} \boldsymbol{Y} \boldsymbol{Y}^\top \right)^{-1/2} \boldsymbol{A}_0 \boldsymbol{x}_\ell}{\left\| \left( \frac{1}{\theta m p} \boldsymbol{Y} \boldsymbol{Y}^\top \right)^{-1/2} \boldsymbol{A}_0 \boldsymbol{x}_\ell \right\|} - \frac{\boldsymbol{A}^\top \boldsymbol{A} \boldsymbol{x}_\ell}{\| \boldsymbol{A} \boldsymbol{x}_\ell \|} \right\| \\
&\leqslant \frac{2 \| \boldsymbol{A} \|}{\| \boldsymbol{A} \boldsymbol{x}_\ell \|} \left\| \left( \frac{1}{\theta m p} \boldsymbol{Y} \boldsymbol{Y}^\top \right)^{-1/2} \boldsymbol{A}_0 \boldsymbol{x}_\ell - \left( \boldsymbol{A}_0 \boldsymbol{A}_0^\top \right)^{-1/2} \boldsymbol{A}_0 \boldsymbol{x}_\ell \right\| \\
&\leqslant 2 \sqrt{K} \frac{\| \boldsymbol{x}_\ell \|}{\| \boldsymbol{A} \boldsymbol{x}_\ell \|} \| \boldsymbol{A}_0 \| \left\| \left( \frac{1}{\theta m p} \boldsymbol{Y} \boldsymbol{Y}^\top \right)^{-1/2} - \left( \boldsymbol{A}_0 \boldsymbol{A}_0^\top \right)^{-1/2} \right\| \\
&= 2 \sqrt{K} \| \boldsymbol{A}_0 \| \left\| \left( \frac{1}{\theta m p} \boldsymbol{Y} \boldsymbol{Y}^\top \right)^{-1/2} - \left( \boldsymbol{A}_0 \boldsymbol{A}_0^\top \right)^{-1/2} \right\|,
\end{aligned}
$$

where for the first inequality we invoked Lemma A.12, and the last equality follows the fact that minimum singular value of $\boldsymbol{A}$ is unity. Next, by Lemma E.18, for some $\epsilon \in (0, 1)$, whenever

$$
p \geqslant C \theta^{-1} K^2 \frac{\kappa^4(\boldsymbol{A}_0)}{\sigma_{\min}^4(\boldsymbol{A}_0)} \epsilon^{-2} \log(m),
$$

we have

$$
\left\| \boldsymbol{\zeta}_{\text{init}} - \widehat{\boldsymbol{\zeta}}_{\text{init}} \right\| \leqslant 8 \sqrt{K} \| \boldsymbol{A}_0 \| \epsilon
$$

holding with probability at least $1 - c_1(m)^{-c_2}$. Here, $c_1$, , $c_2$, $C > 0$ are some numerical constants. Replace $\delta = 8 \sqrt{K} \| \boldsymbol{A}_0 \| \epsilon$, we obtain the desired result. ∎

**Lemma E.7** *Suppose the columns of $\boldsymbol{A}$ are $\mu$-incoherent and satisfies Assumption 3.1, and suppose $\boldsymbol{x}_\ell$ satisfies Assumption E.2. Let $\mathcal{I} = \operatorname{supp}(\boldsymbol{x}_\ell)$. For any $t \geqslant 0$, we have*

$$
\left\| \mathcal{P}_{\mathcal{I}^c} \left( \boldsymbol{A}^\top \boldsymbol{A} \boldsymbol{x}_\ell \right) \right\| \leqslant \left\| \operatorname{offdiag} \left( \boldsymbol{A}^\top \boldsymbol{A} \right) \boldsymbol{x}_\ell \right\| \leqslant t
$$

*holds with probability at least $1 - 4m \exp \left( - \min \left\{ \frac{t^2}{4K^2 \mu^2 \theta m^2}, \frac{t}{4K \mu m \sqrt{m}} \right\} \right)$.*

**Proof** Since we have

$$
\left\| \mathcal{P}_{\mathcal{I}^c} \left( \boldsymbol{A}^\top \boldsymbol{A} \boldsymbol{x}_\ell \right) \right\| \leqslant \left\| \operatorname{offdiag} \left( \boldsymbol{A}^\top \boldsymbol{A} \right) \boldsymbol{x}_\ell \right\|, \tag{E.8}
$$

we could bound $\left\| \mathcal{P}_{\mathcal{I}^c} \boldsymbol{A}^\top \boldsymbol{A} \boldsymbol{x}_\ell \right\|$ via controlling $\left\| \operatorname{offdiag} \left( \boldsymbol{A}^\top \boldsymbol{A} \right) \boldsymbol{x}_\ell \right\|$. Let

$$
\boldsymbol{M} = \operatorname{offdiag} \left( \boldsymbol{A}^\top \boldsymbol{A} \right) = \begin{bmatrix} \boldsymbol{m}_1 & \cdots & \boldsymbol{m}_m \end{bmatrix} \in \mathbb{R}^{m \times m}, \quad \text{and} \quad \boldsymbol{s} = \boldsymbol{M} \boldsymbol{x}_\ell = \sum_{k=1}^{m} \underbrace{\boldsymbol{m}_k x_{\ell k}}_{\boldsymbol{s}_k}.
$$

Thus, we can apply vector version Bernstein inequality. By Lemma A.3 and the fact that $\| \boldsymbol{m}_k \| \leqslant K \mu \sqrt{m}$,

$$
\mathbb{E} \left[ \boldsymbol{s}_k \right] = \boldsymbol{0}, \quad \mathbb{E} \left[ \| \boldsymbol{s}_k \|^p \right] = \theta \| \boldsymbol{m}_k \|^p \mathbb{E}_{g \sim \mathcal{N}(0,1)} \left[ |g|^p \right] \leqslant \frac{m!}{2} \theta \left( K \mu \sqrt{m} \right)^p.
$$

Therefore, by applying Lemma A.6, we obtain

$$
\begin{aligned}
\mathbb{P} \left( \left\| \operatorname{offdiag} \left( \boldsymbol{A}^\top \boldsymbol{A} \right) \boldsymbol{x}_\ell \right\| \geqslant t \right) &= \mathbb{P} \left( \left\| \sum_{k=1}^{m} \boldsymbol{s}_k - \mathbb{E} \left[ \boldsymbol{s} \right] \right\| \geqslant t \right) \\
&\leqslant 2(m+1) \exp \left( - \frac{t^2}{2 \mu^2 K^2 \theta m^2 + 2 K \mu m \sqrt{m} t} \right).
\end{aligned}
$$

Finally, Equation (E.8) gives the desired result. ∎

**Lemma E.8** *We have*

$$\left\| \mathrm{diag}\left( \boldsymbol{A}^{\top}\boldsymbol{A} \right) \boldsymbol{x}_{\ell} \right\|^2 \;\leqslant\; K^2\theta m \;+\; t \tag{E.9}$$

*with probability at least* $1 - \exp\left( -\frac{1}{8}\min\left\{ \frac{t^2}{K^4\theta m}, \frac{t}{K^2 m} \right\} \right)$.

**Proof** First, let

$$\boldsymbol{d} = \mathrm{diag}\left( \boldsymbol{A}^{\top}\boldsymbol{A} \right), \qquad s \;=\; \left\| \mathrm{diag}\left( \boldsymbol{A}^{\top}\boldsymbol{A} \right) \boldsymbol{x}_{\ell} \right\|^2 = \sum_{k=1}^{m} \underbrace{d_k^2 x_{\ell k}^2}_{s_k},$$

where by Lemma A.4, we have

$$\mathbb{E}\left[ |s_k|^p \right] \;\leqslant\; \theta K^{2p}\frac{p!2^p}{2}, \quad \mathbb{E}\left[ s \right] \;=\; \theta \left\| \mathrm{diag}\left( \boldsymbol{A}^{\top}\boldsymbol{A} \right) \right\|_F^2 \;<\; K^2\theta m.$$

Thus, by Bernstein inequality in Lemma A.5, we obtain

$$\mathbb{P}\left( \left\| \mathrm{diag}\left( \boldsymbol{A}^{\top}\boldsymbol{A} \right) \boldsymbol{x}_{\ell} \right\|^2 - K^2\theta m \;\geqslant\; t \right) \leqslant \exp\left( -\frac{t^2}{4K^4\theta m + 4K^2 mt} \right),$$

as desired. ∎

**Lemma E.9** *Suppose $\boldsymbol{x}_{\ell}$ satisfies Assumption E.2. Suppose $\boldsymbol{x}_{\ell}$ satisfies Assumption E.2. Let $\mathcal{I} = \mathrm{supp}\left( \boldsymbol{x}_{\ell} \right)$. Whenever $\theta$ satisfies*

$$c_1\frac{\log m}{m} \;\leqslant\; \theta \;\leqslant\; c_2\frac{\mu^{-2}}{m\log m}, \tag{E.10}$$

*we have*

$$\left\| \mathcal{P}_{\mathcal{I}}\left( \boldsymbol{A}^{\top}\boldsymbol{A}\boldsymbol{x}_{\ell} \right) \right\|^2 \;\geqslant\; \frac{1}{2}K^2\theta m \tag{E.11}$$

*with probability at least $1 - m^{-c}$. Here, $c, c_1, c_2 > 0$ are some numerical constants.*

**Proof** Notice that

$$\left\| \mathcal{P}_{\mathcal{I}}\left( \boldsymbol{A}^{\top}\boldsymbol{A}\boldsymbol{x}_{\ell} \right) \right\|^2$$
$$= \left\| \mathrm{diag}\left( \boldsymbol{A}^{\top}\boldsymbol{A} \right) \boldsymbol{x}_{\ell} + \mathcal{P}_{\mathcal{I}}\left( \mathrm{offdiag}\left( \boldsymbol{A}^{\top}\boldsymbol{A} \right) \boldsymbol{x}_{\ell} \right) \right\|^2$$
$$= \left\| \mathrm{diag}\left( \boldsymbol{A}^{\top}\boldsymbol{A} \right) \boldsymbol{x}_{\ell} \right\|^2 + \left\| \mathcal{P}_{\mathcal{I}}\left( \mathrm{offdiag}\left( \boldsymbol{A}^{\top}\boldsymbol{A} \right) \boldsymbol{x}_{\ell} \right) \right\|^2 + 2\left\langle \mathrm{diag}\left( \boldsymbol{A}^{\top}\boldsymbol{A} \right) \boldsymbol{x}_{\ell}, \mathcal{P}_{\mathcal{I}}\left( \mathrm{offdiag}\left( \boldsymbol{A}^{\top}\boldsymbol{A} \right) \boldsymbol{x}_{\ell} \right) \right\rangle$$
$$\geqslant \left\| \mathrm{diag}\left( \boldsymbol{A}^{\top}\boldsymbol{A} \right) \boldsymbol{x}_{\ell} \right\|^2 - 2\left\| \mathrm{diag}\left( \boldsymbol{A}^{\top}\boldsymbol{A} \right) \boldsymbol{x}_{\ell} \right\| \left\| \mathcal{P}_{\mathcal{I}}\left( \mathrm{offdiag}\left( \boldsymbol{A}^{\top}\boldsymbol{A} \right) \boldsymbol{x}_{\ell} \right) \right\|.$$

By Lemma A.9, Lemma E.7, and Lemma E.8, we have

$$\left\| \mathrm{diag}\left( \boldsymbol{A}^{\top}\boldsymbol{A} \right) \boldsymbol{x}_{\ell} \right\|^2 \;\leqslant\; K^2\theta m \;+\; C_1 K^2\sqrt{\theta m\log m}$$
$$\left\| \mathcal{P}_{\mathcal{I}}\left( \mathrm{offdiag}\left( \boldsymbol{A}^{\top}\boldsymbol{A} \right) \boldsymbol{x}_{\ell} \right) \right\| \;\leqslant\; C_2\theta K\mu m\sqrt{\log m}$$

holds with probability at least $1 - m^{-c_0}$. Thus, we obtain

$$\left\| \mathcal{P}_{\mathcal{I}}\left( \boldsymbol{A}^{\top}\boldsymbol{A}\boldsymbol{x}_{\ell} \right) \right\|^2 \;\geqslant\; K^2\theta m\left( 1 - C_1\sqrt{\frac{\log m}{\theta m}} - C_3\mu\sqrt{\theta m\log m} \right).$$

Finally, by using Equation (E.10), we have

$$\left\| \mathcal{P}_{\mathcal{I}}\left( \boldsymbol{A}^{\top}\boldsymbol{A}\boldsymbol{x}_{\ell} \right) \right\|^2 \;\geqslant\; \frac{1}{2}K^2\theta m$$

as desired. ∎

**Lemma E.10** *Suppose $\boldsymbol{x}_\ell$ satisfies Assumption E.2. Let $\mathcal{I} = \text{supp}(\boldsymbol{x}_\ell)$. Whenever $\theta \in \left( \frac{\log m}{m}, \frac{1}{2} \right)$, then we have*

$$\left\| \mathcal{P}_{\mathbb{S}^{m-1}} \left( \mathcal{P}_{\mathcal{I}} \left( \boldsymbol{A}^\top \boldsymbol{A} \boldsymbol{x}_\ell \right) \right) \right\|_4^4 \;\geqslant\; \frac{1}{2\theta m}$$

*with probability at least $1 - m^{-c}$.*

**Proof** By Lemma A.1, we know that for any $\boldsymbol{z}$,

$$\|\boldsymbol{z}\|_4^4 \;\geqslant\; \|\boldsymbol{z}\|_0^{-1} \|\boldsymbol{z}\|^4,$$

and the fact that $\left\| \mathcal{P}_{\mathbb{S}^{m-1}} \left( \mathcal{P}_{\mathcal{I}} \left( \boldsymbol{A}^\top \boldsymbol{A} \boldsymbol{x}_\ell \right) \right) \right\|_0 = \|\boldsymbol{x}_\ell\|_0$, we have

$$\left\| \mathcal{P}_{\mathbb{S}^{m-1}} \left( \mathcal{P}_{\mathcal{I}} \left( \boldsymbol{A}^\top \boldsymbol{A} \boldsymbol{x}_\ell \right) \right) \right\|_4^4 \;\geqslant\; \|\boldsymbol{x}_\ell\|_0^{-1}.$$

By Lemma A.9, we have

$$\|\boldsymbol{x}_\ell\|_0 \;\leqslant\; 2\theta m \quad \Longrightarrow \quad \left\| \mathcal{P}_{\mathbb{S}^{m-1}} \left( \mathcal{P}_{\mathcal{I}} \left( \boldsymbol{A}^\top \boldsymbol{A} \boldsymbol{x}_\ell \right) \right) \right\|_4^4 \;\geqslant\; \frac{1}{2\theta m}$$

holds with probability at least $1 - m^{-c}$. ∎

### E.4 Concentration and Perturbation

We prove the following concentration results for Riemannian gradient and Hessian, and its function value.

**Proposition E.11** *For some small $\delta \in (0, 1)$, whenever the sample complexity satisfies*

$$p \;\geqslant\; C\delta^{-2}\theta K^4 \max \left\{ \frac{K^6 \kappa^6(\boldsymbol{A}_0)}{\sigma_{\min}^2(\boldsymbol{A}_0)}, \; n \right\} n^2 \log \left( \frac{\theta K n}{\delta} \right) \log^5(mK),$$

*we have*

$$\sup_{\boldsymbol{q} \in \mathbb{S}^{n-1}} \left\| \text{grad}\, \varphi_{\text{CDL}}(\boldsymbol{q}) - \text{grad}\, \varphi_{\text{T}}(\boldsymbol{q}) \right\| \;\leqslant\; \delta$$

$$\sup_{\boldsymbol{q} \in \mathbb{S}^{n-1}} \left\| \text{Hess}\, \varphi_{\text{CDL}}(\boldsymbol{q}) - \text{Hess}\, \varphi_{\text{T}}(\boldsymbol{q}) \right\| \;\leqslant\; \delta$$

*hold with probability at least $1 - c_1(mK)^{-c_2}$. Here, $c_1$, $c_2$, $C > 0$ are some numerical constants.*

**Proof** Let $\widehat{\varphi}_{\text{CDL}}(\boldsymbol{q})$ be introduced as Equation (E.12)

$$\widehat{\varphi}_{\text{CDL}}(\boldsymbol{q}) \;=\; -\frac{1}{12\theta(1-\theta)np} \left\| \boldsymbol{q}^\top \boldsymbol{A} \boldsymbol{X} \right\|_4^4,$$

so that we bound the Riemannian gradient and Hessian separately using triangle inequalities via $\widehat{\varphi}_{\text{CDL}}(\boldsymbol{q})$.

**Riemannian gradient.** Notice that

$$\sup_{\boldsymbol{q} \in \mathbb{S}^{n-1}} \left\| \text{grad}\, \varphi_{\text{CDL}}(\boldsymbol{q}) - \text{grad}\, \varphi_{\text{T}}(\boldsymbol{q}) \right\|$$

$$\leqslant \sup_{\boldsymbol{q} \in \mathbb{S}^{n-1}} \left\| \text{grad}\, \varphi_{\text{CDL}}(\boldsymbol{q}) - \text{grad}\, \widehat{\varphi}_{\text{CDL}}(\boldsymbol{q}) \right\| \;+\; \sup_{\boldsymbol{q} \in \mathbb{S}^{n-1}} \left\| \text{grad}\, \widehat{\varphi}_{\text{CDL}}(\boldsymbol{q}) - \text{grad}\, \varphi_{\text{T}}(\boldsymbol{q}) \right\|.$$

From Proposition E.13, we know that whenever

$$p \;\geqslant\; C_1 \theta K^{10} \frac{\kappa^6(\boldsymbol{A}_0)}{\sigma_{\min}^2(\boldsymbol{A}_0)} \delta^{-2} n^2 \log^5(mK),$$

we have

$$\sup_{\boldsymbol{q} \in \mathbb{S}^{n-1}} \left\| \text{grad}\, \varphi_{\text{CDL}}(\boldsymbol{q}) - \text{grad}\, \widehat{\varphi}_{\text{CDL}}(\boldsymbol{q}) \right\| \;\leqslant\; \frac{\delta}{2}$$

with probability at least $1 - c_1(mK)^{-c_2}$. On the other hand, Corollary F.9 implies that whenever

$$p \geqslant C_2 \delta^{-2} \theta K^5 n^2 \log\left(\frac{\theta K n}{\delta}\right),$$

we have

$$\sup_{\boldsymbol{q} \in \mathbb{S}^{n-1}} \|\operatorname{grad} \widehat{\varphi}_{\mathrm{CDL}}(\boldsymbol{q}) - \operatorname{grad} \varphi_{\mathrm{T}}(\boldsymbol{q})\| \leqslant \frac{\delta}{2}$$

holds with probability at least $1 - c_3 n p^{-2}$. Combining the bounds above gives the desired result on the gradient.

**Riemannian Hessian.** Similarly, we have

$$\sup_{\boldsymbol{q} \in \mathbb{S}^{n-1}} \|\operatorname{Hess} \varphi_{\mathrm{CDL}}(\boldsymbol{q}) - \operatorname{Hess} \varphi_{\mathrm{T}}(\boldsymbol{q})\|$$

$$\leqslant \sup_{\boldsymbol{q} \in \mathbb{S}^{n-1}} \|\operatorname{Hess} \varphi_{\mathrm{CDL}}(\boldsymbol{q}) - \operatorname{Hess} \widehat{\varphi}_{\mathrm{CDL}}(\boldsymbol{q})\| + \sup_{\boldsymbol{q} \in \mathbb{S}^{n-1}} \|\operatorname{Hess} \widehat{\varphi}_{\mathrm{CDL}}(\boldsymbol{q}) - \operatorname{Hess} \varphi_{\mathrm{T}}(\boldsymbol{q})\|.$$

From Proposition E.15, we know that whenever

$$p \geqslant C_3 \theta K^{10} \frac{\kappa^6(\boldsymbol{A}_0)}{\sigma_{\min}^2(\boldsymbol{A}_0)} \delta^{-2} n^2 \log^5(mK),$$

we have

$$\sup_{\boldsymbol{q} \in \mathbb{S}^{n-1}} \|\operatorname{Hess} \varphi_{\mathrm{CDL}}(\boldsymbol{q}) - \operatorname{Hess} \widehat{\varphi}_{\mathrm{CDL}}(\boldsymbol{q})\| \leqslant \frac{\delta}{2}$$

with probability at least $1 - c_4(mK)^{-c_5}$. On the other hand, Corollary F.10 implies that whenever

$$p \geqslant C_4 \theta K^6 \delta^{-2} n^3 \log\left(\theta K n / \delta\right),$$

we have

$$\sup_{\boldsymbol{q} \in \mathbb{S}^{n-1}} \|\operatorname{Hess} \varphi_{\mathrm{DL}}(\boldsymbol{q}) - \operatorname{Hess} \varphi_{\mathrm{T}}(\boldsymbol{q})\| < \frac{\delta}{2}$$

holds with probability at least $1 - c_4 n p^{-2}$. Combining the bounds above gives the desired result on the Hessian. ∎

Similar to Lemma D.1, for convolutional dictionary learning, asymptotically we have

$$\mathbb{E}_{\boldsymbol{X}}\left[\varphi_{\mathrm{CDL}}(\boldsymbol{q})\right] \approx \mathbb{E}_{\boldsymbol{X}}\left[\widehat{\varphi}_{\mathrm{CDL}}(\boldsymbol{q})\right] = \varphi_{\mathrm{T}}(\boldsymbol{q}) - \frac{\theta}{2(1-\theta)} K^2, \qquad \varphi_{\mathrm{T}}(\boldsymbol{q}) = -\frac{1}{4} \left\|\boldsymbol{q}^\top \boldsymbol{A}\right\|_4^4.$$

Next, we turn this asymptotical results into finite sample for the function value via concentration and preconditioning.

**Proposition E.12** *For some small $\delta \in (0, 1)$, whenever the sample complexity satisfies*

$$p \geqslant C \delta^{-2} \theta K^4 \max\left\{\frac{K^6 \kappa^6(\boldsymbol{A}_0)}{\sigma_{\min}^2(\boldsymbol{A}_0)}, \ n\right\} n^2 \log\left(\frac{\theta K n}{\delta}\right) \log^5(mK),$$

*we have*

$$\sup_{\boldsymbol{q} \in \mathbb{S}^{n-1}} \left\|\varphi_{\mathrm{CDL}}(\boldsymbol{q}) - \left(\varphi_{\mathrm{T}}(\boldsymbol{q}) - \frac{\theta}{2(1-\theta)} K^2\right)\right\| \leqslant \delta$$

*hold with probability at least $1 - c_1(mK)^{-c_2}$. Here, $c_1$, $c_2$, $C > 0$ are some numerical constants.*

**Proof** By triangle inequality, we have

$$\sup_{\boldsymbol{q} \in \mathbb{S}^{n-1}} \left|\varphi_{\mathrm{CDL}}(\boldsymbol{q}) - \left(\varphi_{\mathrm{T}}(\boldsymbol{q}) - \frac{\theta}{2(1-\theta)} K^2\right)\right|$$

$$\leqslant \underbrace{\sup_{\boldsymbol{q} \in \mathbb{S}^{n-1}} \left|\varphi_{\mathrm{CDL}}(\boldsymbol{q}) - \widehat{\varphi}_{\mathrm{CDL}}(\boldsymbol{q})\right|}_{\mathcal{T}_1} + \underbrace{\sup_{\boldsymbol{q} \in \mathbb{S}^{n-1}} \left|\widehat{\varphi}_{\mathrm{CDL}}(\boldsymbol{q}) - \mathbb{E}_{\boldsymbol{X}}\left[\widehat{\varphi}_{\mathrm{CDL}}(\boldsymbol{q})\right]\right|}_{\mathcal{T}_2}.$$

Thus, by using Corollary E.14 we can control $\mathcal{T}_1$. For $\mathcal{T}_2$, we can control in a similar way as Corollary F.9 or Corollary F.10. For simplicity, we omitted here. ∎

### E.5 PRECONDITIONING

In this part of appendix, let us introduce

$$\varphi_{\mathrm{CDL}}(\boldsymbol{q}) \;=\; -\frac{1}{12\theta(1-\theta)np} \left\| \boldsymbol{q}^\top (\boldsymbol{P}\boldsymbol{A}_0)\boldsymbol{X} \right\|, \; \widehat{\varphi}_{\mathrm{CDL}}(\boldsymbol{q}) \;:=\; -\frac{1}{12\theta(1-\theta)np} \left\| \boldsymbol{q}^\top \boldsymbol{A}\boldsymbol{X} \right\|. \tag{E.12}$$

In the following, we show that the differences of function value, Riemannian gradient, and Hessian of those two functions are small by preconditioning analysis. For simplicity, let us also introduce

$$\boldsymbol{v}_0(\boldsymbol{q}) \;=\; \boldsymbol{X}^\top (\boldsymbol{P}\boldsymbol{A}_0)^\top \boldsymbol{q}, \qquad \boldsymbol{v}(\boldsymbol{q}) \;=\; \boldsymbol{X}^\top \boldsymbol{A}^\top \boldsymbol{q}. \tag{E.13}$$

#### E.5.1 CONCENTRATION AND PRECONDITIONING FOR RIEMANNIAN GRADIENT AND FUNCTION VALUE

First, the gradients of $\varphi_{\mathrm{CDL}}(\boldsymbol{q})$ and $\widehat{\varphi}_{\mathrm{CDL}}(\boldsymbol{q})$ and their Riemannian variants can be written as

$$\nabla \varphi_{\mathrm{CDL}}(\boldsymbol{q}) \;=\; -\frac{1}{3\theta(1-\theta)np} \boldsymbol{P}\boldsymbol{A}_0 \boldsymbol{X} \boldsymbol{v}_0^{\odot 3}, \qquad \nabla \widehat{\varphi}_{\mathrm{CDL}}(\boldsymbol{q}) \;=\; -\frac{1}{3\theta(1-\theta)np} \boldsymbol{A}\boldsymbol{X} \boldsymbol{v}^{\odot 3},$$

$$\mathrm{grad}\, \varphi_{\mathrm{CDL}}(\boldsymbol{q}) \;=\; \boldsymbol{P}_{\boldsymbol{q}^\perp} \nabla \varphi_{\mathrm{CDL}}(\boldsymbol{q}), \qquad \mathrm{grad}\, \widehat{\varphi}_{\mathrm{CDL}}(\boldsymbol{q}) \;=\; \boldsymbol{P}_{\boldsymbol{q}^\perp} \nabla \widehat{\varphi}_{\mathrm{CDL}}(\boldsymbol{q}),$$

where recall from Section 3 that we introduced the following preconditioning matrix

$$\boldsymbol{P} \;=\; \left( \frac{1}{\theta K m p} \boldsymbol{Y}\boldsymbol{Y}^\top \right)^{-1/2} \;=\; \left[ \boldsymbol{A}_0 \left( \frac{1}{\theta K m p} \sum_{i=1}^p \boldsymbol{X}_i \boldsymbol{X}_i^\top \right) \boldsymbol{A}_0^\top \right]^{-1/2}.$$

In the following, we show that the difference between $\mathrm{grad}\, \varphi_{\mathrm{CDL}}(\boldsymbol{q})$ and $\mathrm{grad}\, \widehat{\varphi}_{\mathrm{CDL}}(\boldsymbol{q})$ is small.

**Proposition E.13** *Suppose* $\theta \in \left( \frac{1}{m}, \frac{1}{2} \right)$. *For any* $\delta \in (0,1)$, *whenever*

$$p \;\geqslant\; C\theta K^{10} \frac{\kappa^6(\boldsymbol{A}_0)}{\sigma_{\min}^2(\boldsymbol{A}_0)} \delta^{-2} n^2 \log^5(mK),$$

*we have*

$$\sup_{\boldsymbol{q} \in \mathbb{S}^{n-1}} \left\| \mathrm{grad}\, \varphi_{\mathrm{CDL}}(\boldsymbol{q}) - \mathrm{grad}\, \widehat{\varphi}_{\mathrm{CDL}}(\boldsymbol{q}) \right\| \;\leqslant\; \delta$$

$$\sup_{\boldsymbol{q} \in \mathbb{S}^{n-1}} \left\| \nabla \varphi_{\mathrm{CDL}}(\boldsymbol{q}) - \nabla \widehat{\varphi}_{\mathrm{CDL}}(\boldsymbol{q}) \right\| \;\leqslant\; \delta$$

*with probability at least* $1 - c_1(mK)^{-c_2}$. *Here,* $c_1, \; c_2, \; C > 0$ *are some numerical constants.*

**Proof** Notice that we have

$$\sup_{\boldsymbol{q} \in \mathbb{S}^{n-1}} \left\| \mathrm{grad}\, \varphi_{\mathrm{CDL}}(\boldsymbol{q}) - \mathrm{grad}\, \widehat{\varphi}_{\mathrm{CDL}}(\boldsymbol{q}) \right\|$$

$$\leqslant\; \sup_{\boldsymbol{q} \in \mathbb{S}^{n-1}} \left\| \nabla \varphi_{\mathrm{CDL}}(\boldsymbol{q}) - \nabla \widehat{\varphi}_{\mathrm{CDL}}(\boldsymbol{q}) \right\|$$

$$\leqslant\; \frac{1}{3\theta(1-\theta)np} \sup_{\boldsymbol{q} \in \mathbb{S}^{n-1}} \left\| \boldsymbol{P}\boldsymbol{A}_0 \boldsymbol{X} \boldsymbol{v}_0^{\odot 3} - \boldsymbol{A}\boldsymbol{X} \boldsymbol{v}^{\odot 3} \right\|$$

$$\leqslant\; \frac{1}{3\theta(1-\theta)np} \left( \underbrace{\sup_{\boldsymbol{q} \in \mathbb{S}^{n-1}} \left\| \boldsymbol{P}\boldsymbol{A}_0 \boldsymbol{X} \left[ \boldsymbol{v}_0^{\odot 3} - \boldsymbol{v}^{\odot 3} \right] \right\|}_{\mathcal{T}_1} + \underbrace{\sup_{\boldsymbol{q} \in \mathbb{S}^{n-1}} \left\| \left( \boldsymbol{P}\boldsymbol{A}_0 - \boldsymbol{A} \right) \boldsymbol{X} \boldsymbol{v}^{\odot 3} \right\|}_{\mathcal{T}_2} \right).$$

**Controlling $\mathcal{T}_1$.** For the first term, we observe

$$\mathcal{T}_1 \;\leqslant\; \frac{1}{3\theta(1-\theta)np} \left\| \boldsymbol{P}\boldsymbol{A}_0 \right\| \left\| \boldsymbol{X} \right\| \sup_{\boldsymbol{q} \in \mathbb{S}^{n-1}} \left\| \boldsymbol{v}_0^{\odot 3} - \boldsymbol{v}^{\odot 3} \right\|,$$

where for all $\boldsymbol{q} \in \mathbb{S}^{n-1}$ we have

$$
\begin{aligned}
\left\| \boldsymbol{v}_0^{\odot 3} - \boldsymbol{v}^{\odot 3} \right\| \;\leqslant\; & \left\| \boldsymbol{v}^{\odot 2} - \boldsymbol{v}_0^{\odot 2} \right\|_\infty \left\| \boldsymbol{v} \right\| \;+\; \left\| \boldsymbol{v} - \boldsymbol{v}_0 \right\| \left\| \boldsymbol{v}_0 \right\|_\infty^2 \\
\leqslant\; & \sqrt{K} \left( \sqrt{K} + \left\| \boldsymbol{P}\boldsymbol{A}_0 \right\| \right) \left\| \boldsymbol{P}\boldsymbol{A}_0 - \boldsymbol{A} \right\| \left( \max_{1 \leqslant k \leqslant np} \left\| \boldsymbol{X}\boldsymbol{e}_k \right\| \right)^2 \left\| \boldsymbol{X} \right\| \\
& +\; \left\| \boldsymbol{P}\boldsymbol{A}_0 - \boldsymbol{A} \right\| \left\| \boldsymbol{X} \right\| \left\| \boldsymbol{P}\boldsymbol{A}_0 \right\|^2 \left( \max_{1 \leqslant k \leqslant np} \left\| \boldsymbol{X}\boldsymbol{e}_k \right\| \right)^2 \\
\leqslant\; & \left( \sqrt{K} + \left\| \boldsymbol{P}\boldsymbol{A}_0 \right\| \right)^2 \left\| \boldsymbol{X} \right\| \left( \max_{1 \leqslant k \leqslant np} \left\| \boldsymbol{X}\boldsymbol{e}_k \right\| \right)^2 \left\| \boldsymbol{P}\boldsymbol{A}_0 - \boldsymbol{A} \right\|
\end{aligned}
$$

where for the last two inequalities we used Lemma E.16. Thus, we have

$$
\mathcal{T}_1 \;\leqslant\; \left( \sqrt{K} + \left\| \boldsymbol{P}\boldsymbol{A}_0 \right\| \right)^2 \left\| \boldsymbol{P}\boldsymbol{A}_0 \right\| \left\| \boldsymbol{X} \right\|^2 \left( \max_{1 \leqslant k \leqslant np} \left\| \boldsymbol{X}\boldsymbol{e}_k \right\| \right)^2 \left\| \boldsymbol{P}\boldsymbol{A}_0 - \boldsymbol{A} \right\|.
$$

**Controlling $\mathcal{T}_2$.** For the second term, by Lemma E.16, we have

$$
\mathcal{T}_2 \;\leqslant\; \left\| \boldsymbol{P}\boldsymbol{A}_0 - \boldsymbol{A} \right\| \left\| \boldsymbol{X} \right\| \left\| \boldsymbol{v} \right\|_6^3 \;\leqslant\; K^{3/2} \left\| \boldsymbol{X} \right\|^2 \left( \max_{1 \leqslant k \leqslant np} \left\| \boldsymbol{X}\boldsymbol{e}_k \right\| \right)^2 \left\| \boldsymbol{P}\boldsymbol{A}_0 - \boldsymbol{A} \right\|.
$$

**Summary.** Putting all the bounds together, we have

$$
\sup_{\boldsymbol{q} \in \mathbb{S}^{n-1}} \left\| \operatorname{grad} \varphi_{\mathrm{CDL}}(\boldsymbol{q}) - \operatorname{grad} \widehat{\varphi}_{\mathrm{CDL}}(\boldsymbol{q}) \right\|
$$

$$
\leqslant \frac{1}{3\theta(1-\theta)np} \left[ \left( \sqrt{K} + \left\| \boldsymbol{P}\boldsymbol{A}_0 \right\| \right)^2 \left\| \boldsymbol{P}\boldsymbol{A}_0 \right\| + K^{3/2} \right] \left\| \boldsymbol{X} \right\|^2 \left( \max_{1 \leqslant k \leqslant np} \left\| \boldsymbol{X}\boldsymbol{e}_k \right\| \right)^2 \left\| \boldsymbol{P}\boldsymbol{A}_0 - \boldsymbol{A} \right\|.
$$

By Lemma E.17 and Lemma E.20, we have

$$
\left\| \boldsymbol{X} \right\| \;\leqslant\; 2\sqrt{\theta m p}, \qquad \max_{1 \leqslant k \leqslant np} \left\| \boldsymbol{X}\boldsymbol{e}_k \right\| \;\leqslant\; 4\sqrt{\theta m} \log(Kp)
$$

with probably at least $1 - 2p^{-2}$. On the other hand, by Lemma E.19, there exists some constant $C > 0$, for any $\epsilon \in (0, 1)$ whenever

$$
p \;\geqslant\; C\theta^{-1} K^3 \frac{\kappa^6(\boldsymbol{A}_0)}{\sigma_{\min}^2(\boldsymbol{A}_0)} \epsilon^{-2} \log(mK),
$$

we have

$$
\left\| \boldsymbol{P}\boldsymbol{A}_0 - \boldsymbol{A} \right\| \;\leqslant\; \epsilon, \qquad \left\| \boldsymbol{P}\boldsymbol{A}_0 \right\| \;\leqslant\; 2\sqrt{K}
$$

hold with probability at least $1 - c_1(mK)^{-c_2}$ for some numerical constants $c_1, c_2 > 0$. These together give

$$
\mathcal{T}_1 \;\leqslant\; CK^{5/2} \theta m \log^2(Km)\epsilon.
$$

Replacing $\delta = CK^{5/2} \theta m \log^2(Km) \epsilon$ gives the desired result. ∎

Here, the perturbation analysis for gradient also leads to the following result

**Corollary E.14** *For some small $\delta \in (0, 1)$, under the same setting of Proposition E.13, we have*

$$
\sup_{\boldsymbol{q} \in \mathbb{S}^{n-1}} \left| \varphi_{\mathrm{CDL}}(\boldsymbol{q}) - \widehat{\varphi}_{\mathrm{CDL}}(\boldsymbol{q}) \right| \;\leqslant\; \delta
$$

*hold with probability at least $1 - c_1(mK)^{-c_2}$. Here, $c_1$, $c_2 > 0$ are some numerical constants.*

**Proof** Under the same setting of Proposition E.13, we have

$$
\begin{aligned}
\sup_{\boldsymbol{q} \in \mathbb{S}^{n-1}} \left| \varphi_{\mathrm{CDL}}(\boldsymbol{q}) - \widehat{\varphi}_{\mathrm{CDL}}(\boldsymbol{q}) \right| \;=\; & \sup_{\boldsymbol{q} \in \mathbb{S}^{n-1}} \frac{1}{4} \left| \frac{1}{3\theta(1-\theta)np} \left\| \boldsymbol{v}_0 \right\|_4^4 - \frac{1}{3\theta(1-\theta)np} \left\| \boldsymbol{v} \right\|_4^4 \right| \\
=\; & \sup_{\boldsymbol{q} \in \mathbb{S}^{n-1}} \frac{1}{4} \left| \frac{1}{3\theta(1-\theta)np} \left\langle \boldsymbol{q}, \boldsymbol{P}\boldsymbol{A}_0 \boldsymbol{X} \boldsymbol{v}_0^{\odot 3} - \boldsymbol{A}\boldsymbol{X}\boldsymbol{v}^{\odot 3} \right\rangle \right| \\
\leqslant\; & \frac{1}{4} \sup_{\boldsymbol{q} \in \mathbb{S}^{n-1}} \left\| \nabla \varphi_{\mathrm{CDL}}(\boldsymbol{q}) - \widehat{\varphi}_{\mathrm{CDL}}(\boldsymbol{q}) \right\| \;\leqslant\; \frac{\delta}{4},
\end{aligned}
$$

as desired. ∎

### E.5.2 Concentration and Preconditioning for Riemannian Hessian

For simplicity, let $\boldsymbol{v}_0$ and $\boldsymbol{v}$ be as introduced in [Equation (E.13)]. Similarly, the Riemannian Hessian of $\varphi_{\mathrm{CDL}}(\boldsymbol{q})$ and $\widehat{\varphi}_{\mathrm{CDL}}(\boldsymbol{q})$ can be written as

$$\mathrm{Hess}\,\varphi_{\mathrm{CDL}}(\boldsymbol{q}) = -\frac{1}{3\theta(1-\theta)np}\boldsymbol{P}_{\boldsymbol{q}^\perp}\left[3\left(\boldsymbol{P}\boldsymbol{A}_0\right)\boldsymbol{X}\,\mathrm{diag}\left(\boldsymbol{v}_0^{\odot 2}\right)\boldsymbol{X}^\top\left(\boldsymbol{P}\boldsymbol{A}_0\right)^\top - \|\boldsymbol{v}_0\|_4^4\,\boldsymbol{I}\right]\boldsymbol{P}_{\boldsymbol{q}^\perp},$$

$$\mathrm{Hess}\,\widehat{\varphi}_{\mathrm{CDL}}(\boldsymbol{q}) = -\frac{1}{3\theta(1-\theta)np}\boldsymbol{P}_{\boldsymbol{q}^\perp}\left[3\boldsymbol{A}\boldsymbol{X}\,\mathrm{diag}\left(\boldsymbol{v}^{\odot 2}\right)\boldsymbol{X}^\top\boldsymbol{A}^\top - \|\boldsymbol{v}\|_4^4\,\boldsymbol{I}\right]\boldsymbol{P}_{\boldsymbol{q}^\perp},$$

respectively. In the following, we show that the difference between $\mathrm{grad}\,\varphi_{\mathrm{CDL}}(\boldsymbol{q})$ and $\mathrm{grad}\,\widehat{\varphi}_{\mathrm{CDL}}(\boldsymbol{q})$ is small.

**Proposition E.15** *Suppose $\theta \in \left(\frac{1}{m},\frac{1}{2}\right)$. For any $\delta \in (0,1)$, whenever*

$$p \;\geqslant\; C\theta K^{10}\frac{\kappa^6(\boldsymbol{A}_0)}{\sigma_{\min}^2(\boldsymbol{A}_0)}\delta^{-2}n^2\log^5(mK),$$

*we have*

$$\sup_{\boldsymbol{q}\in\mathbb{S}^{n-1}}\|\mathrm{Hess}\,\varphi_{\mathrm{CDL}}(\boldsymbol{q}) - \mathrm{Hess}\,\widehat{\varphi}_{\mathrm{CDL}}(\boldsymbol{q})\| \;\leqslant\; \delta$$

*with probability at least $1 - c_1(mK)^{-c_2}$. Here, $c_1$, $c_2$, $C > 0$ are some numerical constants.*

**Proof** Notice that

$$\sup_{\boldsymbol{q}\in\mathbb{S}^{n-1}}\|\mathrm{Hess}\,\varphi_{\mathrm{CDL}}(\boldsymbol{q}) - \mathrm{Hess}\,\widehat{\varphi}_{\mathrm{CDL}}(\boldsymbol{q})\|$$

$$\leqslant \frac{1}{\theta(1-\theta)np}\underbrace{\sup_{\boldsymbol{q}\in\mathbb{S}^{n-1}}\left\|(\boldsymbol{P}\boldsymbol{A}_0-\boldsymbol{A})\boldsymbol{X}\,\mathrm{diag}\left(\boldsymbol{v}_0^{\odot 2}\right)\boldsymbol{X}^\top\left(\boldsymbol{P}\boldsymbol{A}_0\right)^\top\right\|}_{\mathcal{T}_1}$$

$$+ \frac{1}{\theta(1-\theta)np}\underbrace{\sup_{\boldsymbol{q}\in\mathbb{S}^{n-1}}\left\|\boldsymbol{A}\boldsymbol{X}\,\mathrm{diag}\left(\boldsymbol{v}^{\odot 2}\right)\boldsymbol{X}\left(\boldsymbol{P}\boldsymbol{A}_0-\boldsymbol{A}\right)^\top\right\|}_{\mathcal{T}_2}$$

$$+ \frac{1}{\theta(1-\theta)np}\underbrace{\sup_{\boldsymbol{q}\in\mathbb{S}^{n-1}}\left\|\boldsymbol{A}\boldsymbol{X}\,\mathrm{diag}\left(\boldsymbol{v}_0^{\odot 2}-\boldsymbol{v}^{\odot 2}\right)\boldsymbol{X}^\top\left(\boldsymbol{P}\boldsymbol{A}_0\right)^\top\right\|}_{\mathcal{T}_3}$$

$$+ \frac{1}{3\theta(1-\theta)np}\sup_{\boldsymbol{q}\in\mathbb{S}^{n-1}}\underbrace{\left|\|\boldsymbol{v}\|_4^4-\|\boldsymbol{v}_0\|_4^4\right|}_{\mathcal{T}_4}.$$

By using Lemma [E.16], we have

$$\mathcal{T}_1 \;\leqslant\; \|\boldsymbol{P}\boldsymbol{A}_0\|\,\|\boldsymbol{X}\|^2\,\|\boldsymbol{P}\boldsymbol{A}_0-\boldsymbol{A}\|\sup_{\boldsymbol{q}\in\mathbb{S}^{n-1}}\|\boldsymbol{v}_0\|_\infty^2 \;\leqslant\; \|\boldsymbol{P}\boldsymbol{A}_0\|^3\,\|\boldsymbol{X}\|^2\left(\max_{1\leqslant k\leqslant np}\|\boldsymbol{X}\boldsymbol{e}_k\|\right)^2\|\boldsymbol{P}\boldsymbol{A}_0-\boldsymbol{A}\|,$$

$$\mathcal{T}_2 \;\leqslant\; \|\boldsymbol{A}\|\,\|\boldsymbol{X}\|^2\sup_{\boldsymbol{q}\in\mathbb{S}^{n-1}}\|\boldsymbol{v}\|_\infty^2 \;\leqslant\; K^{3/2}\,\|\boldsymbol{X}\|^2\left(\max_{1\leqslant k\leqslant np}\|\boldsymbol{X}\boldsymbol{e}_k\|\right)^2\|\boldsymbol{P}\boldsymbol{A}_0-\boldsymbol{A}\|.$$

Similarly, Lemma [E.16] implies that

$$\mathcal{T}_3 \;\leqslant\; \|\boldsymbol{P}\boldsymbol{A}_0\|\,\|\boldsymbol{A}\|\,\|\boldsymbol{X}\|^2\sup_{\boldsymbol{q}\in\mathbb{S}^{n-1}}\left\|\boldsymbol{v}_0^{\odot 2}-\boldsymbol{v}^{\odot 2}\right\|_\infty$$

$$\leqslant\; \sqrt{K}\left(\sqrt{K}+\|\boldsymbol{P}\boldsymbol{A}_0\|\right)\|\boldsymbol{P}\boldsymbol{A}_0\|\,\|\boldsymbol{X}\|^2\left(\max_{1\leqslant k\leqslant np}\|\boldsymbol{X}\boldsymbol{e}_k\|\right)^2\|\boldsymbol{P}\boldsymbol{A}_0-\boldsymbol{A}\|,$$

and

$$
\begin{aligned}
\mathcal{T}_4 \;\leqslant\; \sup_{\boldsymbol{q}\in\mathbb{S}^{n-1}} \left| \|\boldsymbol{v}\|_4^4 - \|\boldsymbol{v}_0\|_4^4 \right| \;&\leqslant\; 2 \sup_{\boldsymbol{q}\in\mathbb{S}^{n-1}} \left| \left\langle \boldsymbol{v} - \boldsymbol{v}_0, 4\boldsymbol{v}^{\odot 3} \right\rangle \right| \\
&\leqslant\; 8 \sup_{\boldsymbol{q}\in\mathbb{S}^{n-1}} \|\boldsymbol{v} - \boldsymbol{v}_0\| \, \|\boldsymbol{v}\|_6^3 \\
&\leqslant\; 8K^{3/2} \|\boldsymbol{X}\|^2 \left( \max_{1\leqslant k\leqslant np} \|\boldsymbol{X}\boldsymbol{e}_k\| \right)^2 \|\boldsymbol{P}\boldsymbol{A}_0 - \boldsymbol{A}\| .
\end{aligned}
$$

Thus, combining all the results above, we obtain

$$
\sup_{\boldsymbol{q}\in\mathbb{S}^{n-1}} \| \mathrm{Hess}\, \varphi_{\mathrm{CDL}}(\boldsymbol{q}) - \mathrm{Hess}\, \widehat{\varphi}_{\mathrm{CDL}}(\boldsymbol{q}) \|
$$
$$
\leqslant \frac{1}{\theta(1-\theta)np} \left[ \left( \sqrt{K} + \|\boldsymbol{P}\boldsymbol{A}_0\| \right) \|\boldsymbol{P}\boldsymbol{A}_0\|^2 + K \|\boldsymbol{P}\boldsymbol{A}_0\| + 4K^{3/2} \right] \|\boldsymbol{X}\|^2 \left( \max_{1\leqslant k\leqslant np} \|\boldsymbol{X}\boldsymbol{e}_k\| \right)^2 \|\boldsymbol{P}\boldsymbol{A}_0 - \boldsymbol{A}\| .
$$

By Lemma E.17 and Lemma E.20, we have

$$
\|\boldsymbol{X}\| \;\leqslant\; 2\sqrt{\theta m p}, \qquad \max_{1\leqslant k\leqslant np} \|\boldsymbol{X}\boldsymbol{e}_k\| \;\leqslant\; 4\sqrt{\theta m} \log(Kp)
$$

with probably at least $1 - 2p^{-2}$. On the other hand, by Lemma E.19, there exists some constant $C > 0$, for any $\epsilon \in (0,1)$ whenever

$$
p \;\geqslant\; C\theta^{-1} K^3 \frac{\kappa^6(\boldsymbol{A}_0)}{\sigma_{\min}^2(\boldsymbol{A}_0)} \epsilon^{-2} \log(mK),
$$

we have

$$
\|\boldsymbol{P}\boldsymbol{A}_0 - \boldsymbol{A}\| \;\leqslant\; \epsilon, \qquad \|\boldsymbol{P}\boldsymbol{A}_0\| \;\leqslant\; 2\sqrt{K}
$$

hold with probability at least $1 - c_1(mK)^{-c_2}$ for some numerical constants $c_1, c_2 > 0$. These together gives

$$
\sup_{\boldsymbol{q}\in\mathbb{S}^{n-1}} \| \mathrm{Hess}\, \varphi_{\mathrm{CDL}}(\boldsymbol{q}) - \mathrm{Hess}\, \widehat{\varphi}_{\mathrm{CDL}}(\boldsymbol{q}) \| \;\leqslant\; C' K^{5/2} \theta m \log^2(Kp)\, \epsilon .
$$

Replacing $\delta = C' K^{5/2} \theta m \log^2(Kp)\, \epsilon$ gives the desired result. ∎

### E.5.3 Auxiliary Results

**Lemma E.16** *Let $\boldsymbol{v}_0$ and $\boldsymbol{v}$ be defined as in [Equation (E.13)](#), with*

$$
\boldsymbol{v}_0(\boldsymbol{q}) \;=\; \boldsymbol{X}^\top \left( \boldsymbol{P}\boldsymbol{A}_0 \right)^\top \boldsymbol{q}, \qquad \boldsymbol{v}(\boldsymbol{q}) \;=\; \boldsymbol{X}^\top \boldsymbol{A}^\top \boldsymbol{q},
$$

*For all $\boldsymbol{q} \in \mathbb{S}^{n-1}$, we have*

$$
\|\boldsymbol{v}\|_\infty \;\leqslant\; \sqrt{K} \max_{1\leqslant k\leqslant np} \|\boldsymbol{X}\boldsymbol{e}_k\|, \qquad \|\boldsymbol{v}_0\|_\infty \;\leqslant\; \|\boldsymbol{P}\boldsymbol{A}_0\| \max_{1\leqslant k\leqslant np} \|\boldsymbol{X}\boldsymbol{e}_k\|,
$$

$$
\|\boldsymbol{v}\| \;\leqslant\; \sqrt{K} \|\boldsymbol{X}\|, \qquad \|\boldsymbol{v}\|_6^6 \;\leqslant\; K^3 \|\boldsymbol{X}\|^2 \left( \max_{1\leqslant k\leqslant np} \|\boldsymbol{X}\boldsymbol{e}_k\| \right)^4,
$$

$$
\left\| \boldsymbol{v}^{\odot 2} - \boldsymbol{v}_0^{\odot 2} \right\|_\infty \;\leqslant\; \left( \sqrt{K} + \|\boldsymbol{P}\boldsymbol{A}_0\| \right) \|\boldsymbol{P}\boldsymbol{A}_0 - \boldsymbol{A}\| \left( \max_{1\leqslant k\leqslant np} \|\boldsymbol{X}\boldsymbol{e}_k\| \right)^2,
$$

$$
\|\boldsymbol{v} - \boldsymbol{v}_0\| \;\leqslant\; \|\boldsymbol{P}\boldsymbol{A}_0 - \boldsymbol{A}\| \|\boldsymbol{X}\| .
$$

**Proof** In the following, we bound each term, respectively.

**Bounding norms of $v$ and $v_0$.** For the $\ell^2$-norm, notice that

$$\|\boldsymbol{v}\| \leqslant \|\boldsymbol{X}\| \|\boldsymbol{A}\| \leqslant \sqrt{K} \|\boldsymbol{X}\|$$

On the other hand, for the $\ell^\infty$-norm, we have

$$\|\boldsymbol{v}\|_\infty = \max_{1\leqslant k\leqslant np} \left\|\boldsymbol{e}_k^\top \boldsymbol{X}^\top \boldsymbol{A}^\top \boldsymbol{q}\right\| \leqslant \sqrt{K} \max_{1\leqslant k\leqslant np} \|\boldsymbol{X}\boldsymbol{e}_k\|$$

$$\|\boldsymbol{v}_0\|_\infty = \max_{1\leqslant k\leqslant np} \left\|\boldsymbol{e}_k^\top \boldsymbol{X}^\top \left(\boldsymbol{P}\boldsymbol{A}_0\right)^\top \boldsymbol{q}\right\| \leqslant \|\boldsymbol{P}\boldsymbol{A}_0\| \max_{1\leqslant k\leqslant np} \|\boldsymbol{X}\boldsymbol{e}_k\|.$$

Thus, the results above give

$$\|\boldsymbol{v}\|_6^6 \leqslant \|\boldsymbol{v}\|_\infty^4 \|\boldsymbol{v}\|^2 \leqslant K^3 \|\boldsymbol{X}\|^2 \left(\max_{1\leqslant k\leqslant np} \|\boldsymbol{X}\boldsymbol{e}_k\|\right)^4.$$

**Bounding the difference between $v$ and $v_0$.** First, we bound the difference in $\ell^2$-norm,

$$\|\boldsymbol{v} - \boldsymbol{v}_0\| = \left\|\boldsymbol{X}^\top \left(\boldsymbol{P}\boldsymbol{A}_0 - \boldsymbol{A}\right)^\top \boldsymbol{q}\right\| \leqslant \|\boldsymbol{P}\boldsymbol{A}_0 - \boldsymbol{A}\| \|\boldsymbol{X}\|.$$

On the other hand, we have

$$\left\|\boldsymbol{v}^{\odot 2} - \boldsymbol{v}_0^{\odot 2}\right\|_\infty \leqslant \|\boldsymbol{v} - \boldsymbol{v}_0\|_\infty \|\boldsymbol{v} + \boldsymbol{v}_0\|_\infty \leqslant (\|\boldsymbol{v}\|_\infty + \|\boldsymbol{v}_0\|_\infty) \|\boldsymbol{v} - \boldsymbol{v}_0\|_\infty,$$

where

$$\|\boldsymbol{v} - \boldsymbol{v}_0\|_\infty = \max_{1\leqslant k\leqslant np} \left\|\boldsymbol{e}_k^\top \boldsymbol{X}^\top \left(\boldsymbol{P}\boldsymbol{A}_0 - \boldsymbol{A}\right)^\top \boldsymbol{q}\right\| \leqslant \|\boldsymbol{P}\boldsymbol{A}_0 - \boldsymbol{A}\| \max_{1\leqslant k\leqslant np} \|\boldsymbol{X}\boldsymbol{e}_k\|,$$

Thus, we obtain

$$\left\|\boldsymbol{v}^{\odot 2} - \boldsymbol{v}_0^{\odot 2}\right\|_\infty \leqslant \left(\sqrt{K} + \|\boldsymbol{P}\boldsymbol{A}_0\|\right) \|\boldsymbol{P}\boldsymbol{A}_0 - \boldsymbol{A}\| \left(\max_{1\leqslant k\leqslant np} \|\boldsymbol{X}\boldsymbol{e}_k\|\right)^2,$$

as desired. ∎

**Lemma E.17** *Suppose $\boldsymbol{X}$ satisfies Assumption E.2, we have*

$$\max_{1\leqslant k\leqslant np} \|\boldsymbol{X}\boldsymbol{e}_k\| \leqslant 4\sqrt{\theta m} \log(Kp)$$

*with probability at least $1 - p^{-2\theta m}$.*

**Proof** Let us write

$$\boldsymbol{X}_i = \begin{bmatrix} \widetilde{\boldsymbol{x}}_{i1} & \widetilde{\boldsymbol{x}}_{i2} & \cdots & \widetilde{\boldsymbol{x}}_{in} \end{bmatrix}, \quad \text{with} \quad \widetilde{\boldsymbol{x}}_{ij} = \begin{bmatrix} \mathsf{s}_{j-1}\left[\boldsymbol{x}_{i1}\right] \\ \vdots \\ \mathsf{s}_{j-1}\left[\boldsymbol{x}_{iK}\right] \end{bmatrix} \quad 1 \leqslant i \leqslant p, \quad 1 \leqslant j \leqslant n,$$

where $\mathsf{s}_\ell\left[\cdot\right]$ denotes circulant shift of length $\ell$. Given $\boldsymbol{X} = \begin{bmatrix} \boldsymbol{X}_1 & \cdots & \boldsymbol{X}_p \end{bmatrix}$, we have

$$\max_{1\leqslant k\leqslant np} \|\boldsymbol{X}\boldsymbol{e}_k\| = \max_{1\leqslant i\leqslant p, 1\leqslant j\leqslant n} \|\widetilde{\boldsymbol{x}}_{ij}\| = \max_{1\leqslant i\leqslant p, 1\leqslant j\leqslant n} \sqrt{\sum_{\ell=1}^K \left\|\mathsf{s}_{j-1}\left[\boldsymbol{x}_{i\ell}\right]\right\|^2}$$

$$\leqslant \sqrt{K} \max_{1\leqslant i\leqslant p, 1\leqslant \ell\leqslant K} \|\boldsymbol{x}_{i\ell}\|.$$

Next, we bound $\max_{1\leqslant i\leqslant p, 1\leqslant \ell\leqslant K} \|\boldsymbol{x}_{i\ell}\|$. By using Bernstein inequality in Lemma A.5, we obtain

$$\mathbb{P}\left(\left|\|\boldsymbol{x}_{i\ell}\|^2 - n\theta\right| \geqslant t\right) \leqslant 2\exp\left(-\frac{t^2}{4n\theta + 4t}\right)$$

Thus, by using a union bound, we obtain

$$\max_{1\leqslant i\leqslant p, 1\leqslant \ell\leqslant K} \|\boldsymbol{x}_{i\ell}\| \leqslant 4\sqrt{\theta n} \log(Kp),$$

with probability at least $1 - p^{-2\theta m}$. Summarizing the bounds above, we obtain the desired result. ∎

### E.6 INTERMEDIATE RESULTS FOR PRECONDITIONING

**Lemma E.18** *Suppose $\boldsymbol{X}$ satisfies Assumption E.2. For any $\delta \in (0, 1)$, whenever*

$$p \;\geqslant\; C\theta^{-1}K^2 \frac{\kappa^4(\boldsymbol{A}_0)}{\sigma_{\min}^4(\boldsymbol{A}_0)}\delta^{-2}\log(m),$$

*we have*

$$\left\|\left(\frac{1}{\theta mp}\boldsymbol{Y}\boldsymbol{Y}^\top\right)^{-1/2} - \left(\boldsymbol{A}_0\boldsymbol{A}_0^\top\right)^{-1/2}\right\| \;\leqslant\; \delta,$$

$$\left\|\left(\frac{1}{\theta mp}\boldsymbol{Y}\boldsymbol{Y}^\top\right)^{1/2}\left(\boldsymbol{A}_0\boldsymbol{A}_0^\top\right)^{-1/2} - \boldsymbol{I}\right\| \;\leqslant\; \sigma_{\min}(\boldsymbol{A}_0)\cdot\delta,$$

*hold with probability at least $1 - c_1(mK)^{-c_2}$. Here, $c_1,\ c_2,\ C > 0$ are some numerical constants.*

**Proof** Notice that

$$\frac{1}{\theta mp}\boldsymbol{Y}\boldsymbol{Y}^\top \;=\; \frac{1}{\theta mp}\boldsymbol{A}_0\boldsymbol{X}\boldsymbol{X}^\top\boldsymbol{A}_0^\top \;=\; \underbrace{\boldsymbol{A}_0\boldsymbol{A}_0^\top}_{\boldsymbol{B}} + \underbrace{\boldsymbol{A}_0\left(\frac{1}{\theta mp}\boldsymbol{X}\boldsymbol{X}^\top - \boldsymbol{I}\right)\boldsymbol{A}_0^\top}_{\boldsymbol{\Delta}}.$$

By Lemma E.20, for any $\epsilon \in (0, 1/K)$, whenever

$$p \;\geqslant\; C\theta^{-1}K^2\epsilon^{-2}\log(mK),$$

we have

$$\left\|\frac{1}{\theta mp}\boldsymbol{X}\boldsymbol{X}^\top - \boldsymbol{I}\right\| \;\leqslant\; \epsilon,$$

with probability at least $1 - c_1(mK)^{-c_2}$. Thus, by the first inequality in Lemma A.10 we observe

$$\left\|\left(\frac{1}{\theta mp}\boldsymbol{Y}\boldsymbol{Y}^\top\right)^{-1/2} - (\boldsymbol{A}_0\boldsymbol{A}_0)^{-1/2}\right\| \;=\; \left\|(\boldsymbol{B}+\boldsymbol{\Delta})^{-1/2} - \boldsymbol{B}^{-1/2}\right\|$$

$$\leqslant\; 4\sigma_{\min}^{-2}(\boldsymbol{B})\,\|\boldsymbol{\Delta}\|$$

$$\leqslant\; \frac{4\kappa^2(\boldsymbol{A}_0)}{\sigma_{\min}^2(\boldsymbol{A}_0)}\left\|\frac{1}{\theta mp}\boldsymbol{X}\boldsymbol{X}^\top - \boldsymbol{I}\right\| \;\leqslant\; \frac{4\kappa^2(\boldsymbol{A}_0)}{\sigma_{\min}^2(\boldsymbol{A}_0)}\cdot\epsilon.$$

On the other hand, by using the second inequality in Lemma A.10, we have

$$\left\|\left(\frac{1}{\theta mp}\boldsymbol{Y}\boldsymbol{Y}^\top\right)^{1/2}\left(\boldsymbol{A}_0\boldsymbol{A}_0^\top\right)^{-1/2} - \boldsymbol{I}\right\| \;=\; \left\|(\boldsymbol{B}+\boldsymbol{\Delta})^{1/2}\boldsymbol{B}^{-1/2} - \boldsymbol{I}\right\|$$

$$\leqslant\; 4\sigma_{\min}^{-3/2}(\boldsymbol{B})\,\|\boldsymbol{\Delta}\|$$

$$\leqslant\; \frac{4\kappa^2(\boldsymbol{A}_0)}{\sigma_{\min}(\boldsymbol{A}_0)}\left\|\frac{1}{\theta mp}\boldsymbol{X}\boldsymbol{X}^\top - \boldsymbol{I}\right\| \;\leqslant\; \frac{4\kappa^2(\boldsymbol{A}_0)}{\sigma_{\min}(\boldsymbol{A}_0)}\cdot\epsilon.$$

Choose $\epsilon = \left(\frac{4\kappa^2(\boldsymbol{A}_0)}{\sigma_{\min}^2(\boldsymbol{A}_0)}\right)^{-1}\delta$, we obtain the desired results. ■

Given the definition of preconditioning matrix $\boldsymbol{P}$, the result above leads to the following corollary.

**Corollary E.19** *Under the same settings of Lemma E.18, for any $\delta \in (0, 1)$, whenever*

$$p \;\geqslant\; C\theta^{-1}K^3\frac{\kappa^6(\boldsymbol{A}_0)}{\sigma_{\min}^2(\boldsymbol{A}_0)}\delta^{-2}\log(mK),$$

*we have*

$$\|\boldsymbol{P}\boldsymbol{A}_0 - \boldsymbol{A}\| \;\leqslant\; \delta, \quad \|\boldsymbol{P}^{-1}\| \;\leqslant\; 2K^{-1/2}\|\boldsymbol{A}_0\|,$$

$$\|\boldsymbol{P}\boldsymbol{A}_0\| \;\leqslant\; \|\boldsymbol{A}\| + \delta \;\leqslant\; \sqrt{K} + \delta$$

*hold with probability at least $1 - c_1(mK)^{-c_2}$. Here, $c_1,\ c_2,\ C > 0$ are some numerical constants.*

**Proof** For the first inequality, we have

$$\|\boldsymbol{P}\boldsymbol{A}_0 - \boldsymbol{A}\| \;\leqslant\; \sqrt{K}\left\|\left(\frac{1}{\theta mp}\boldsymbol{Y}\boldsymbol{Y}^\top\right)^{-1/2} - \left(\boldsymbol{A}_0\boldsymbol{A}_0^\top\right)^{-1/2}\right\|\,\|\boldsymbol{A}_0\|\,.$$

Thus, for any $\delta \in (0,1)$, Lemma E.18 implies that whenever

$$p \;\geqslant\; C\theta^{-1}K^3\frac{\kappa^6(\boldsymbol{A}_0)}{\sigma_{\min}^2(\boldsymbol{A}_0)}\delta^{-2}\log(mK),$$

we have

$$\|\boldsymbol{P}\boldsymbol{A}_0 - \boldsymbol{A}\| \;\leqslant\; \delta, \quad \|\boldsymbol{P}\boldsymbol{A}_0\| \;\leqslant\; \|\boldsymbol{A}\| \;+\; \|\boldsymbol{P}\boldsymbol{A}_0 - \boldsymbol{A}\| \;\leqslant\; \sqrt{K} + \delta$$

with probability at least $1 - c_1(mK)^{-c_2}$. On the other hand, by Lemma E.18 we have

$$
\begin{aligned}
\|\boldsymbol{P}^{-1}\| \;&\leqslant\; \left\|\boldsymbol{P}^{-1} - \left(K^{-1}\boldsymbol{A}_0\boldsymbol{A}_0\right)^{1/2}\right\| \;+\; \left\|\left(K^{-1}\boldsymbol{A}_0\boldsymbol{A}_0\right)^{1/2}\right\| \\
&\leqslant\; \left\|\left(K^{-1}\boldsymbol{A}_0\boldsymbol{A}_0\right)^{1/2}\right\|\left(1 \;+\; \left\|\boldsymbol{P}^{-1}\left(K^{-1}\boldsymbol{A}_0\boldsymbol{A}_0\right)^{-1/2} - \boldsymbol{I}\right\|\right) \\
&\leqslant\; K^{-1/2}\|\boldsymbol{A}_0\|\left(1 + \left\|\left(\frac{1}{\theta mp}\boldsymbol{Y}\boldsymbol{Y}^\top\right)^{1/2}\left(\boldsymbol{A}_0\boldsymbol{A}_0^\top\right)^{-1/2} - \boldsymbol{I}\right\|\right) \;\leqslant\; 2K^{-1/2}\|\boldsymbol{A}_0\|\,,
\end{aligned}
$$

as desired. ■

**Lemma E.20** *Suppose* $\boldsymbol{X}$ *satisfies Assumption E.2. For any* $\delta \in (0,1)$*, we have*

$$\left\|\frac{1}{\theta mp}\boldsymbol{X}\boldsymbol{X}^\top - \boldsymbol{I}\right\| \;\leqslant\; \delta, \qquad \|\boldsymbol{X}\| \;\leqslant\; \sqrt{\theta mp}\,(1+\delta)$$

*with probability at least* $1 - c_1 mK\exp\left(-c_2\theta p\min\left\{\left(\frac{\delta}{K}\right)^2, \frac{\delta}{K}\right\}\right)$. *Here,* $c_1,\ c_2 > 0$ *are some numerical constants.*

**Proof** By using the fact that $\boldsymbol{X} = \begin{bmatrix}\boldsymbol{X}_1 & \boldsymbol{X}_2 & \cdots & \boldsymbol{X}_p\end{bmatrix}$, we observe

$$\boldsymbol{X}\boldsymbol{X}^\top \;=\; \sum_{k=1}^p \boldsymbol{X}_k\boldsymbol{X}_k^\top, \quad \boldsymbol{X}_k = \begin{bmatrix}\boldsymbol{C}_{\boldsymbol{x}_{k1}} \\ \vdots \\ \boldsymbol{C}_{\boldsymbol{x}_{kK}}\end{bmatrix}$$

For any $\boldsymbol{z} \in \mathbb{S}^{n-1}$, write $\boldsymbol{z} = \begin{bmatrix}\boldsymbol{z}_1 \\ \vdots \\ \boldsymbol{z}_K\end{bmatrix}$. We have

$$
\begin{aligned}
\left\|\frac{1}{\theta mp}\boldsymbol{X}\boldsymbol{X}^\top - \boldsymbol{I}\right\| \;&=\; \sup_{\boldsymbol{z}\in\mathbb{S}^{n-1}}\left|\boldsymbol{z}^\top\left(\frac{1}{\theta mp}\boldsymbol{X}\boldsymbol{X}^\top - \boldsymbol{I}\right)\boldsymbol{z}\right| \\
&=\; \sup_{\boldsymbol{z}\in\mathbb{S}^{n-1}}\left|\frac{1}{\theta mp}\boldsymbol{z}^\top\left(\sum_{i=1}^p\boldsymbol{X}_i\boldsymbol{X}_i^\top\right)\boldsymbol{z} - \|\boldsymbol{z}\|^2\right| \\
&=\; \sup_{\boldsymbol{z}\in\mathbb{S}^{n-1}}\left|\frac{1}{\theta mp}\sum_{i=1}^p\left(\sum_{k=1}^K\boldsymbol{C}_{\boldsymbol{x}_{ik}}\boldsymbol{z}_k\right)^\top\left(\sum_{k=1}^K\boldsymbol{C}_{\boldsymbol{x}_{ik}}\boldsymbol{z}_k\right) - \|\boldsymbol{z}\|^2\right| \\
&=\; \sup_{\boldsymbol{z}\in\mathbb{S}^{n-1}}\left|\frac{1}{\theta mp}\sum_{i=1}^p\left(\sum_{k=1}^K\boldsymbol{z}_k^\top\boldsymbol{C}_{\boldsymbol{x}_{ik}}^\top\boldsymbol{C}_{\boldsymbol{x}_{ik}}\boldsymbol{z}_k + 2\sum_{k\neq\ell}\boldsymbol{z}_k^\top\boldsymbol{C}_{\boldsymbol{x}_{ik}}^\top\boldsymbol{C}_{\boldsymbol{x}_{i\ell}}\boldsymbol{z}_\ell\right) - \|\boldsymbol{z}\|^2\right| \\
&\leqslant\; \sup_{\boldsymbol{z}\in\mathbb{S}^{n-1}}\sum_{k=1}^K\left|\boldsymbol{z}_k^\top\left(\frac{1}{\theta mp}\sum_{i=1}^p\boldsymbol{C}_{\boldsymbol{x}_{ik}}^\top\boldsymbol{C}_{\boldsymbol{x}_{ik}} - \boldsymbol{I}\right)\boldsymbol{z}_k\right| + 2\sum_{k\neq\ell}\left|\boldsymbol{z}_k^\top\left(\frac{1}{\theta mp}\sum_{i=1}^p\boldsymbol{C}_{\boldsymbol{x}_{ik}}^\top\boldsymbol{C}_{\boldsymbol{x}_{i\ell}}\right)\boldsymbol{z}_\ell\right| \\
&\leqslant\; K^{-1}\sum_{k=1}^K\left\|\frac{1}{\theta np}\sum_{i=1}^p\boldsymbol{C}_{\boldsymbol{x}_{ik}}^\top\boldsymbol{C}_{\boldsymbol{x}_{ik}} - \boldsymbol{I}\right\| + 2K^{-1}\sum_{k\neq\ell}\left\|\frac{1}{\theta np}\sum_{i=1}^p\boldsymbol{C}_{\boldsymbol{x}_{ik}}^\top\boldsymbol{C}_{\boldsymbol{x}_{i\ell}}\right\|.
\end{aligned}
$$

By Lemma E.21, we obtain

$$\left\| \frac{1}{\theta m p} \boldsymbol{X} \boldsymbol{X}^\top - \boldsymbol{I} \right\| \;\leqslant\; t_1 + 2K t_2 \;\leqslant\; \delta$$

with probability at least

$$1 - 2m \exp\left( -c_1 \theta p \min\left\{ \delta^2, \delta \right\} \right) - 2mK \exp\left( -c_2 \theta p \min\left\{ \left(K^{-1}\delta\right)^2, K^{-1}\delta \right\} \right).$$

Finally, the second inequality directly follows from the fact that

$$\left\| \frac{1}{\theta m p} \boldsymbol{X} \boldsymbol{X}^\top - \boldsymbol{I} \right\| \;\leqslant\; \delta \quad \implies \quad \|\boldsymbol{X}\|^2 \;\leqslant\; (\theta m p)\,(1 + \delta),$$

as desired. ∎

**Lemma E.21** *Suppose $\boldsymbol{x}_{ij}$ satisfies Assumption E.2. For any $j \in [K]$, we have*

$$\left\| \frac{1}{\theta n p} \sum_{i=1}^p \boldsymbol{C}_{\boldsymbol{x}_{ij}}^\top \boldsymbol{C}_{\boldsymbol{x}_{ij}} - \boldsymbol{I} \right\| \;\leqslant\; t_1$$

*holding with probability at least $1 - 2m \exp\left( -\frac{\theta p}{8} \min\left\{ \frac{t_1^2}{2}, t_1 \right\} \right)$. Moreover, for any $k, \ell \in [K]$ with $k \neq \ell$, we have*

$$\left\| \frac{1}{\theta n p} \sum_{i=1}^p \boldsymbol{C}_{\boldsymbol{x}_{ik}}^\top \boldsymbol{C}_{\boldsymbol{x}_{i\ell}} \right\| \;\leqslant\; t_2$$

*holding with probability at least $1 - 2\frac{m^2}{n} \exp\left( -\frac{\theta p}{2} \min\left\{ t_2^2, t_2 \right\} \right)$.*

**Proof**  Notice that

$$\boldsymbol{C}_{\boldsymbol{x}_{ij}}^\top \boldsymbol{C}_{\boldsymbol{x}_{ij}} \;=\; \boldsymbol{F}^* \operatorname{diag}\left( |\boldsymbol{F}\boldsymbol{x}_{ij}|^{\odot 2} \right) \boldsymbol{F}, \qquad \boldsymbol{C}_{\boldsymbol{x}_{ik}}^\top \boldsymbol{C}_{\boldsymbol{x}_{i\ell}} \;=\; \boldsymbol{F}^* \operatorname{diag}\left( \overline{\boldsymbol{F}\boldsymbol{x}_{ik}} \odot \boldsymbol{F}\boldsymbol{x}_{i\ell} \right) \boldsymbol{F}. \quad \text{(E.14)}$$

**Bounding** $\left\| \frac{1}{\theta n p} \sum_{i=1}^p \boldsymbol{C}_{\boldsymbol{x}_{ij}}^\top \boldsymbol{C}_{\boldsymbol{x}_{ij}} - \boldsymbol{I} \right\|$.  From Equation (E.14), we have

$$\left\| \frac{1}{\theta n p} \sum_{i=1}^p \boldsymbol{C}_{\boldsymbol{x}_{ij}}^\top \boldsymbol{C}_{\boldsymbol{x}_{ij}} - \boldsymbol{I} \right\| \;=\; \left\| \boldsymbol{F}^* \operatorname{diag}\left( \frac{1}{\theta n p} \sum_{i=1}^p |\boldsymbol{F}\boldsymbol{x}_{ij}|^{\odot 2} - \boldsymbol{1} \right) \boldsymbol{F} \right\|$$

$$\leqslant\; \left\| \frac{1}{\theta n p} \sum_{i=1}^p |\boldsymbol{F}\boldsymbol{x}_{ij}|^{\odot 2} - \boldsymbol{1} \right\|_\infty.$$

Let $\boldsymbol{f}_k^*$ be a row of $\boldsymbol{F}$, by Lemma A.3 we have for any $\ell \geqslant 1$,

$$\mathbb{E}\left[ |\boldsymbol{f}_k^* \boldsymbol{x}_{ij}|^{2\ell} \right] \;\leqslant\; \frac{2^\ell \ell!}{2} \mathbb{E}_{\boldsymbol{b}_k \sim \mathrm{Ber}(\theta)} \left[ \|\boldsymbol{b}_k \odot \boldsymbol{f}_k\|^{2\ell} \right] \;\leqslant\; \frac{\ell!}{2} \theta (2n)^\ell.$$

Thus, by Bernstein inequality in Lemma A.5, we have

$$\mathbb{P}\left( \left| \frac{1}{\theta n p} \sum_{i=1}^p |\boldsymbol{f}_k^* \boldsymbol{x}_{ij}|^{\odot 2} - 1 \right| \geqslant t_1 \right) \;\leqslant\; 2 \exp\left( -\frac{p \theta t_1^2}{8 + 4 t_1} \right).$$

Thus, by using union bounds, we obtain

$$\left\| \frac{1}{\theta n p} \sum_{i=1}^p \boldsymbol{C}_{\boldsymbol{x}_{ij}}^\top \boldsymbol{C}_{\boldsymbol{x}_{ij}} - \boldsymbol{I} \right\| \;\leqslant\; \left\| \frac{1}{\theta n p} \sum_{i=1}^p |\boldsymbol{F}\boldsymbol{x}_{ij}|^{\odot 2} - \boldsymbol{1} \right\|_\infty \;\leqslant\; t_1$$

for all $1 \leqslant j \leqslant K$ with probability at least $1 - 2nK \exp\left( -\frac{\theta p}{8} \min\left\{ \frac{t_1^2}{2}, t_1 \right\} \right)$.

**Bounding** $\left\| \frac{1}{\theta n p} \sum_{i=1}^{p} \boldsymbol{C}_{\boldsymbol{x}_{ik}}^{\top} \boldsymbol{C}_{\boldsymbol{x}_{i\ell}} \right\|$. On the other hand, by Equation (E.14), we know that

$$\left\| \frac{1}{\theta n p} \sum_{i=1}^{p} \boldsymbol{C}_{\boldsymbol{x}_{ik}}^{\top} \boldsymbol{C}_{\boldsymbol{x}_{i\ell}} \right\| \leqslant \left\| \frac{1}{\theta n p} \sum_{i=1}^{p} \overline{\boldsymbol{F} \boldsymbol{x}_{ik}} \odot \boldsymbol{F} \boldsymbol{x}_{i\ell} \right\|_{\infty}.$$

Let $z_{id}^{k\ell} = \overline{\boldsymbol{f}_d^* \boldsymbol{x}_{ik}} \boldsymbol{f}_d^* \boldsymbol{x}_{i\ell} = \boldsymbol{x}_{ik}^{\top} \boldsymbol{f}_d \boldsymbol{f}_d^* \boldsymbol{x}_{i\ell}$ $(1 \leqslant d \leqslant n)$, we have its moments for $s \geqslant 1$

$$\mathbb{E}\left[ \left| z_{id}^{k\ell} \right|^s \right] \leqslant \mathbb{E}\left[ \left| \boldsymbol{x}_{ik}^{\top} \boldsymbol{f}_d \right|^s \right] \mathbb{E}\left[ \left| \boldsymbol{f}_d^* \boldsymbol{x}_{i\ell} \right|^s \right] \leqslant \frac{s!}{2} \mathbb{E}_{\boldsymbol{b}_d \sim \mathrm{Ber}(\theta)} \left[ \left\| \boldsymbol{b}_d \odot \boldsymbol{f}_d \right\|^{2s} \right] \leqslant \frac{s!}{2} \theta n^s.$$

Thus, by Bernstein inequality in Lemma A.5, we obtain

$$\mathbb{P}\left( \frac{1}{\theta n p} \left| \sum_{i=1}^{p} z_{id}^{k\ell} \right| \geqslant t_2 \right) \leqslant 2 \exp\left( -\frac{\theta p t_2^2}{2 + 2t_2} \right).$$

Thus, by applying union bounds, we have

$$\left\| \frac{1}{\theta n p} \sum_{i=1}^{p} \boldsymbol{C}_{\boldsymbol{x}_{ik}}^{\top} \boldsymbol{C}_{\boldsymbol{x}_{i\ell}} \right\| \leqslant t_2$$

for all $1 \leqslant k, \ell \leqslant K$ and $k \neq \ell$ with probability at least $1 - 2mK \exp\left( -\frac{\theta p}{2} \min\{t_2^2, t_2\} \right)$. ∎

# F MEASURE CONCENTRATION

In this part of the appendix, we show measure concentration of Riemannian gradient and Hessian for both $\varphi_{\mathrm{DL}}(\boldsymbol{q})$ and $\varphi_{\mathrm{CDL}}(\boldsymbol{q})$ over the sphere. Before that, we first show the following preliminary results that are key for our proof. For simplicity, we also use $K = m/n$ throughout the section.

## F.1 PRELIMINARY RESULTS

Here, as the gradient and Hessian of $\ell^4$-loss is heavy-tailed, traditional concentration tools do not directly apply to our cases. Therefore, we first develop some general tools for concentrations of superema of heavy-tailed empirical process over the sphere. In later part of this appendix, we will apply these results for concentration of Riemannian gradient and Hessian for both overcomplete dictionary learning and convolutional dictionary learning.

**Theorem F.1 (Concentration of heavy-tailed random matrices over the sphere)** *Let* $\boldsymbol{Z}_1, \boldsymbol{Z}_2, \cdots, \boldsymbol{Z}_p \in \mathbb{R}^{n_1 \times n_2}$ *be i.i.d. centered subgaussian random matrices, with* $\boldsymbol{Z}_i \equiv_{\mathrm{d}} \boldsymbol{Z}$ $(1 \leqslant i \leqslant p)$ *and*

$$\mathbb{E}\left[ Z_{ij} \right] = 0, \qquad \mathbb{P}\left( \left| Z_{ij} \right| > t \right) \leqslant 2 \exp\left( -\frac{t^2}{2\sigma^2} \right).$$

*For a fixed* $\boldsymbol{q} \in \mathbb{S}^{n-1}$, *let us define a function* $f_{\boldsymbol{q}}(\cdot) : \mathbb{R}^{n_1 \times n_2} \mapsto \mathbb{R}^{d_1 \times d_2}$, *such that*

1. $f_{\boldsymbol{q}}(\boldsymbol{Z})$ *is a heavy tailed process of* $\boldsymbol{Z}$, *in the sense of* $\mathbb{P}\left( \left\| f_{\boldsymbol{q}}(\boldsymbol{Z}) \right\| \geqslant t \right) \leqslant 2 \exp\left( -C\sqrt{t} \right)$.

2. *The expectation* $\mathbb{E}\left[ f_{\boldsymbol{q}}(\boldsymbol{Z}) \right]$ *is bounded and* $L_f$-*Lipschitz, i.e.,*

$$\left\| \mathbb{E}\left[ f_{\boldsymbol{q}}(\boldsymbol{Z}) \right] \right\| \leqslant B_f, \qquad and \qquad \left\| \mathbb{E}\left[ f_{\boldsymbol{q}_1}(\boldsymbol{Z}) \right] - \mathbb{E}\left[ f_{\boldsymbol{q}_2}(\boldsymbol{Z}) \right] \right\| \leqslant L_f \left\| \boldsymbol{q}_1 - \boldsymbol{q}_2 \right\|, \ \forall \ \boldsymbol{q}_1, \ \boldsymbol{q}_2 \in \mathbb{S}^{n-1}. \tag{F.1}$$

3. *Let* $\overline{\boldsymbol{Z}}$ *be a truncated random matrix of* $\boldsymbol{Z}$, *such that*

$$\boldsymbol{Z} = \overline{\boldsymbol{Z}} + \widehat{\boldsymbol{Z}}, \qquad \overline{Z}_{ij} = \begin{cases} Z_{ij} & \text{if } |Z_{ij}| < B, \\ 0 & \text{otherwise.} \end{cases} \tag{F.2}$$

*with* $B = 2\sigma\sqrt{\log(n_1 n_2 p)}$. *For the truncated matrix* $\overline{\boldsymbol{Z}}$, *we further assume that*

$$\left\| f_{\boldsymbol{q}}(\overline{\boldsymbol{Z}}) \right\| \leqslant R_1(\sigma), \quad \max\left\{ \left\| \mathbb{E}\left[ f_{\boldsymbol{q}}(\overline{\boldsymbol{Z}})^{\top} f_{\boldsymbol{q}}(\overline{\boldsymbol{Z}}) \right] \right\|, \ \left\| f_{\boldsymbol{q}}(\overline{\boldsymbol{Z}}) f_{\boldsymbol{q}}(\overline{\boldsymbol{Z}})^{\top} \right\| \right\} \leqslant R_2(\sigma), \tag{F.3}$$

$$\left\| f_{\boldsymbol{q}_1}(\overline{\boldsymbol{Z}}) - f_{\boldsymbol{q}_2}(\overline{\boldsymbol{Z}}) \right\| \leqslant \overline{L}_f(\sigma) \left\| \boldsymbol{q}_1 - \boldsymbol{q}_2 \right\|, \ \forall \ \boldsymbol{q}_1, \ \boldsymbol{q}_2 \in \mathbb{S}^{n-1}. \tag{F.4}$$

*Then for any $\delta \in \left(0, 6\frac{R_2}{R_1}\right)$, whenever*

$$p \;\geqslant\; C \max\left\{ \frac{\min\{d_1, d_2\} B_f}{n_1 n_2 \delta}, \; \delta^{-2} R_2 \left[ n \log\left( \frac{6\left(L_f + \overline{L}_f\right)}{\delta} \right) + \log(d_1 + d_2) \right] \right\}$$

*we have*

$$\sup_{\boldsymbol{q} \in \mathbb{S}^{n-1}} \left\| \frac{1}{p} \sum_{i=1}^{p} f_{\boldsymbol{q}}(\overline{\boldsymbol{Z}}_i) - \mathbb{E}\left[f_{\boldsymbol{q}}(\boldsymbol{Z})\right] \right\| \;\leqslant\; \delta,$$

*holding with probability at least $1 - (n_1 n_2 p)^{-2} - n^{-c\log\left((L_f + \overline{L}_f)/\delta\right)}$ for some constant $c, C > 0$.*

**Proof** As aforementioned, traditional concentration tools does not directly apply due to the heavy-tailed behavior of $f_{\boldsymbol{q}}(\boldsymbol{Z})$. To circumvent the difficulties, we first truncate $\boldsymbol{Z}$ and introduce bounded random variable $\overline{\boldsymbol{Z}}$ as in Equation (F.2), with truncation level $B = 2\sigma\sqrt{\log(n_1 n_2 p)}$. Thus, we have

$$\mathbb{P}\left( \sup_{\boldsymbol{q} \in \mathbb{S}^{n-1}} \left\| \frac{1}{p} \sum_{i=1}^{p} f_{\boldsymbol{q}}(\boldsymbol{Z}_i) - \mathbb{E}\left[f_{\boldsymbol{q}}(\boldsymbol{Z})\right] \right\| \;\geqslant\; t \right)$$

$$\leqslant \underbrace{\mathbb{P}\left( \sup_{\boldsymbol{q} \in \mathbb{S}^{n-1}} \left\| \frac{1}{p} \sum_{i=1}^{p} f_{\boldsymbol{q}}(\overline{\boldsymbol{Z}}_i) - \mathbb{E}\left[f_{\boldsymbol{q}}(\boldsymbol{Z})\right] \right\| \;\geqslant\; t \right)}_{\mathcal{P}_1(t)} + \underbrace{\mathbb{P}\left( \max_{1 \leqslant i \leqslant p} \|\boldsymbol{Z}_i\|_\infty \;\geqslant\; B \right)}_{\mathcal{P}_2}.$$

As $f_{\boldsymbol{q}}(\overline{\boldsymbol{Z}})$ is also bounded, then we can apply classical concentration tools to $\mathcal{P}_1(t)$, and bound $\mathcal{P}_2$ by using subgaussian tails of $\boldsymbol{Z}$. In the following, we make this argument rigorous with more technical details.

**Tail bound for $\mathcal{P}_2$.** Since $Z_{jk}^i$ is centered subgaussian, by an union bound, we have

$$\mathcal{P}_2 \;=\; \mathbb{P}\left( \max_{1 \leqslant i \leqslant p} \|\boldsymbol{Z}_i\|_\infty \;\geqslant\; B \right) \;\leqslant\; n_1 n_2 p \mathbb{P}\left( |Z_{jk}^i| \;\geqslant\; B \right) \;\leqslant\; \exp\left( -\frac{B^2}{2\sigma^2} + \log(n_1 n_2 p) \right).$$

Choose $B = 2\sigma\sqrt{\log(n_1 n_2 p)}$, we obtain

$$\mathcal{P}_2 \;=\; \mathbb{P}\left( \max_{1 \leqslant i \leqslant p} \|\boldsymbol{Z}_i\|_\infty \;\geqslant\; B \right) \;\leqslant\; (n_1 n_2 p)^{-2}.$$

**Tail Bound for $\left\| \frac{1}{p} \sum_{i=1}^{p} f_{\boldsymbol{q}}(\overline{\boldsymbol{Z}}_i) - \mathbb{E}\left[f_{\boldsymbol{q}}(\boldsymbol{Z})\right] \right\|$ with a fixed $\boldsymbol{q} \in \mathbb{S}^{n-1}$.** First, we control the quantity for a given $\boldsymbol{q} \in \mathbb{S}^{n-1}$. Later, we will turn the tail bound result to a uniform bound over the sphere for all $\boldsymbol{q} \in \mathbb{S}^{n-1}$. We first apply triangle inequality, where we have

$$\left\| \frac{1}{p} \sum_{i=1}^{p} f_{\boldsymbol{q}}(\overline{\boldsymbol{Z}}_i) - \mathbb{E}\left[f_{\boldsymbol{q}}(\boldsymbol{Z})\right] \right\| \;\leqslant\; \left\| \frac{1}{p} \sum_{i=1}^{p} f_{\boldsymbol{q}}(\overline{\boldsymbol{Z}}_i) - \mathbb{E}\left[f_{\boldsymbol{q}}(\overline{\boldsymbol{Z}})\right] \right\| + \left\| \mathbb{E}\left[f_{\boldsymbol{q}}(\boldsymbol{Z})\right] - \mathbb{E}\left[f_{\boldsymbol{q}}(\overline{\boldsymbol{Z}})\right] \right\|,$$

such that

$$\mathbb{P}\left( \left\| \frac{1}{p} \sum_{i=1}^{p} f_{\boldsymbol{q}}(\overline{\boldsymbol{Z}}_i) - \mathbb{E}\left[f_{\boldsymbol{q}}(\boldsymbol{Z})\right] \right\| \;\geqslant\; t \right)$$

$$\leqslant \mathbb{P}\left( \left\| \frac{1}{p} \sum_{i=1}^{p} f_{\boldsymbol{q}}(\overline{\boldsymbol{Z}}_i) - \mathbb{E}\left[f_{\boldsymbol{q}}(\overline{\boldsymbol{Z}})\right] \right\| \;\geqslant\; t - \left\| \mathbb{E}\left[f_{\boldsymbol{q}}(\boldsymbol{Z})\right] - \mathbb{E}\left[f_{\boldsymbol{q}}(\overline{\boldsymbol{Z}})\right] \right\| \right).$$

Notice that

$$\left\| \mathbb{E}\left[f_{\boldsymbol{q}}(\overline{\boldsymbol{Z}})\right] - \mathbb{E}\left[f_{\boldsymbol{q}}(\boldsymbol{Z})\right] \right\| \;\leqslant\; \left\| \mathbb{E}\left[f_{\boldsymbol{q}}(\boldsymbol{Z}) \odot \mathbb{1}_{\boldsymbol{Z} \neq \overline{\boldsymbol{Z}}}\right] \right\|_F \;\leqslant\; \left\| \mathbb{E}\left[f_{\boldsymbol{q}}(\boldsymbol{Z})\right] \right\|_F \left\| \mathbb{E}\left[\mathbb{1}_{\boldsymbol{Z} \neq \overline{\boldsymbol{Z}}}\right] \right\|_F$$

$$\leqslant\; \min\{d_1, d_2\} B_f \sqrt{ \sum_{ij} \mathbb{P}\left( Z_{ij} \neq \overline{Z}_{ij} \right) }$$

$$\leqslant\; \min\{d_1, d_2\} B_f \sqrt{ n_1 n_2 \exp\left( -\frac{B^2}{2\sigma^2} \right) },$$

where for the second inequality we used Cauchy-Schwarz inequality, the third one follows from and the last one follows from the fact in $Z$ is subgaussian. With $B = 2\sigma\sqrt{\log{(n_1 n_2 p)}}$, we obtain

$$\left\| \mathbb{E}\left[ f_{\boldsymbol{q}}(\overline{\boldsymbol{Z}}) \right] - \mathbb{E}\left[ f_{\boldsymbol{q}}(\boldsymbol{Z}) \right] \right\| \leqslant \frac{\min\{d_1, d_2\} B_f}{n_1 n_2 p},$$

so that

$$\mathbb{P}\left( \left\| \frac{1}{p}\sum_{i=1}^{p} f_{\boldsymbol{q}}(\overline{\boldsymbol{Z}}_i) - \mathbb{E}\left[ f_{\boldsymbol{q}}(\boldsymbol{Z}) \right] \right\| \geqslant t \right) \leqslant \mathbb{P}\left( \left\| \frac{1}{p}\sum_{i=1}^{p} f_{\boldsymbol{q}}(\overline{\boldsymbol{Z}}_i) - \mathbb{E}\left[ f_{\boldsymbol{q}}(\overline{\boldsymbol{Z}}) \right] \right\| \geqslant t - \frac{B_f}{n_1 n_2 p} \right).$$

Next, we need to show concentration of $\left\| \frac{1}{p}\sum_{i=1}^{p} f_{\boldsymbol{q}}(\overline{\boldsymbol{Z}}_i) - \mathbb{E}\left[ f_{\boldsymbol{q}}(\overline{\boldsymbol{Z}}) \right] \right\|$ to finish this part of proof. By our assumption in Equation (F.3), we apply bounded Bernstein's inequality in Lemma A.7, such that

$$\mathbb{P}\left( \left\| \frac{1}{p}\sum_{i=1}^{p} f_{\boldsymbol{q}}(\overline{\boldsymbol{Z}}_i) - \mathbb{E}\left[ f_{\boldsymbol{q}}(\overline{\boldsymbol{Z}}) \right] \right\| \geqslant t_1 \right) \leqslant (d_1 + d_2)\exp\left( -\frac{p t_1^2}{2R_2 + 4R_1 t_2/3} \right).$$

Choose $p$ large enough such that

$$p \geqslant \frac{2\min\{d_1, d_2\} B_f}{n_1 n_2 t} \quad \Longrightarrow \quad \frac{\min\{d_1, d_2\} B_f}{n_1 n_2 p} \leqslant \frac{t}{2}.$$

Thus, for a fixed $\boldsymbol{q} \in \mathbb{S}^{n-1}$, we have

$$\mathbb{P}\left( \left\| \frac{1}{p}\sum_{i=1}^{p} f_{\boldsymbol{q}}(\overline{\boldsymbol{Z}}_i) - \mathbb{E}\left[ f_{\boldsymbol{q}}(\boldsymbol{Z}) \right] \right\| \geqslant t \right) \leqslant \mathbb{P}\left( \left\| \frac{1}{p}\sum_{i=1}^{p} f_{\boldsymbol{q}}(\overline{\boldsymbol{Z}}_i) - \mathbb{E}\left[ f_{\boldsymbol{q}}(\overline{\boldsymbol{Z}}) \right] \right\| \geqslant t/2 \right)$$

$$\leqslant (d_1 + d_2)\exp\left( -\frac{p t^2}{8R_2 + 8R_1 t/3} \right).$$

**Bounding $\mathcal{P}_1(t)$ via covering over the sphere $\mathbb{S}^{n-1}$.** Finally, we finish by . Let $\mathcal{N}(\epsilon)$ be an epsilon net of the sphere, where we know that

$$\forall\, \boldsymbol{q} \in \mathbb{S}^{n-1}, \quad \exists\, \boldsymbol{q}' \in \mathcal{N}(\epsilon), \quad \text{s.t. } \left\| \boldsymbol{q} - \boldsymbol{q}' \right\| \leqslant \epsilon, \qquad \text{and} \quad \#\mathcal{N}(\epsilon) \leqslant \left( \frac{3}{\epsilon} \right)^{n-1}.$$

Thus, we have

$$\sup_{\boldsymbol{q}\in\mathbb{S}^{n-1}} \left\| \frac{1}{p}\sum_{i=1}^{p} f_{\boldsymbol{q}}(\overline{\boldsymbol{Z}}_i) - \mathbb{E}\left[ f_{\boldsymbol{q}}(\boldsymbol{Z}) \right] \right\|$$

$$= \sup_{\boldsymbol{q}'\in\mathcal{N}(\epsilon), \|\boldsymbol{e}\|\leqslant\epsilon} \left\| \frac{1}{p}\sum_{i=1}^{p} f_{\boldsymbol{q}'+\boldsymbol{e}}(\overline{\boldsymbol{Z}}_i) - \mathbb{E}\left[ f_{\boldsymbol{q}'+\boldsymbol{e}}(\boldsymbol{Z}) \right] \right\|$$

$$\leqslant \sup_{\boldsymbol{q}'\in\mathcal{N}(\epsilon)} \left\| \frac{1}{p}\sum_{i=1}^{p} f_{\boldsymbol{q}'}(\overline{\boldsymbol{Z}}_i) - \mathbb{E}\left[ f_{\boldsymbol{q}'}(\boldsymbol{Z}) \right] \right\| + \sup_{\boldsymbol{q}'\in\mathcal{N}(\epsilon), \|\boldsymbol{e}\|\leqslant\epsilon} \left\| \frac{1}{p}\sum_{i=1}^{p} f_{\boldsymbol{q}'+\boldsymbol{e}}(\overline{\boldsymbol{Z}}_i) - \frac{1}{p}\sum_{i=1}^{p} f_{\boldsymbol{q}'}(\overline{\boldsymbol{Z}}_i) \right\|$$

$$+ \sup_{\boldsymbol{q}'\in\mathcal{N}(\epsilon), \|\boldsymbol{e}\|\leqslant\epsilon} \left\| \mathbb{E}\left[ f_{\boldsymbol{q}'+\boldsymbol{e}}(\boldsymbol{Z}) \right] - \mathbb{E}\left[ f_{\boldsymbol{q}'}(\boldsymbol{Z}) \right] \right\|.$$

By our Lipschitz continuity assumption in Equation (F.1) and Equation (F.4), for any $\boldsymbol{q} \in \mathbb{S}^{n-1}$, we obtain

$$\left\| \mathbb{E}\left[ f_{\boldsymbol{q}'+\boldsymbol{e}}(\boldsymbol{Z}) \right] - \mathbb{E}\left[ f_{\boldsymbol{q}'}(\boldsymbol{Z}) \right] \right\| \leqslant L_f \left\| \boldsymbol{e} \right\|,$$

$$\left\| \frac{1}{p}\sum_{i=1}^{p} f_{\boldsymbol{q}'+\boldsymbol{e}}(\overline{\boldsymbol{Z}}_i) - \frac{1}{p}\sum_{i=1}^{p} f_{\boldsymbol{q}'}(\overline{\boldsymbol{Z}}_i) \right\| \leqslant \left\| f_{\boldsymbol{q}'+\boldsymbol{e}}(\overline{\boldsymbol{Z}}) - f_{\boldsymbol{q}'}(\overline{\boldsymbol{Z}}) \right\| \leqslant \overline{L}_f \left\| \boldsymbol{e} \right\|,$$

which implies that

$$\sup_{\boldsymbol{q}\in\mathbb{S}^{n-1}} \left\| \frac{1}{p}\sum_{i=1}^{p} f_{\boldsymbol{q}}(\overline{\boldsymbol{Z}}_i) - \mathbb{E}\left[ f_{\boldsymbol{q}}(\boldsymbol{Z}) \right] \right\| \leqslant \sup_{\boldsymbol{q}'\in\mathcal{N}(\epsilon)} \left\| \frac{1}{p}\sum_{i=1}^{p} f_{\boldsymbol{q}'}(\overline{\boldsymbol{Z}}_i) - \mathbb{E}\left[ f_{\boldsymbol{q}'}(\boldsymbol{Z}) \right] \right\| + \left( L_f + \overline{L}_f \right)\epsilon.$$

Therefore, for any $t > 0$, choose

$$\epsilon \;\leqslant\; \frac{t}{2(L_f + \overline{L}_f)},$$

so that we obtain

$$\mathbb{P}\left(\sup_{\boldsymbol{q}\in\mathbb{S}^{n-1}}\left\|\frac{1}{p}\sum_{i=1}^{p}f_{\boldsymbol{q}}(\overline{\boldsymbol{Z}}_i) - \mathbb{E}\left[f_{\boldsymbol{q}}(\boldsymbol{Z})\right]\right\| \;\geqslant\; t\right)$$

$$\leqslant \mathbb{P}\left(\sup_{\boldsymbol{q}'\in\mathcal{N}(\epsilon)}\left\|\frac{1}{p}\sum_{i=1}^{p}f_{\boldsymbol{q}'}(\overline{\boldsymbol{Z}}_i) - \mathbb{E}\left[f_{\boldsymbol{q}'}(\boldsymbol{Z})\right]\right\| \;\geqslant\; t - \left(L_f + \overline{L}_f\right)\epsilon\right)$$

$$\leqslant \mathbb{P}\left(\sup_{\boldsymbol{q}'\in\mathcal{N}(\epsilon)}\left\|\frac{1}{p}\sum_{i=1}^{p}f_{\boldsymbol{q}'}(\overline{\boldsymbol{Z}}_i) - \mathbb{E}\left[f_{\boldsymbol{q}'}(\boldsymbol{Z})\right]\right\| \;\geqslant\; t/2\right)$$

$$\leqslant \#\mathcal{N}(\epsilon)\cdot\mathbb{P}\left(\left\|\frac{1}{p}\sum_{i=1}^{p}f_{\boldsymbol{q}}(\overline{\boldsymbol{Z}}_i) - \mathbb{E}\left[f_{\boldsymbol{q}}(\boldsymbol{Z})\right]\right\| \;\geqslant\; t/2\right)$$

$$\leqslant \left(\frac{3}{\epsilon}\right)^{n-1}(d_1 + d_2)\exp\left(-\frac{pt^2}{32R_2 + 16R_1 t/3}\right)$$

$$\leqslant \exp\left(-\min\left\{\frac{pt^2}{64R_2}, \frac{3pt}{32R_1}\right\} + n\log\left(\frac{6\left(L_f + \overline{L}_f\right)}{t}\right) + \log(d_1 + d_2)\right).$$

**Summary of the results.** Therefore, combining all the results above, for any $\delta \in \left(0, 6\frac{R_2}{R_1}\right)$, whenever

$$p \;\geqslant\; C\max\left\{\frac{\min\left\{d_1, d_2\right\}B_f}{n_1 n_2 \delta},\; \delta^{-2}R_2\left[n\log\left(\frac{6\left(L_f + \overline{L}_f\right)}{\delta}\right) + \log(d_1 + d_2)\right]\right\},$$

we have

$$\sup_{\boldsymbol{q}\in\mathbb{S}^{n-1}}\left\|\frac{1}{p}\sum_{i=1}^{p}f_{\boldsymbol{q}}(\boldsymbol{Z}_i) - \mathbb{E}\left[f_{\boldsymbol{q}}(\boldsymbol{Z})\right]\right\| \;\leqslant\; \delta,$$

holding with probability at least $1 - (n_1 n_2 p)^{-2} - n^{-c\log\left((L_f + \overline{L}_f)/\delta\right)}$ for some constant $c, C > 0$. $\blacksquare$

**Corollary F.2 (Concentration of heavy-tailed random vectors over the sphere)** *Let* $\boldsymbol{z}_1, \boldsymbol{z}_2, \cdots, \boldsymbol{z}_p \in \mathbb{R}^{n_1}$ *be i.i.d. centered subgaussian random matrices, with* $\boldsymbol{z}_i \equiv_{\mathrm{d}} \boldsymbol{z}\ (1 \leqslant i \leqslant p)$ *and*

$$\mathbb{E}\left[z_i\right] \;=\; 0, \qquad \mathbb{P}\left(|z_i| > t\right) \;\leqslant\; 2\exp\left(-\frac{t^2}{2\sigma^2}\right).$$

*For a fixed* $\boldsymbol{q} \in \mathbb{S}^{n-1}$*, let us define a function* $f_{\boldsymbol{q}}(\cdot) : \mathbb{R}^{n_1} \mapsto \mathbb{R}^{d_1}$*, such that*

1. $f_{\boldsymbol{q}}(\boldsymbol{z})$ *is a heavy tailed process of* $\boldsymbol{z}$*, in the sense of* $\mathbb{P}\left(\|f_{\boldsymbol{q}}(\boldsymbol{z})\| \geqslant t\right) \leqslant 2\exp\left(-C\sqrt{t}\right)$*.*

2. *The expectation* $\mathbb{E}\left[f_{\boldsymbol{q}}(\boldsymbol{z})\right]$ *is bounded and* $L_f$*-Lipschitz, i.e.,*

$$\|\mathbb{E}\left[f_{\boldsymbol{q}}(\boldsymbol{z})\right]\| \;\leqslant\; B_f, \qquad and \quad \|\mathbb{E}\left[f_{\boldsymbol{q}_1}(\boldsymbol{z})\right] - \mathbb{E}\left[f_{\boldsymbol{q}_2}(\boldsymbol{z})\right]\| \;\leqslant\; L_f\|\boldsymbol{q}_1 - \boldsymbol{q}_2\|,\ \forall\, \boldsymbol{q}_1,\, \boldsymbol{q}_2 \in \mathbb{S}^{n-1}.$$
$$\text{(F.5)}$$

3. *Let* $\overline{\boldsymbol{z}}$ *be a truncated random matrix of* $\boldsymbol{z}$*, such that*

$$\boldsymbol{z} \;=\; \overline{\boldsymbol{z}} + \widehat{\boldsymbol{z}}, \qquad \overline{z}_i \;=\; \begin{cases} z_i & if\ |z_i| < B, \\ 0 & otherwise. \end{cases} \tag{F.6}$$

*with* $B = 2\sigma\sqrt{\log(n_1 p)}$*. For the truncated matrix* $\overline{\boldsymbol{z}}$*, we further assume that*

$$\|f_{\boldsymbol{q}}(\overline{\boldsymbol{z}})\| \;\leqslant\; R_1(\sigma), \quad \mathbb{E}\left[\|f_{\boldsymbol{q}}(\overline{\boldsymbol{z}})\|^2\right] \;\leqslant\; R_2(\sigma), \tag{F.7}$$

$$\|f_{\boldsymbol{q}_1}(\overline{\boldsymbol{z}}) - f_{\boldsymbol{q}_2}(\overline{\boldsymbol{z}})\| \;\leqslant\; \overline{L}_f(\sigma)\|\boldsymbol{q}_1 - \boldsymbol{q}_2\|,\ \forall\, \boldsymbol{q}_1,\, \boldsymbol{q}_2 \in \mathbb{S}^{n-1}. \tag{F.8}$$

*Then for any* $\delta \in \left(0, 6\frac{R_2}{R_1}\right)$, *whenever*

$$p \;\geqslant\; C \max\left\{\frac{B_f}{n_1\delta},\; \delta^{-2}R_2\left[n\log\left(\frac{6\left(L_f + \overline{L}_f\right)}{\delta}\right) + \log(d_1)\right]\right\},$$

*we have*

$$\sup_{\boldsymbol{q}\in\mathbb{S}^{n-1}} \left\|\frac{1}{p}\sum_{i=1}^{p} f_{\boldsymbol{q}}(\boldsymbol{z}_i) - \mathbb{E}\left[f_{\boldsymbol{q}}(\boldsymbol{z})\right]\right\| \;\leqslant\; \delta,$$

*holding with probability at least* $1 - (n_1 p)^{-2} - n^{-c\log\left((L_f + \overline{L}_f)/\delta\right)}$ *for some constant* $c, C > 0$.

**Proof** The proof is analogous to that of Theorem F.1. The slight difference is that we need to apply vector version Bernstein's inequality in Lemma A.8 instead of matrix version in Lemma A.7, by utilizing our assumption in Equation (F.7). We omit the detailed proof here. ∎

## F.2    CONCENTRATION FOR OVERCOMPLETE DICTIONARY LEARNING

In this part of appendix, we assume that the dictionary $\boldsymbol{A}$ is tight frame with $\ell^2$-norm bounded columns

$$\frac{1}{K}\boldsymbol{A}\boldsymbol{A}^\top \;=\; \boldsymbol{I}, \quad \|\boldsymbol{a}_i\| \;\leqslant\; M \;\; (1 \leqslant i \leqslant m). \tag{F.9}$$

for some $M$ with $1 \leqslant M \leqslant \sqrt{K}$.

### F.2.1    CONCENTRATION OF $\operatorname{grad}\varphi_{\mathrm{DL}}(\cdot)$

First, we show concentration of $\operatorname{grad}\varphi_{\mathrm{DL}}(\boldsymbol{q})$ to its expectation $\mathbb{E}\left[\operatorname{grad}\varphi_{\mathrm{DL}}(\boldsymbol{q})\right] = \operatorname{grad}\varphi_{\mathrm{T}}(\boldsymbol{q})$,

$$\operatorname{grad}\varphi_{\mathrm{DL}}(\boldsymbol{q}) = -\frac{1}{3\theta(1-\theta)p}\boldsymbol{P}_{\boldsymbol{q}^\perp}\sum_{k=1}^{p}\left(\boldsymbol{q}^\top\boldsymbol{A}\boldsymbol{x}_k\right)^3\left(\boldsymbol{A}\boldsymbol{x}_k\right) \quad\longrightarrow\quad \operatorname{grad}\varphi_{\mathrm{T}}(\boldsymbol{q}) = -\boldsymbol{P}_{\boldsymbol{q}^\perp}\boldsymbol{A}\left(\boldsymbol{A}^\top\boldsymbol{q}\right)^{\odot 3},$$

where $\boldsymbol{x}_k$ follows i.i.d. $\mathcal{BG}(\theta)$ distribution in Assumption 2.2. Concretely, we have the following result.

**Proposition F.3 (Concentration of** $\operatorname{grad}\varphi_{\mathrm{DL}}(\cdot)$**)** *Suppose* $\boldsymbol{A}$ *satisfies Equation (F.9) and* $\boldsymbol{X} \in \mathbb{R}^{m\times p}$ *follows* $\mathcal{BG}(\theta)$ *with* $\theta \in \left(\frac{1}{m}, \frac{1}{2}\right)$. *For any given* $\delta \in \left(0, cK^2/(m\log^2 p\log^2 np)\right)$, *whenever*

$$p \;\geqslant\; C\delta^{-2}\theta K^5 n^2\log\left(\frac{\theta K n}{\delta}\right),$$

*we have*

$$\sup_{\boldsymbol{q}\in\mathbb{S}^{n-1}} \|\operatorname{grad}\varphi_{\mathrm{DL}}(\boldsymbol{q}) - \operatorname{grad}\varphi_{\mathrm{T}}(\boldsymbol{q})\| \;<\; \delta$$

*holds with probability at least* $1 - c'p^{-2}$. *Here,* $c, c', C > 0$ *are some numerical constants.*

**Proof** Since we have

$$\operatorname{grad}\varphi_{\mathrm{DL}}(\boldsymbol{q}) = -\frac{1}{3\theta(1-\theta)p}\boldsymbol{P}_{\boldsymbol{q}^\perp}\sum_{k=1}^{p}\left(\boldsymbol{q}^\top\boldsymbol{A}\boldsymbol{x}_k\right)^3\left(\boldsymbol{A}\boldsymbol{x}_k\right),$$

we invoke Corollary F.2 to show this result by letting

$$f_{\boldsymbol{q}}(\boldsymbol{x}) = -\frac{1}{3\theta(1-\theta)}\left(\boldsymbol{q}^\top\boldsymbol{A}\boldsymbol{x}\right)^3\boldsymbol{P}_{\boldsymbol{q}^\perp}\boldsymbol{A}\boldsymbol{x} \;\in\; \mathbb{R}^n, \tag{F.10}$$

where $\boldsymbol{x} \sim \mathcal{BG}(\theta)$ and we need to check the conditions in Equation (F.5), Equation (F.7), and Equation (F.8).

**Calculating subgaussian parameter $\sigma^2$ for $x$ and truncation.** Since each entry of $x$ follows $x_i \sim_{i.i.d.} \mathcal{BG}(\theta)$, its tail behavior is very similar and can be upper bounded by the tail of Gaussian, i.e.,

$$\mathbb{P}\left(|x_i| \geqslant t\right) \leqslant \exp\left(-t^2/2\right),$$

so that we choose the truncation level $B = 2\sqrt{\log(np)}$.

**Calculating $R_1$ and $R_2$ in Equation (F.7).** First, for each $i$ ($1 \leqslant i \leqslant p$), we have

$$\|f_{\boldsymbol{q}}(\overline{\boldsymbol{x}}_i)\| = \frac{1}{3\theta(1-\theta)}\left\|\left(\boldsymbol{q}^\top \boldsymbol{A}\overline{\boldsymbol{x}}_i\right)^3 \boldsymbol{P}_{\boldsymbol{q}^\perp} \boldsymbol{A}\overline{\boldsymbol{x}}_i\right\| \leqslant \frac{\|\boldsymbol{A}\overline{\boldsymbol{x}}_i\|^4}{3\theta(1-\theta)} \leqslant \frac{\|\boldsymbol{A}\|^4 \|\overline{\boldsymbol{x}}_i\|^4}{3\theta(1-\theta)} \leqslant \frac{K^2 \|\overline{\boldsymbol{x}}_i\|^4}{3\theta(1-\theta)}.$$

By Lemma A.9 and a union bound, we know that for any $1 \leqslant i \leqslant p$,

$$\|\boldsymbol{x}_i\|_0 \leqslant 4\theta m \log p, \qquad \|\overline{\boldsymbol{x}}_i\|_0 \leqslant 4\theta m \log p \implies \|\overline{\boldsymbol{x}}_i\|^2 \leqslant B^2 \|\overline{\boldsymbol{x}}_i\|_0 = 4B^2\theta m \log p \tag{F.11}$$

with probability at least $1 - p^{-2\theta m}$. Thus, by our truncation level, we have w.h.p.

$$\|f_{\boldsymbol{q}}(\overline{\boldsymbol{x}}_i)\| \leqslant \frac{6\theta}{(1-\theta)}K^2 B^4 m^2 \log^2 p = R_1.$$

On the other hand, by Lemma F.5, for the second moment we have

$$\mathbb{E}\left[\|f_{\boldsymbol{q}}(\overline{\boldsymbol{x}}_i)\|^2\right] \leqslant \mathbb{E}\left[\|f_{\boldsymbol{q}}(\boldsymbol{x}_i)\|^2\right] \leqslant c\theta K^4 m$$

for some constant $c > 0$. Thus, we obtain

$$R_1 = \frac{6\theta}{(1-\theta)}K^2 B^4 m^2 \log^2 p, \qquad R_2 = c\theta K^4 m. \tag{F.12}$$

**Calculating $\overline{L}_f$ in Equation (F.8).** Notice that for any $\boldsymbol{q}_1, \boldsymbol{q}_2 \in \mathbb{S}^{n-1}$, let $\boldsymbol{\zeta}_i = \boldsymbol{A}^\top \boldsymbol{q}_i$ ($i = 1, 2$), by Lemma F.4 we have

$$\begin{aligned}
\|f_{\boldsymbol{q}_1}(\overline{\boldsymbol{x}}) - f_{\boldsymbol{q}_2}(\overline{\boldsymbol{x}})\| &= \frac{1}{3\theta(1-\theta)}\left\|\left(\boldsymbol{\zeta}_1^\top \overline{\boldsymbol{x}}\right)^3 \boldsymbol{P}_{\boldsymbol{q}_1^\perp} \boldsymbol{A}\overline{\boldsymbol{x}} - \left(\boldsymbol{\zeta}_2^\top \overline{\boldsymbol{x}}\right)^3 \boldsymbol{P}_{\boldsymbol{q}_2^\perp} \boldsymbol{A}\overline{\boldsymbol{x}}\right\| \\
&\leqslant \frac{\|\boldsymbol{A}\| \|\overline{\boldsymbol{x}}\|}{3\theta(1-\theta)}\left\|\left(\boldsymbol{\zeta}_1^\top \overline{\boldsymbol{x}}\right)^3 \boldsymbol{P}_{\boldsymbol{q}_1^\perp} - \left(\boldsymbol{\zeta}_2^\top \overline{\boldsymbol{x}}\right)^3 \boldsymbol{P}_{\boldsymbol{q}_2^\perp}\right\| \\
&\leqslant \frac{\|\boldsymbol{A}\| \|\overline{\boldsymbol{x}}\|}{3\theta(1-\theta)}\left[\left|\boldsymbol{\zeta}_1^\top \overline{\boldsymbol{x}}\right|^3 \left\|\boldsymbol{P}_{\boldsymbol{q}_1^\perp} - \boldsymbol{P}_{\boldsymbol{q}_2^\perp}\right\| + \left|\left(\boldsymbol{\zeta}_1^\top \overline{\boldsymbol{x}}\right)^3 - \left(\boldsymbol{\zeta}_2^\top \overline{\boldsymbol{x}}\right)^3\right|\right] \\
&\leqslant \frac{\|\boldsymbol{A}\| \|\overline{\boldsymbol{x}}\|}{3\theta(1-\theta)}\left[2\|\boldsymbol{A}\|^3 \|\overline{\boldsymbol{x}}\|^3 \|\boldsymbol{q}_1 - \boldsymbol{q}_2\| + 3\|\boldsymbol{A}\|^3 \|\overline{\boldsymbol{x}}\|^3 \|\boldsymbol{q}_1 - \boldsymbol{q}_2\|\right] \\
&\leqslant \frac{2\|\boldsymbol{A}\|^4 \|\overline{\boldsymbol{x}}\|^4}{\theta(1-\theta)}\|\boldsymbol{q}_1 - \boldsymbol{q}_2\|.
\end{aligned}$$

where for the last two inequalities we used Lemma A.11 and

$$\begin{aligned}
\left|\left(\boldsymbol{\zeta}_1^\top \overline{\boldsymbol{x}}\right)^3 - \left(\boldsymbol{\zeta}_2^\top \overline{\boldsymbol{x}}\right)^3\right| &= \left|\left(\boldsymbol{\zeta}_1 - \boldsymbol{\zeta}_2\right)^\top \overline{\boldsymbol{x}}\right|\left|\left(\boldsymbol{\zeta}_1^\top \overline{\boldsymbol{x}}\right)^2 + \left(\boldsymbol{\zeta}_1^\top \overline{\boldsymbol{x}}\right)\left(\boldsymbol{\zeta}_2^\top \overline{\boldsymbol{x}}\right) + \left(\boldsymbol{\zeta}_2^\top \overline{\boldsymbol{x}}\right)^2\right| \\
&\leqslant \|\boldsymbol{A}\| \|\overline{\boldsymbol{x}}\| \|\boldsymbol{q}_1 - \boldsymbol{q}_2\|\left[\left(\boldsymbol{\zeta}_1^\top \overline{\boldsymbol{x}}\right)^2 + \left(\boldsymbol{\zeta}_2^\top \overline{\boldsymbol{x}}\right)^2 + \left|\boldsymbol{\zeta}_1^\top \overline{\boldsymbol{x}}\right|\left|\boldsymbol{\zeta}_2^\top \overline{\boldsymbol{x}}\right|\right] \\
&\leqslant 3\|\boldsymbol{A}\|^3 \|\overline{\boldsymbol{x}}\|^3 \|\boldsymbol{q}_1 - \boldsymbol{q}_2\|.
\end{aligned}$$

Furthermore, by Equation (F.11) we obtain

$$\|f_{\boldsymbol{q}_1}(\overline{\boldsymbol{x}}) - f_{\boldsymbol{q}_2}(\overline{\boldsymbol{x}})\| \leqslant \frac{2\|\boldsymbol{A}\|^4 \|\overline{\boldsymbol{x}}\|^4}{\theta(1-\theta)}\|\boldsymbol{q}_1 - \boldsymbol{q}_2\| \leqslant \frac{32\theta}{1-\theta}K^2 B^4 m^2 \log^2 p \|\boldsymbol{q}_1 - \boldsymbol{q}_2\|.$$

This gives

$$\overline{L}_f = \frac{32\theta}{1-\theta}K^2 B^4 m^2 \log^2 p. \tag{F.13}$$

**Calculating $B_f$ and $L_f$ in Equation (F.5).** From Lemma F.4 we know that $\mathbb{E}\left[f_{\boldsymbol{q}}(\boldsymbol{x})\right] = \boldsymbol{P}_{\boldsymbol{q}^\perp}\boldsymbol{A}\boldsymbol{\zeta}^{\odot 3}$, so that

$$
\begin{aligned}
\left\|\mathbb{E}\left[f_{\boldsymbol{q}}(\boldsymbol{x})\right]\right\| \;=\; \left\|\boldsymbol{P}_{\boldsymbol{q}^\perp}\boldsymbol{A}\left(\boldsymbol{A}^\top\boldsymbol{q}\right)^{\odot 3}\right\| \;&\leqslant\; \left\|\boldsymbol{P}_{\boldsymbol{q}^\perp}\right\|\left\|\boldsymbol{A}\right\|\left\|\boldsymbol{A}^\top\boldsymbol{q}\right\|_6^3 \\
&\leqslant\; \left\|\boldsymbol{A}\right\|\left\|\boldsymbol{A}^\top\boldsymbol{q}\right\|^3 \;\leqslant\; \left\|\boldsymbol{A}\right\|^4 \;=\; K^2 \;=\; B_f,
\end{aligned}
\tag{F.14}
$$

where we used Lemma A.1 for the second inequality. Moreover, we have

$$
\begin{aligned}
&\left\|\mathbb{E}\left[f_{\boldsymbol{q}_1}(\boldsymbol{x})\right] - \mathbb{E}\left[f_{\boldsymbol{q}_2}(\boldsymbol{x})\right]\right\| \\
&\leqslant\; \left\|\boldsymbol{P}_{\boldsymbol{q}_1^\perp}\boldsymbol{A}\boldsymbol{\zeta}_1^{\odot 3} - \boldsymbol{P}_{\boldsymbol{q}_1^\perp}\boldsymbol{A}\boldsymbol{\zeta}_2^{\odot 3}\right\| \;+\; \left\|\boldsymbol{P}_{\boldsymbol{q}_1^\perp}\boldsymbol{A}\boldsymbol{\zeta}_2^{\odot 3} - \boldsymbol{P}_{\boldsymbol{q}_2^\perp}\boldsymbol{A}\boldsymbol{\zeta}_2^{\odot 3}\right\| \\
&\leqslant\; \left\|\boldsymbol{A}\right\|\left\|\boldsymbol{\zeta}_1^{\odot 3} - \boldsymbol{\zeta}_2^{\odot 3}\right\| \;+\; \left\|\boldsymbol{P}_{\boldsymbol{q}_1^\perp} - \boldsymbol{P}_{\boldsymbol{q}_2^\perp}\right\|\left\|\boldsymbol{A}\right\|\left\|\boldsymbol{\zeta}_2^{\odot 3}\right\| \\
&\leqslant\; \left\|\boldsymbol{A}\right\|\left\|(\boldsymbol{\zeta}_1 - \boldsymbol{\zeta}_2)\odot\left(\boldsymbol{\zeta}_1^{\odot 2} + \boldsymbol{\zeta}_1\odot\boldsymbol{\zeta}_2 + \boldsymbol{\zeta}_1^{\odot 2}\right)\right\| \;+\; 2\left\|\boldsymbol{A}\right\|\left\|\boldsymbol{\zeta}_2\right\|^3\left\|\boldsymbol{q}_1 - \boldsymbol{q}_2\right\| \\
&\leqslant\; 5\left\|\boldsymbol{A}\right\|^4\left\|\boldsymbol{q}_1 - \boldsymbol{q}_2\right\| \;=\; 5K^2\left\|\boldsymbol{q}_1 - \boldsymbol{q}_2\right\| \;=\; L_f\left\|\boldsymbol{q}_1 - \boldsymbol{q}_2\right\|.
\end{aligned}
\tag{F.15}
$$

where for the last inequality, we used the fact that

$$
\begin{aligned}
\left\|(\boldsymbol{\zeta}_1 - \boldsymbol{\zeta}_2)\odot\left(\boldsymbol{\zeta}_1^{\odot 2} + \boldsymbol{\zeta}_1\odot\boldsymbol{\zeta}_2 + \boldsymbol{\zeta}_1^{\odot 2}\right)\right\| \;&\leqslant\; \left\|\boldsymbol{\zeta}_1 - \boldsymbol{\zeta}_2\right\|_4\left\|\boldsymbol{\zeta}_1^{\odot 2} + \boldsymbol{\zeta}_1\odot\boldsymbol{\zeta}_2 + \boldsymbol{\zeta}_1^{\odot 2}\right\|_4 \\
&\leqslant\; \left\|\boldsymbol{A}^\top\left(\boldsymbol{q}_1 - \boldsymbol{q}_2\right)\right\|\left(\left\|\boldsymbol{\zeta}_1^{\odot 2}\right\| + \left\|\boldsymbol{\zeta}_1\odot\boldsymbol{\zeta}_2\right\| + \left\|\boldsymbol{\zeta}_1^{\odot 2}\right\|\right) \\
&\leqslant\; 3\left\|\boldsymbol{A}\right\|^3\left\|\boldsymbol{q}_1 - \boldsymbol{q}_2\right\|.
\end{aligned}
$$

Thus, from Equation (F.14) and Equation (F.15), we obtain

$$
B_f \;=\; K^2, \qquad L_f \;=\; 5K^2.
\tag{F.16}
$$

**Final calculation.** Finally, we are now ready to put all the estimations in Equations (F.12), (F.13) and (F.16) together and apply Corollary F.2 to obtain our result. For any $\delta \in \left(0, 6\frac{R_2}{R_1}\right)$, whenever

$$
p \;\geqslant\; C\delta^{-2}\theta K^5 n^2 \log\left(\theta K n/\delta\right),
$$

we have

$$
\sup_{\boldsymbol{q}\in\mathbb{S}^{n-1}}\left\|\frac{1}{p}\sum_{i=1}^{p}f_{\boldsymbol{q}}(\boldsymbol{z}_i) - \mathbb{E}\left[f_{\boldsymbol{q}}(\boldsymbol{z})\right]\right\| \;\leqslant\; \delta,
$$

holding with probability at least $1 - (np)^{-2} - n^{-c_1\log(\theta K n/\delta)} - p^{-2\theta m}$ for some constant $c_1, C > 0$. ∎

**Lemma F.4 (Expectation of $\operatorname{grad}\varphi_{\mathrm{DL}}(\cdot)$)** $\forall\boldsymbol{q}\in\mathbb{S}^{n-1}$, *the expectation of* $\operatorname{grad}\varphi_{\mathrm{DL}}(\cdot)$ *satisfies*

$$
\operatorname{grad}\varphi_{\mathrm{DL}}(\boldsymbol{q}) = \operatorname{grad}\varphi_{\mathrm{T}}(\boldsymbol{q}) = -\boldsymbol{P}_{\boldsymbol{q}^\perp}\boldsymbol{A}\left(\boldsymbol{A}^\top\boldsymbol{q}\right)^{\odot 3}
$$

**Proof** Direct calculation. ∎

**Lemma F.5** *Suppose* $\boldsymbol{x}\sim\mathcal{BG}(\theta)$ *and let* $f_{\boldsymbol{q}}(\boldsymbol{x})$ *be defined as Equation (F.10), then we have*

$$
\mathbb{E}\left[\left\|f_{\boldsymbol{q}}(\boldsymbol{x})\right\|^2\right] \;\leqslant\; C\theta K^4 m \quad (K = m/n).
$$

**Proof** Since $\boldsymbol{x}\sim\mathcal{BG}(\theta)$, we write $\boldsymbol{x} = \boldsymbol{b}\odot\boldsymbol{g}$ with $\sim\operatorname{Ber}(\theta)$ and $\boldsymbol{g}\sim\mathcal{N}(\boldsymbol{0},\boldsymbol{I})$. Let $\mathcal{I}$ be the nonzero support of $\boldsymbol{x}$ with $\mathcal{I} = \operatorname{supp}\boldsymbol{x}$. And let $\mathcal{P}_{\mathcal{I}}(\cdot)$ be an operator that restricts a vector to the support $\mathcal{I}$, so that we can write $\boldsymbol{x} = \mathcal{P}_{\mathcal{I}}(\boldsymbol{g})$. Notice that

$$
\mathbb{E}\left\|f_{\boldsymbol{q}}(\boldsymbol{x})\right\|^2 = \mathbb{E}\left[\sum_{k=1}^{m}\left[f_{\boldsymbol{q}}^{\odot 2}(\boldsymbol{x})\right]_k\right] \leqslant m\max_{k\in[m]}\mathbb{E}\left[f_q^{\odot 2}(\boldsymbol{x})\right]_k.
$$

Let $\boldsymbol{W} = \boldsymbol{P}_{\boldsymbol{q}^\perp} \boldsymbol{A}$ with $\boldsymbol{w}_k$ being the $k^{\text{th}}$ row of $\boldsymbol{W}$. For $\forall k \in [n]$,

$$
\begin{aligned}
\left[\mathbb{E} f_{\boldsymbol{q}}^{\odot 2}(\boldsymbol{x})\right]_k &= \frac{1}{9\theta^2(1-\theta)^2} \mathbb{E}\left[\left(\boldsymbol{q}^\top \boldsymbol{A}\boldsymbol{x}\right)^6 \left(\sum_{i=1}^m w_{k,i} x_i\right)^2\right] \\
&\leqslant \frac{1}{9\theta^2(1-\theta)^2} \left(\mathbb{E}\left\langle \boldsymbol{A}^\top \boldsymbol{q}, \boldsymbol{x}\right\rangle^{12}\right)^{\frac{1}{2}} \left(\mathbb{E}\left\langle \boldsymbol{w}_k, \boldsymbol{x}\right\rangle^4\right)^{\frac{1}{2}} \\
&= \frac{1}{9\theta^2(1-\theta)^2} \left(\mathbb{E}\left\langle \mathcal{P}_{\mathcal{I}}\left(\boldsymbol{A}^\top \boldsymbol{q}\right), \boldsymbol{g}\right\rangle^{12}\right)^{\frac{1}{2}} \left(\mathbb{E}\left\langle \mathcal{P}_{\mathcal{I}}\left(\boldsymbol{w}_k\right), \boldsymbol{g}\right\rangle^4\right)^{\frac{1}{2}}.
\end{aligned}
$$

Notice that

$$
\left\langle \mathcal{P}_{\mathcal{I}}\left(\boldsymbol{A}^\top \boldsymbol{q}\right), \boldsymbol{v}\right\rangle \sim \mathcal{N}(0, \left\|\mathcal{P}_{\mathcal{I}}\left(\boldsymbol{A}^\top \boldsymbol{q}\right)\right\|^2) \quad \text{and} \quad \left\langle \mathcal{P}_{\mathcal{I}}\left(\boldsymbol{w}_k\right), \boldsymbol{v}\right\rangle \sim \mathcal{N}(0, \left\|\mathcal{P}_{\mathcal{I}}\left(\boldsymbol{w}_k\right)\right\|^2),
$$

hence

$$
\left(\mathbb{E}\left\langle \mathcal{P}_{\mathcal{I}}\left(\boldsymbol{A}^\top \boldsymbol{q}\right), \boldsymbol{v}\right\rangle^{12}\right)^{\frac{1}{2}} = \sqrt{11!!} \left(\mathbb{E}_{\mathcal{I}} \left\|\mathcal{P}_{\mathcal{I}}\left(\boldsymbol{A}^\top \boldsymbol{q}\right)\right\|^{12}\right)^{\frac{1}{2}}.
$$

Let $\boldsymbol{A}^\top \boldsymbol{q} = \boldsymbol{\zeta}$, then we have

$$
\mathbb{E}_{\mathcal{I}} \left\|\mathcal{P}_{\mathcal{I}}\left(\boldsymbol{A}^\top \boldsymbol{q}\right)\right\|^{12} = \sum_{k_1,k_2,\ldots,k_6} m_{k_1}^2 \mathbb{1}_{k_1 \in \mathcal{I}} \zeta_{k_2}^2 \mathbb{1}_{k_2 \in \mathcal{I}} \zeta_{k_3}^2 \mathbb{1}_{k_3 \in \mathcal{I}} \zeta_{k_4}^2 \mathbb{1}_{k_4 \in \mathcal{I}} \zeta_{k_5}^2 \mathbb{1}_{k_5 \in \mathcal{I}} \zeta_{k_6}^2 \mathbb{1}_{k_6 \in \mathcal{I}}, \quad \text{(F.17)}
$$

for bounding equation F.17, we discuss the following cases:

- When only one index among $k_1, k_2, \ldots, k_6$ is in $\mathcal{I}$:

$$
\mathbb{E}_{\mathcal{I}} \left\|\mathcal{P}_{\mathcal{I}}\left(\boldsymbol{A}^\top \boldsymbol{q}\right)\right\|^{12} = \theta \sum_{k_1} \zeta_{k_1}^{12} \leqslant \theta K^6
$$

- When only two indices among $k_1, k_2, \ldots, k_6$ are in $\mathcal{I}$:

$$
\mathbb{E}_{\mathcal{I}} \left\|\mathcal{P}_{\mathcal{I}}\left(\boldsymbol{A}^\top \boldsymbol{q}\right)\right\|^{12} = \theta^2 \sum_{k_1,k_2} \left(\zeta_{k_1}^2 \zeta_{k_2}^{10} + \zeta_{k_1}^4 \zeta_{k_2}^8 + \zeta_{k_1}^6 \zeta_{k_2}^6\right) \leqslant 3\theta^2 K^6
$$

- When only three indices among $k_1, k_2, \ldots, k_6$ are in $\mathcal{I}$:

$$
\mathbb{E}_{\mathcal{I}} \left\|\mathcal{P}_{\mathcal{I}}\left(\boldsymbol{A}^\top \boldsymbol{q}\right)\right\|^{12} = \theta^3 \sum_{k_1,k_2,k_3} \left(\zeta_{k_1}^2 \zeta_{k_2}^2 \zeta_{k_3}^8 + \zeta_{k_1}^2 \zeta_{k_2}^4 \zeta_{k_3}^6 + \zeta_{k_1}^4 \zeta_{k_2}^4 \zeta_{k_3}^4\right) \leqslant 3\theta^3 K^6
$$

- When only four indices among $k_1, k_2, \ldots, k_6$ are in $\mathcal{I}$:

$$
\mathbb{E}_{\mathcal{I}} \left\|\mathcal{P}_{\mathcal{I}}\left(\boldsymbol{A}^\top \boldsymbol{q}\right)\right\|^{12} = \theta^4 \sum_{k_1,k_2,k_3,k_4} \left(\zeta_{k_1}^2 \zeta_{k_2}^2 \zeta_{k_3}^2 \zeta_{k_4}^6 + \zeta_{k_1}^2 \zeta_{k_2}^2 \zeta_{k_3}^4 \zeta_{k_4}^4\right) \leqslant 2\theta^4 K^6
$$

- When only five indices among $k_1, k_2, \ldots, k_6$ are in $\mathcal{I}$:

$$
\mathbb{E}_{\mathcal{I}} \left\|\mathcal{P}_{\mathcal{I}}\left(\boldsymbol{A}^\top \boldsymbol{q}\right)\right\|^{12} = \theta^5 \sum_{k_1,k_2,k_3,k_4,k_5} \left(\zeta_{k_1}^2 \zeta_{k_2}^2 \zeta_{k_3}^2 \zeta_{k_4}^2 \zeta_{k_5}^4\right) \leqslant \theta^5 K^6
$$

- When all six indices of $k_1, k_2, \ldots, k_6$ are in $\mathcal{I}$:

$$
\mathbb{E}_{\mathcal{I}} \left\|\mathcal{P}_{\mathcal{I}}\left(\boldsymbol{A}^\top \boldsymbol{q}\right)\right\|^{12} = \theta^6 \sum_{k_1,k_2,k_3,k_4,k_5,k_6} \left(\zeta_{k_1}^2 \zeta_{k_2}^2 \zeta_{k_3}^2 \zeta_{k_4}^2 \zeta_{k_5}^2 \zeta_{k_6}^2\right) \leqslant \theta^6 K^6.
$$

Hence, we have

$$
\mathbb{E}_{\mathcal{I}} \left\|\mathcal{P}_{\mathcal{I}}\left(\boldsymbol{A}^\top \boldsymbol{q}\right)\right\|^{12} = \theta K^6 + 3\theta^2 K^6 + 3\theta^3 K^6 + 2\theta^4 K^6 + \theta^5 K^6 + \theta^6 K^6 \leqslant C_1 \theta K^6
$$

for a constant $C_1 > 11$. Similarly, we have

$$
\left(\mathbb{E}\left\langle \mathcal{P}_{\mathcal{I}}\left(\boldsymbol{w}_k\right), \boldsymbol{v}\right\rangle^4\right)^{\frac{1}{2}} = \sqrt{3} \left(\mathbb{E}_{\mathcal{I}} \left\|\mathcal{P}_{\mathcal{I}}\left(\boldsymbol{w}_k\right)\right\|^4\right)^{\frac{1}{2}},
$$

and

$$\mathbb{E}_{\mathcal{I}} \|\mathcal{P}_{\mathcal{I}}(\boldsymbol{w}_k)\|^4 = \sum_{k_1,k_2} w_{k,k_1}^2 \mathbb{1}_{k_1 \in \mathcal{I}} w_{k,k_2}^2 \mathbb{1}_{k_2 \in \mathcal{I}} \leqslant C_2 \theta \frac{m^2}{n^2},$$

for a constant $C_2 > 2$. Hence, we have

$$\left(\mathbb{E}\langle \mathcal{P}_{\mathcal{I}}(\boldsymbol{A}^\top \boldsymbol{q}), \boldsymbol{g}\rangle^{12}\right)^{\frac{1}{2}} \left(\mathbb{E}\langle \mathcal{P}_{\mathcal{I}} \boldsymbol{w}_k, \boldsymbol{g}\rangle^4\right)^{\frac{1}{2}} \leqslant C_3 \theta \frac{m^4}{n^4},$$

for a constant $C_3 > 829$. Hence, we know that $\forall k \in [n]$,

$$\left[\mathbb{E} f_{\boldsymbol{q}}^{\odot 2}(\boldsymbol{x})\right]_k \leqslant \frac{C_4}{\theta(1-\theta)^2} \frac{m^4}{n^4} = C\theta K^4,$$

for a constant $C_4 > 93$. Therefore

$$\mathbb{E} \|f_{\boldsymbol{q}}(\boldsymbol{x})\|^2 \leqslant C\theta K^4 m,$$

for a constant $C > \frac{93}{\theta^2(1-\theta)^2}$. ∎

### F.2.2 CONCENTRATION OF $\operatorname{Hess} \varphi_{\mathrm{DL}}(\cdot)$

**Proposition F.6 (Concentration of $\operatorname{Hess} \varphi_{\mathrm{DL}}(\cdot)$)** *Suppose $\boldsymbol{A}$ satisfies Equation (F.9) and $\boldsymbol{X} \in \mathbb{R}^{m \times p}$ follows $\mathcal{BG}(\theta)$ with $\theta \in \left(\frac{1}{m}, \frac{1}{2}\right)$. For any given $\delta \in \left(0, cK^2/(\log^2 p \log^2 np)\right)$, whenever*

$$p \geqslant C\delta^{-2}\theta K^6 n^3 \log(\theta K n/\delta),$$

*we have*

$$\sup_{\boldsymbol{q} \in \mathbb{S}^{n-1}} \|\operatorname{Hess} \varphi_{\mathrm{DL}}(\boldsymbol{q}) - \operatorname{Hess} \varphi_{\mathrm{T}}(\boldsymbol{q})\| < \delta$$

*holds with probability at least $1 - c'p^{-2}$. Here, $c, c', C > 0$ are some numerical constants.*

**Proof** Since we have

$$\operatorname{Hess} \varphi_{\mathrm{DL}}(\boldsymbol{q}) = -\frac{1}{3\theta(1-\theta)p} \sum_{k=1}^{p} \boldsymbol{P}_{\boldsymbol{q}^\perp} \left[3\left(\boldsymbol{q}^\top \boldsymbol{A} \boldsymbol{x}_k\right)^2 \boldsymbol{A} \boldsymbol{x}_k \left(\boldsymbol{A} \boldsymbol{x}_k\right)^\top - \left(\boldsymbol{q}^\top \boldsymbol{A} \boldsymbol{x}_k\right)^4 \boldsymbol{I}\right] \boldsymbol{P}_{\boldsymbol{q}^\perp},$$

we invoke Theorem F.1 to show our result by letting

$$f_{\boldsymbol{q}}(\boldsymbol{x}) = -\frac{1}{3\theta(1-\theta)} \boldsymbol{P}_{\boldsymbol{q}^\perp} \left[3\left(\boldsymbol{q}^\top \boldsymbol{A} \boldsymbol{x}\right)^2 \boldsymbol{A} \boldsymbol{x} \left(\boldsymbol{A} \boldsymbol{x}\right)^\top - \left(\boldsymbol{q}^\top \boldsymbol{A} \boldsymbol{x}\right)^4 \boldsymbol{I}\right] \boldsymbol{P}_{\boldsymbol{q}^\perp} \in \mathbb{R}^{n \times n}, \quad \text{(F.18)}$$

where $\boldsymbol{x} \sim \mathcal{BG}(\theta)$ and we need to check the conditions in Equation (F.1), Equation (F.3), and Equation (F.4).

**Calculating subgaussian parameter $\sigma^2$ for $x$ and truncation.** Since each entry of $\boldsymbol{x}$ follows $x_i \sim_{i.i.d.} \mathcal{BG}(\theta)$, its tail behavior is very similar and can be upper bounded by the tail of Gaussian, i.e.,

$$\mathbb{P}(|x_i| \geqslant t) \leqslant \exp\left(-t^2/2\right),$$

so that we choose the truncation level $B = 2\sqrt{\log(np)}$. By Lemma A.9 and a union bound, we know that for any $1 \leqslant i \leqslant p$,

$$\|\boldsymbol{x}_i\|_0 \leqslant 4\theta m \log p, \qquad \|\overline{\boldsymbol{x}}_i\|_0 \leqslant 4\theta m \log p \implies \|\overline{\boldsymbol{x}}_i\|^2 \leqslant B^2 \|\overline{\boldsymbol{x}}_i\|_0 = 4B^2 \theta m \log p \tag{F.19}$$

with probability at least $1 - p^{-2\theta m}$.

**Calculating $R_1$ and $R_2$ in Equation (F.3).** For simplicity, let $\overline{\boldsymbol{\xi}} = \boldsymbol{A}\overline{\boldsymbol{x}}$. First of all, we have

$$
\begin{aligned}
\left\| f_{\boldsymbol{q}}(\overline{\boldsymbol{x}}) \right\| &= \frac{1}{3\theta(1-\theta)} \left\| \boldsymbol{P}_{\boldsymbol{q}^\perp} \left[ 3 \left( \boldsymbol{q}^\top \overline{\boldsymbol{\xi}} \right)^2 \overline{\boldsymbol{\xi}}\,\overline{\boldsymbol{\xi}}^\top - \left( \boldsymbol{q}^\top \overline{\boldsymbol{\xi}} \right)^4 \boldsymbol{I} \right] \boldsymbol{P}_{\boldsymbol{q}^\perp} \right\| \\
&\leqslant \frac{1}{3\theta(1-\theta)} \left( \boldsymbol{q}^\top \overline{\boldsymbol{\xi}} \right)^2 \left\| 3 \overline{\boldsymbol{\xi}}\,\overline{\boldsymbol{\xi}}^\top - \left( \boldsymbol{q}^\top \overline{\boldsymbol{\xi}} \right)^2 \boldsymbol{I} \right\| \\
&\leqslant \frac{4}{3\theta(1-\theta)} \left\| \overline{\boldsymbol{\xi}} \right\|^4 \leqslant \frac{4}{3\theta(1-\theta)} \left\| \boldsymbol{A} \right\|^4 \left\| \overline{\boldsymbol{x}} \right\|^4 \leqslant \frac{64 B^4}{3(1-\theta)} \theta K^2 m^2 \log^2 p.
\end{aligned}
$$

On the other hand, by Lemma F.7, we have

$$
\left\| \mathbb{E} \left[ f_{\boldsymbol{q}}(\overline{\boldsymbol{x}}) f_{\boldsymbol{q}}(\overline{\boldsymbol{x}})^\top \right] \right\| = \left\| \mathbb{E} \left[ f_{\boldsymbol{q}}(\overline{\boldsymbol{x}})^\top f_{\boldsymbol{q}}(\overline{\boldsymbol{x}}) \right] \right\| \leqslant \left\| \mathbb{E} \left[ f_{\boldsymbol{q}}(\boldsymbol{x})^\top f_{\boldsymbol{q}}(\boldsymbol{x}) \right] \right\| \leqslant c_1 \theta K^4 m^2,
$$

for some numerical constant $c_1 > 0$. In summary, we obtain

$$
R_1 = \frac{64 B^4}{3(1-\theta)} \theta K^2 m^2 \log^2 p, \qquad R_2 = c_1 K^4 \theta m^2. \tag{F.20}
$$

**Calculating $\overline{L}_f$ in Equation (F.4).** For any $\boldsymbol{q}_1,\ \boldsymbol{q}_2 \in \mathbb{S}^{n-1}$, we have

$$
\begin{aligned}
&\left\| f_{\boldsymbol{q}_1}(\overline{\boldsymbol{x}}) - f_{\boldsymbol{q}_2}(\overline{\boldsymbol{x}}) \right\| \\
&= \frac{1}{3\theta(1-\theta)} \left\| \boldsymbol{P}_{\boldsymbol{q}_1^\perp} \left[ 3 \left( \boldsymbol{q}_1^\top \overline{\boldsymbol{\xi}} \right)^2 \overline{\boldsymbol{\xi}}\,\overline{\boldsymbol{\xi}}^\top - \left( \boldsymbol{q}_1^\top \overline{\boldsymbol{\xi}} \right)^4 \boldsymbol{I} \right] \boldsymbol{P}_{\boldsymbol{q}_1^\perp} - \boldsymbol{P}_{\boldsymbol{q}_2^\perp} \left[ 3 \left( \boldsymbol{q}_2^\top \overline{\boldsymbol{\xi}} \right)^2 \overline{\boldsymbol{\xi}}\,\overline{\boldsymbol{\xi}}^\top - \left( \boldsymbol{q}_2^\top \overline{\boldsymbol{\xi}} \right)^4 \boldsymbol{I} \right] \boldsymbol{P}_{\boldsymbol{q}_2^\perp} \right\| \\
&\leqslant \frac{1}{\theta(1-\theta)} \underbrace{\left\| \boldsymbol{P}_{\boldsymbol{q}_1^\perp} \left( \boldsymbol{q}_1^\top \overline{\boldsymbol{\xi}} \right)^2 \overline{\boldsymbol{\xi}}\,\overline{\boldsymbol{\xi}}^\top \boldsymbol{P}_{\boldsymbol{q}_1^\perp} - \boldsymbol{P}_{\boldsymbol{q}_2^\perp} \left( \boldsymbol{q}_2^\top \overline{\boldsymbol{\xi}} \right)^2 \overline{\boldsymbol{\xi}}\,\overline{\boldsymbol{\xi}}^\top \boldsymbol{P}_{\boldsymbol{q}_2^\perp} \right\|}_{\mathcal{T}_1} + \frac{1}{3\theta(1-\theta)} \underbrace{\left\| \left( \boldsymbol{q}_1^\top \overline{\boldsymbol{\xi}} \right)^4 \boldsymbol{P}_{\boldsymbol{q}_1^\perp} - \left( \boldsymbol{q}_2^\top \overline{\boldsymbol{\xi}} \right)^4 \boldsymbol{P}_{\boldsymbol{q}_2^\perp} \right\|}_{\mathcal{T}_2},
\end{aligned}
$$

where by Lemma A.11, we have

$$
\begin{aligned}
\mathcal{T}_1 &\leqslant \left\| \boldsymbol{P}_{\boldsymbol{q}_1^\perp} \left( \boldsymbol{q}_1^\top \overline{\boldsymbol{\xi}} \right)^2 \overline{\boldsymbol{\xi}}\,\overline{\boldsymbol{\xi}}^\top \boldsymbol{P}_{\boldsymbol{q}_1^\perp} - \boldsymbol{P}_{\boldsymbol{q}_1^\perp} \left( \boldsymbol{q}_1^\top \overline{\boldsymbol{\xi}} \right)^2 \overline{\boldsymbol{\xi}}\,\overline{\boldsymbol{\xi}}^\top \boldsymbol{P}_{\boldsymbol{q}_2^\perp} \right\| + \left\| \boldsymbol{P}_{\boldsymbol{q}_1^\perp} \left( \boldsymbol{q}_1^\top \overline{\boldsymbol{\xi}} \right)^2 \overline{\boldsymbol{\xi}}\,\overline{\boldsymbol{\xi}}^\top \boldsymbol{P}_{\boldsymbol{q}_2^\perp} - \boldsymbol{P}_{\boldsymbol{q}_2^\perp} \left( \boldsymbol{q}_2^\top \overline{\boldsymbol{\xi}} \right)^2 \overline{\boldsymbol{\xi}}\,\overline{\boldsymbol{\xi}}^\top \boldsymbol{P}_{\boldsymbol{q}_2^\perp} \right\| \\
&\leqslant \left\| \overline{\boldsymbol{\xi}} \right\|^4 \left\| \boldsymbol{P}_{\boldsymbol{q}_1^\perp} - \boldsymbol{P}_{\boldsymbol{q}_2^\perp} \right\| + \left\| \boldsymbol{P}_{\boldsymbol{q}_1^\perp} \left( \boldsymbol{q}_1^\top \overline{\boldsymbol{\xi}} \right)^2 \overline{\boldsymbol{\xi}}\,\overline{\boldsymbol{\xi}}^\top - \boldsymbol{P}_{\boldsymbol{q}_1^\perp} \left( \boldsymbol{q}_2^\top \overline{\boldsymbol{\xi}} \right)^2 \overline{\boldsymbol{\xi}}\,\overline{\boldsymbol{\xi}}^\top \right\| + \left\| \boldsymbol{P}_{\boldsymbol{q}_1^\perp} \left( \boldsymbol{q}_2^\top \overline{\boldsymbol{\xi}} \right)^2 \overline{\boldsymbol{\xi}}\,\overline{\boldsymbol{\xi}}^\top - \boldsymbol{P}_{\boldsymbol{q}_2^\perp} \left( \boldsymbol{q}_2^\top \overline{\boldsymbol{\xi}} \right)^2 \overline{\boldsymbol{\xi}}\,\overline{\boldsymbol{\xi}}^\top \right\| \\
&\leqslant \left\| \overline{\boldsymbol{\xi}} \right\|^4 \left\| \boldsymbol{P}_{\boldsymbol{q}_1^\perp} - \boldsymbol{P}_{\boldsymbol{q}_2^\perp} \right\| + \left\| \overline{\boldsymbol{\xi}} \right\|^2 \left( \boldsymbol{q}_1^\top \overline{\boldsymbol{\xi}} + \boldsymbol{q}_2^\top \overline{\boldsymbol{\xi}} \right) \left( \boldsymbol{q}_1^\top \overline{\boldsymbol{\xi}} - \boldsymbol{q}_2^\top \overline{\boldsymbol{\xi}} \right) \\
&\leqslant 4 \left\| \overline{\boldsymbol{\xi}} \right\|^4 \left\| \boldsymbol{q}_1 - \boldsymbol{q}_2 \right\| \leqslant 4 \left\| \boldsymbol{A} \right\|^4 \left\| \overline{\boldsymbol{x}} \right\|^4 \left\| \boldsymbol{q}_1 - \boldsymbol{q}_2 \right\| \leqslant 64 K^2 B^4 \theta^2 m^2 \log^2 p \left\| \boldsymbol{q}_1 - \boldsymbol{q}_2 \right\|,
\end{aligned}
$$

and

$$
\begin{aligned}
\mathcal{T}_2 &\leqslant \left\| \left( \boldsymbol{q}_1^\top \overline{\boldsymbol{\xi}} \right)^4 \boldsymbol{P}_{\boldsymbol{q}_1^\perp} - \left( \boldsymbol{q}_2^\top \overline{\boldsymbol{\xi}} \right)^4 \boldsymbol{P}_{\boldsymbol{q}_1^\perp} \right\| + \left\| \left( \boldsymbol{q}_2^\top \overline{\boldsymbol{\xi}} \right)^4 \boldsymbol{P}_{\boldsymbol{q}_1^\perp} - \left( \boldsymbol{q}_2^\top \overline{\boldsymbol{\xi}} \right)^4 \boldsymbol{P}_{\boldsymbol{q}_2^\perp} \right\| \\
&\leqslant \left( \left( \boldsymbol{q}_1^\top \overline{\boldsymbol{\xi}} \right)^2 + \left( \boldsymbol{q}_2^\top \overline{\boldsymbol{\xi}} \right)^2 \right) \left( \boldsymbol{q}_1 + \boldsymbol{q}_2 \right)^\top \overline{\boldsymbol{\xi}}\,\overline{\boldsymbol{\xi}}^\top \left( \boldsymbol{q}_1 - \boldsymbol{q}_2 \right) + 2 \left\| \overline{\boldsymbol{\xi}} \right\|^4 \left\| \boldsymbol{q}_1 - \boldsymbol{q}_2 \right\| \\
&\leqslant 6 \left\| \overline{\boldsymbol{\xi}} \right\|^4 \left\| \boldsymbol{q}_1 - \boldsymbol{q}_2 \right\| \leqslant 6 \left\| \boldsymbol{A} \right\|^4 \left\| \overline{\boldsymbol{x}} \right\|^4 \left\| \boldsymbol{q}_1 - \boldsymbol{q}_2 \right\| \leqslant 96 K^2 B^4 \theta^2 m^2 \log^2 p \left\| \boldsymbol{q}_1 - \boldsymbol{q}_2 \right\|,
\end{aligned}
$$

where for the last inequality we used Equation (F.19). Therefore, we have

$$
\left\| f_{\boldsymbol{q}_1}(\overline{\boldsymbol{x}}) - f_{\boldsymbol{q}_2}(\overline{\boldsymbol{x}}) \right\| \leqslant \frac{96\theta}{1-\theta} K^2 B^4 m^2 \log^2 p \left\| \boldsymbol{q}_1 - \boldsymbol{q}_2 \right\|,
$$

so that

$$
\overline{L}_f = \frac{96\theta}{1-\theta} K^2 B^4 m^2 \log^2 p. \tag{F.21}
$$

**Calculating $B_f$ and $L_f$ in Equation (F.1).** We have

$$
\begin{aligned}
\left\| \mathbb{E} \left[ f_{\boldsymbol{q}}(\boldsymbol{x}) \right] \right\| &= \left\| \boldsymbol{P}_{\boldsymbol{q}^\perp} \left[ 3 \boldsymbol{A} \operatorname{diag} \left( \boldsymbol{\zeta}^{\odot 2} \right) \boldsymbol{A}^\top - \left\| \boldsymbol{\zeta} \right\|_4^4 \boldsymbol{I} \right] \boldsymbol{P}_{\boldsymbol{q}^\perp} \right\| \\
&\leqslant \left\| 3 \boldsymbol{A} \operatorname{diag} \left( \boldsymbol{\zeta}^{\odot 2} \right) \boldsymbol{A}^\top - \left\| \boldsymbol{\zeta} \right\|_4^4 \boldsymbol{I} \right\| \\
&\leqslant 3 \left\| \boldsymbol{A} \right\|^2 \left\| \boldsymbol{A} \right\|_{\ell^1 \to \ell^2}^2 + \left\| \boldsymbol{A} \right\|^4 \leqslant K \left( 3 M^2 + K \right),
\end{aligned}
$$

where $\|\boldsymbol{A}\|_{\ell^1 \to \ell^2} = \max_{1 \leqslant k \leqslant m} \|\boldsymbol{a}_k\| \leqslant M$. On the other hand, for any $\boldsymbol{q}_1, \ \boldsymbol{q}_2 \in \mathbb{S}^{n-1}$, we have

$$\|\mathbb{E}\left[f_{\boldsymbol{q}_1}(\boldsymbol{x})\right] - \mathbb{E}\left[f_{\boldsymbol{q}_2}(\boldsymbol{x})\right]\|$$

$$= \left\| \boldsymbol{P}_{\boldsymbol{q}_1^\perp} \left[3\boldsymbol{A}\operatorname{diag}\left(\boldsymbol{\zeta}_1^{\odot 2}\right)\boldsymbol{A}^\top - \|\boldsymbol{\zeta}_1\|_4^4 \boldsymbol{I}\right]\boldsymbol{P}_{\boldsymbol{q}_1^\perp} - \boldsymbol{P}_{\boldsymbol{q}_2^\perp} \left[3\boldsymbol{A}\operatorname{diag}\left(\boldsymbol{\zeta}_2^{\odot 2}\right)\boldsymbol{A}^\top - \|\boldsymbol{\zeta}_2\|_4^4 \boldsymbol{I}\right]\boldsymbol{P}_{\boldsymbol{q}_2^\perp} \right\|$$

$$\leqslant 3\underbrace{\left\| \boldsymbol{P}_{\boldsymbol{q}_1^\perp} \boldsymbol{A}\operatorname{diag}\left(\boldsymbol{\zeta}_1^{\odot 2}\right)\boldsymbol{A}^\top\boldsymbol{P}_{\boldsymbol{q}_1^\perp} - \boldsymbol{P}_{\boldsymbol{q}_2^\perp}\boldsymbol{A}\operatorname{diag}\left(\boldsymbol{\zeta}_2^{\odot 2}\right)\boldsymbol{A}^\top\boldsymbol{P}_{\boldsymbol{q}_2^\perp}\right\|}_{\mathcal{L}_1} + \underbrace{\left\|\|\boldsymbol{\zeta}_1\|_4^4\boldsymbol{P}_{\boldsymbol{q}_1^\perp} - \|\boldsymbol{\zeta}_2\|_4^4\boldsymbol{P}_{\boldsymbol{q}_2^\perp}\right\|}_{\mathcal{L}_2}.$$

By direct calculation, we have

$$\mathcal{L}_1 \leqslant \left\| \boldsymbol{P}_{\boldsymbol{q}_1^\perp}\boldsymbol{A}\operatorname{diag}\left(\boldsymbol{\zeta}_1^{\odot 2}\right)\boldsymbol{A}^\top\boldsymbol{P}_{\boldsymbol{q}_1^\perp} - \boldsymbol{P}_{\boldsymbol{q}_2^\perp}\boldsymbol{A}\operatorname{diag}\left(\boldsymbol{\zeta}_2^{\odot 2}\right)\boldsymbol{A}^\top\boldsymbol{P}_{\boldsymbol{q}_2^\perp}\right\|$$

$$\leqslant \left\|\boldsymbol{P}_{\boldsymbol{q}_1^\perp}\boldsymbol{A}\operatorname{diag}\left(\boldsymbol{\zeta}_1^{\odot 2}\right)\boldsymbol{A}^\top\left(\boldsymbol{P}_{\boldsymbol{q}_1^\perp} - \boldsymbol{P}_{\boldsymbol{q}_2^\perp}\right)\right\| + \left\|\left[\boldsymbol{P}_{\boldsymbol{q}_1^\perp}\boldsymbol{A}\operatorname{diag}\left(\boldsymbol{\zeta}_1^{\odot 2}\right) - \boldsymbol{P}_{\boldsymbol{q}_2^\perp}\boldsymbol{A}\operatorname{diag}\left(\boldsymbol{\zeta}_2^{\odot 2}\right)\right]\boldsymbol{A}^\top\boldsymbol{P}_{\boldsymbol{q}_2^\perp}\right\|$$

$$\leqslant \|\boldsymbol{A}\|^2 \|\boldsymbol{\zeta}_1\|_\infty^2 \left\|\boldsymbol{P}_{\boldsymbol{q}_1^\perp} - \boldsymbol{P}_{\boldsymbol{q}_2^\perp}\right\| + \|\boldsymbol{A}\|\left(\left\|\left(\boldsymbol{P}_{\boldsymbol{q}_1^\perp} - \boldsymbol{P}_{\boldsymbol{q}_2^\perp}\right)\boldsymbol{A}\operatorname{diag}\left(\boldsymbol{\zeta}_1^{\odot 2}\right)\right\| + \left\|\boldsymbol{P}_{\boldsymbol{q}_2^\perp}\boldsymbol{A}\operatorname{diag}\left(\boldsymbol{\zeta}_1^{\odot 2} - \boldsymbol{\zeta}_2^{\odot 2}\right)\right\|\right)$$

$$\leqslant 2\|\boldsymbol{A}\|^2\|\boldsymbol{\zeta}_1\|_\infty^2\|\boldsymbol{q}_1 - \boldsymbol{q}_2\| + 2\|\boldsymbol{A}\|^2\|\boldsymbol{\zeta}_1\|_\infty^2\|\boldsymbol{q}_1 - \boldsymbol{q}_2\| + \|\boldsymbol{A}\|^2\|\boldsymbol{\zeta}_1 + \boldsymbol{\zeta}_2\|_\infty\|\boldsymbol{\zeta}_1 - \boldsymbol{\zeta}_2\|_\infty$$

$$\leqslant 6\|\boldsymbol{A}\|^2\|\boldsymbol{A}\|_{\ell^1 \to \ell^2}^2\|\boldsymbol{q}_1 - \boldsymbol{q}_2\| \leqslant 6KM^2\|\boldsymbol{q}_1 - \boldsymbol{q}_2\|,$$

and

$$\mathcal{L}_2 \leqslant \|\boldsymbol{\zeta}_1\|_4^4\left\|\boldsymbol{P}_{\boldsymbol{q}_1^\perp} - \boldsymbol{P}_{\boldsymbol{q}_2^\perp}\right\| + \left|\|\boldsymbol{\zeta}_1\|_4^4 - \|\boldsymbol{\zeta}_2\|_4^4\right|\left\|\boldsymbol{P}_{\boldsymbol{q}_2^\perp}\right\|$$

$$\leqslant 2\|\boldsymbol{A}\|^4\|\boldsymbol{q}_1 - \boldsymbol{q}_2\| + \left|\|\boldsymbol{\zeta}_1\|_4 - \|\boldsymbol{\zeta}_2\|_4\right|\left(\|\boldsymbol{\zeta}_1\|_4 + \|\boldsymbol{\zeta}_2\|_4\right)\left(\|\boldsymbol{\zeta}_1\|_4^2 + \|\boldsymbol{\zeta}_2\|_4^2\right)$$

$$\leqslant 2\|\boldsymbol{A}\|^4\|\boldsymbol{q}_1 - \boldsymbol{q}_2\| + \|\boldsymbol{\zeta}_1 - \boldsymbol{\zeta}_2\|\left(\|\boldsymbol{\zeta}_1\| + \|\boldsymbol{\zeta}_2\|\right)\left(\|\boldsymbol{\zeta}_1\|^2 + \|\boldsymbol{\zeta}_2\|^2\right)$$

$$\leqslant 6\|\boldsymbol{A}\|^4\|\boldsymbol{q}_1 - \boldsymbol{q}_2\| = 6K^2\|\boldsymbol{q}_1 - \boldsymbol{q}_2\|.$$

These together give us

$$\|\mathbb{E}\left[f_{\boldsymbol{q}_1}(\boldsymbol{x})\right] - \mathbb{E}\left[f_{\boldsymbol{q}_2}(\boldsymbol{x})\right]\| \leqslant 6K\left(K + M^2\right)\|\boldsymbol{q}_1 - \boldsymbol{q}_2\|.$$

Summarizing everything together, we have

$$B_f = K\left(3M^2 + K\right), \qquad L_f = 6K\left(K + M^2\right). \tag{F.22}$$

**Final calculation.** Finally, we are now ready to put all the estimations in Equations (F.20) to (F.22) together and apply Theorem F.1 to obtain our result. For any $\delta \in \left(0, 6\frac{R_2}{R_1}\right)$, whenever

$$p \geqslant C\delta^{-2}\theta K^6 n^3 \log\left(\theta K n/\delta\right),$$

we have

$$\sup_{\boldsymbol{q} \in \mathbb{S}^{n-1}} \left\|\frac{1}{p}\sum_{i=1}^p f_{\boldsymbol{q}}(\boldsymbol{z}_i) - \mathbb{E}\left[f_{\boldsymbol{q}}(\boldsymbol{z})\right]\right\| \leqslant \delta,$$

holding with probability at least $1 - (np)^{-2} - n^{-c_1 \log(\theta K n/\delta)} - p^{-2\theta m}$ for some constant $c_1, C > 0$.
∎

**Lemma F.7** *Suppose* $\theta \in \left(\frac{1}{m}, \frac{1}{2}\right)$. *Let* $f_{\boldsymbol{q}}(\boldsymbol{x})$ *be defined as in* Equation (F.18). *We have*

$$\left\|\mathbb{E}\left[f_{\boldsymbol{q}}(\boldsymbol{x})^\top f_{\boldsymbol{q}}(\boldsymbol{x})\right]\right\| \leqslant CK^4\theta m^2$$

*for some numerical constant* $C > 0$.

**Proof** Let $\boldsymbol{x} = \boldsymbol{b} \odot \boldsymbol{g}$ with $\boldsymbol{b} \sim \operatorname{Ber}(\theta)$ and $\boldsymbol{g} \sim \mathcal{N}(\boldsymbol{0}, \boldsymbol{I})$. First, let $\boldsymbol{\xi} = \boldsymbol{A}\boldsymbol{x}$, we have

$$\left\|\mathbb{E}\left[f_{\boldsymbol{q}}(\boldsymbol{x})^\top f_{\boldsymbol{q}}(\boldsymbol{x})\right]\right\| = \left\|\mathbb{E}\left[9\left(\boldsymbol{q}^\top\boldsymbol{\xi}\right)^4\boldsymbol{P}_{\boldsymbol{q}^\perp}\boldsymbol{\xi}\boldsymbol{\xi}^\top\boldsymbol{P}_{\boldsymbol{q}^\perp}\boldsymbol{\xi}\boldsymbol{\xi}^\top\boldsymbol{P}_{\boldsymbol{q}^\perp} - 6\left(\boldsymbol{q}^\top\boldsymbol{\xi}\right)^6\boldsymbol{P}_{\boldsymbol{q}^\perp}\boldsymbol{\xi}\boldsymbol{\xi}\boldsymbol{P}_{\boldsymbol{q}^\perp} + \left(\boldsymbol{q}^\top\boldsymbol{\xi}\right)^8\boldsymbol{P}_{\boldsymbol{q}^\perp}\right]\right\|$$

$$\leqslant 9\underbrace{\left\|\mathbb{E}\left[\left(\boldsymbol{q}^\top\boldsymbol{\xi}\right)^4\boldsymbol{P}_{\boldsymbol{q}^\perp}\boldsymbol{\xi}\boldsymbol{\xi}^\top\boldsymbol{P}_{\boldsymbol{q}^\perp}\boldsymbol{\xi}\boldsymbol{\xi}^\top\boldsymbol{P}_{\boldsymbol{q}^\perp}\right]\right\|}_{\mathcal{T}_1} + 6\underbrace{\left\|\boldsymbol{P}_{\boldsymbol{q}^\perp}\mathbb{E}\left[\left(\boldsymbol{q}^\top\boldsymbol{\xi}\right)^6\boldsymbol{\xi}\boldsymbol{\xi}^\top\right]\boldsymbol{P}_{\boldsymbol{q}^\perp}\right\|}_{\mathcal{T}_2} + \underbrace{\mathbb{E}\left[\left(\boldsymbol{q}^\top\boldsymbol{\xi}\right)^8\right]}_{\mathcal{T}_3}.$$

Bound

$$
\begin{aligned}
\mathcal{T}_1 &= \left\| \mathbb{E}\left[ \left(\boldsymbol{q}^\top \boldsymbol{\xi}\right)^4 \boldsymbol{P}_{\boldsymbol{q}^\perp} \boldsymbol{\xi}\boldsymbol{\xi}^\top \boldsymbol{P}_{\boldsymbol{q}^\perp} \boldsymbol{\xi}\boldsymbol{\xi}^\top \boldsymbol{P}_{\boldsymbol{q}^\perp} \right] \right\| \leqslant \left\| \mathbb{E}\left[ \left(\boldsymbol{q}^\top \boldsymbol{\xi}\right)^4 \boldsymbol{\xi}\boldsymbol{\xi}^\top \boldsymbol{P}_{\boldsymbol{q}^\perp} \boldsymbol{\xi}\boldsymbol{\xi}^\top \right] \right\| \\
&= \left\| \mathbb{E}\left[ \left(\boldsymbol{q}^\top \boldsymbol{\xi}\right)^4 \left\| \boldsymbol{P}_{\boldsymbol{q}^\perp} \boldsymbol{\xi} \right\|^2 \boldsymbol{\xi}\boldsymbol{\xi}^\top \right] \right\| \leqslant \mathbb{E}\left[ \left(\boldsymbol{q}^\top \boldsymbol{\xi}\right)^4 \left\| \boldsymbol{\xi} \right\|^4 \right] \leqslant \left\{ \mathbb{E}(\boldsymbol{q}^\top \boldsymbol{\xi})^8 \right\}^{1/2} \left\{ \mathbb{E}\left\| \boldsymbol{\xi} \right\|^8 \right\}^{1/2} \\
&= \left\{ \mathbb{E}\left[ \left\langle \mathcal{P}_{\mathcal{I}}(\boldsymbol{A}^\top \boldsymbol{q}), \boldsymbol{g} \right\rangle^8 \right] \right\}^{1/2} \left\{ \left(\frac{m}{n}\right)^4 \mathbb{E}\left[ (\boldsymbol{x}^\top \boldsymbol{x})^4 \right] \right\}^{1/2},
\end{aligned}
$$

where

$$
\left\{ \mathbb{E}\left[ \left\langle \mathcal{P}_{\mathcal{I}} \boldsymbol{A}^\top \boldsymbol{q}, \boldsymbol{g} \right\rangle^8 \right] \right\}^{\frac{1}{2}} = \sqrt{7!!} \left( \mathbb{E}_{\mathcal{I}} \left\| \mathcal{P}_{\mathcal{I}} \boldsymbol{A}^\top \boldsymbol{q} \right\|^8 \right)^{\frac{1}{2}} \leqslant C_1 \theta \left(\frac{m}{n}\right)^2 \tag{F.23}
$$

the proof of the last inequality is omitted, more details can be found in Lemma F.5, and

$$
\begin{aligned}
\mathbb{E}\left[ \left(\boldsymbol{x}^\top \boldsymbol{x}\right)^4 \right] = \mathbb{E}\left[ \left\langle \mathcal{P}_{\mathcal{I}} \boldsymbol{x}, \mathcal{P}_{\mathcal{I}} \boldsymbol{x} \right\rangle^4 \right] &= \mathbb{E}\left[ \left\langle \mathcal{P}_{\mathcal{I}}(\mathbf{1}_m), \boldsymbol{g}^{\odot 2} \right\rangle^4 \right] \\
&\leqslant c_1 m \theta + c_2 m^2 \theta^2 + c_3 m^3 \theta^3 + c_4 m^4 \theta^4. \tag{F.24}
\end{aligned}
$$

combine, equation F.23 and equation F.24, yield

$$
\mathcal{T}_1 \leqslant C_1 \theta^3 m^2 \left(\frac{m}{n}\right)^4.
$$

$$
\begin{aligned}
\mathcal{T}_2 &= \left\| \boldsymbol{P}_{\boldsymbol{q}^\perp} \mathbb{E}\left[ \left(\boldsymbol{q}^\top \boldsymbol{\xi}\right)^6 \boldsymbol{\xi}\boldsymbol{\xi}^\top \right] \boldsymbol{P}_{\boldsymbol{q}^\perp} \right\| \leqslant \left\| \mathbb{E}\left[ \left(\boldsymbol{q}^\top \boldsymbol{\xi}\right)^6 \boldsymbol{\xi}\boldsymbol{\xi}^\top \right] \right\| = \mathbb{E}\left[ \left(\boldsymbol{q}^\top \boldsymbol{\xi}\right)^6 \left\| \boldsymbol{\xi} \right\|^2 \right] \leqslant \left\{ \mathbb{E}(\boldsymbol{q}^\top \boldsymbol{\xi})^{12} \right\}^{1/2} \left\{ \mathbb{E}\left\| \boldsymbol{\xi} \right\|^4 \right\}^{1/2} \\
&\leqslant \left\{ \mathbb{E}\left\langle \boldsymbol{A}^\top \boldsymbol{q}, \boldsymbol{x} \right\rangle^{12} \right\}^{1/2} \left\{ \mathbb{E}\left\| \boldsymbol{A}\boldsymbol{x} \right\|^4 \right\}^{1/2} = \left\{ \mathbb{E}\left\langle \mathcal{P}_{\mathcal{I}}(\boldsymbol{A}^\top \boldsymbol{q}), \boldsymbol{g} \right\rangle^{12} \right\}^{1/2} \left\{ \left(\frac{m}{n}\right)^2 \mathbb{E}(\boldsymbol{x}^\top \boldsymbol{x})^2 \right\}^{1/2} \\
&\leqslant C_2 \mathbb{E}_{\mathcal{I}}\left[ \left\| \mathcal{P}_{\mathcal{I}}(\boldsymbol{A}^\top \boldsymbol{q}) \right\|^{12} \right]^{1/2} \left[ \left(\frac{m}{n}\right)^2 \left(3m\theta + m(m-1)\theta^2\right) \right]^{1/2} \leqslant C_2 \theta^2 m \left(\frac{m}{n}\right)^4.
\end{aligned}
$$

the proof of the first inequality in the last line is omitted, more details can be found in Lemma F.5.

$$
\mathcal{T}_3 = \mathbb{E}\left[ \left\langle \mathcal{P}_{\mathcal{I}}\left(\boldsymbol{A}^\top \boldsymbol{q}\right), \boldsymbol{g} \right\rangle^8 \right] \leqslant C_3 \mathbb{E}_{\mathcal{I}}\left[ \left\| \mathcal{P}_{\mathcal{I}}\left(\boldsymbol{A}^\top \boldsymbol{q}\right) \right\|^8 \right] \leqslant C_3 \theta \left\| \boldsymbol{A} \right\|^8 \leqslant C_3 \theta \left(\frac{m}{n}\right)^4.
$$

Hence, summarizing all the results above, we obtain

$$
\left\| \mathbb{E}\left[ f_{\boldsymbol{q}}(\boldsymbol{x})^\top f_{\boldsymbol{q}}(\boldsymbol{x}) \right] \right\| \leqslant C\theta m^2 \left(\frac{m}{n}\right)^4
$$

as desired. ∎

**Lemma F.8 (Expectation of** $\operatorname{Hess}\varphi_{\mathrm{DL}}(\cdot)$**)** $\forall \boldsymbol{q} \in \mathbb{S}^{n-1}$, *the expectation of* $\operatorname{Hess}\varphi_{\mathrm{DL}}(\cdot)$ *satisfies*

$$
\operatorname{Hess}\varphi_{\mathrm{DL}}(\boldsymbol{q}) = \operatorname{Hess}\varphi_{\mathrm{T}}(\boldsymbol{q}) = -\boldsymbol{P}_{\boldsymbol{q}^\perp}\left[ 3\boldsymbol{A}\operatorname{diag}\left((\boldsymbol{A}\boldsymbol{q}^\top)^{\odot 2}\right)\boldsymbol{A}^\top - \left\| \boldsymbol{q}^\top \boldsymbol{A} \right\|_4^4 \boldsymbol{I} \right] \boldsymbol{P}_{\boldsymbol{q}^\perp}
$$

**Proof** Direct calculation. ∎

### F.3 Concentration for Convolutional Dictionary Learning

In this section, we show concentration for the Riemannian gradient and Hessian of the following objective for convolutional dictionary learning,

$$
\widehat{\varphi}_{\mathrm{CDL}}(\boldsymbol{q}) = -\frac{1}{12\theta(1-\theta)np} \left\| \boldsymbol{q}^\top \boldsymbol{A}\boldsymbol{X} \right\|_4^4 = -\frac{1}{12\theta(1-\theta)np} \sum_{i=1}^{p} \left\| \boldsymbol{q}^\top \boldsymbol{A}\boldsymbol{X}_i \right\|_4^4
$$

with

$$
\boldsymbol{X} = \begin{bmatrix} \boldsymbol{X}_1 & \boldsymbol{X}_2 & \cdots & \boldsymbol{X}_p \end{bmatrix}, \qquad \boldsymbol{X}_i = \begin{bmatrix} \boldsymbol{C}_{\boldsymbol{x}_{i1}} \\ \vdots \\ \boldsymbol{C}_{\boldsymbol{x}_{iK}} \end{bmatrix}, \tag{F.25}
$$

as we introduced in Section 3, where $\boldsymbol{x}_{ij}$ follows i.i.d. $\mathcal{BG}(\theta)$ distribution as in Assumption E.2. Since $\boldsymbol{C}_{\boldsymbol{x}_{ij}}$ is a circulant matrix generated from $\boldsymbol{x}_{ij}$, it should be noted that each row and column of $\boldsymbol{X}$ is *not* statistically independent, so that our concentration result of dictionary learning in the previous subsection does not directly apply here. However, from Lemma D.1, asymptotically we still have

$$\mathbb{E}_{\boldsymbol{X}}\left[\widehat{\varphi}_{\mathrm{CDL}}(\boldsymbol{q})\right] \;=\; \varphi_{\mathrm{T}}(\boldsymbol{q}) - \frac{\theta}{2(1-\theta)}K^2, \qquad \varphi_{\mathrm{T}}(\boldsymbol{q}) \;=\; -\frac{1}{4}\left\|\boldsymbol{q}^{\top}\boldsymbol{A}\right\|_4^4,$$

in the following we prove finite sample concentration of $\widehat{\varphi}_{\mathrm{CDL}}(\boldsymbol{q})$ to its expectation $\varphi_{\mathrm{T}}(\boldsymbol{q})$ by leveraging our previous results for overcomplete dictionary learning in Proposition F.3 and Proposition F.6.

### F.3.1 Concentration for $\mathrm{grad}\,\widehat{\varphi}_{\mathrm{CDL}}(\cdot)$

**Corollary F.9 (Concentration of $\mathrm{grad}\,\widehat{\varphi}_{\mathrm{CDL}}(\cdot)$)** *Suppose $\boldsymbol{A}$ satisfies Equation (F.9) and $\boldsymbol{X} \in \mathbb{R}^{m \times np}$ is generated as in Equation (F.25) with $\boldsymbol{x}_{ij} \sim_{i.i.d.} \mathcal{BG}(\theta)$ $(1 \leqslant i \leqslant p, 1 \leqslant j \leqslant K)$ and $\theta \in \left(\frac{1}{m}, \frac{1}{2}\right)$. For any given $\delta \in \left(0, cK^2/(m\log^2 p\log^2 np)\right)$, whenever*

$$p \;\geqslant\; C\delta^{-2}\theta K^5 n^2 \log\left(\frac{\theta K n}{\delta}\right),$$

*we have*

$$\sup_{\boldsymbol{q}\in\mathbb{S}^{n-1}} \left\|\mathrm{grad}\,\widehat{\varphi}_{\mathrm{CDL}}(\boldsymbol{q}) - \mathrm{grad}\,\varphi_{\mathrm{T}}(\boldsymbol{q})\right\| \;<\; \delta$$

*holds with probability at least $1 - c'np^{-2}$. Here, $c, c', C > 0$ are some numerical constants.*

**Remark.** Note that our prove have not utilized the convolutional structure of the problem, so that our sample complexity could be loose of a factor of order $n$.

**Proof** Let us write

$$\boldsymbol{X}_i \;=\; \begin{bmatrix} \widetilde{\boldsymbol{x}}_{i1} & \widetilde{\boldsymbol{x}}_{i2} & \cdots & \widetilde{\boldsymbol{x}}_{in} \end{bmatrix}, \quad \text{with} \quad \widetilde{\boldsymbol{x}}_{ij} = \begin{bmatrix} \mathrm{s}_{j-1}\left[\boldsymbol{x}_{i1}\right] \\ \vdots \\ \mathrm{s}_{j-1}\left[\boldsymbol{x}_{iK}\right] \end{bmatrix} \quad 1 \leqslant i \leqslant p, \quad 1 \leqslant j \leqslant n, \tag{F.26}$$

where $\mathrm{s}_\ell\left[\cdot\right]$ denotes circulant shift of length $\ell$. Thus, the Riemannian gradient of $\widehat{\varphi}_{\mathrm{CDL}}(\boldsymbol{q})$ can be written as

$$\begin{aligned} \mathrm{grad}\,\widehat{\varphi}_{\mathrm{CDL}}(\boldsymbol{q}) &= -\frac{1}{3\theta(1-\theta)np}\boldsymbol{P}_{\boldsymbol{q}^{\perp}}\sum_{i=1}^{p}\sum_{j=1}^{n}\left(\boldsymbol{q}^{\top}\boldsymbol{A}\widetilde{\boldsymbol{x}}_{ij}\right)^3\left(\boldsymbol{A}\widetilde{\boldsymbol{x}}_{ij}\right) \\ &= \frac{1}{n}\sum_{j=1}^{n}\left[\underbrace{-\frac{1}{3\theta(1-\theta)p}\boldsymbol{P}_{\boldsymbol{q}^{\perp}}\sum_{i=1}^{p}\left(\boldsymbol{q}^{\top}\boldsymbol{A}\widetilde{\boldsymbol{x}}_{ij}\right)^3\left(\boldsymbol{A}\widetilde{\boldsymbol{x}}_{ij}\right)}_{\mathrm{grad}_j\,\widehat{\varphi}_{\mathrm{CDL}}(\boldsymbol{q})}\right], \end{aligned}$$

so that for each $j$ with $1 \leqslant j \leqslant n$,

$$\mathrm{grad}_j\,\widehat{\varphi}_{\mathrm{CDL}}(\boldsymbol{q}) \;=\; -\frac{1}{3\theta(1-\theta)p}\boldsymbol{P}_{\boldsymbol{q}^{\perp}}\sum_{i=1}^{p}\left(\boldsymbol{q}^{\top}\boldsymbol{A}\widetilde{\boldsymbol{x}}_{ij}\right)^3\left(\boldsymbol{A}\widetilde{\boldsymbol{x}}_{ij}\right)$$

is a summation of independent random vectors across $p$. Hence, we have

$$\sup_{\boldsymbol{q}\in\mathbb{S}^{n-1}}\left\|\mathrm{grad}\,\widehat{\varphi}_{\mathrm{CDL}}(\boldsymbol{q}) - \mathrm{grad}\,\varphi_{\mathrm{T}}(\boldsymbol{q})\right\| \;<\; \frac{1}{n}\sum_{j=1}^{n}\left(\sup_{\boldsymbol{q}\in\mathbb{S}^{n-1}}\left\|\mathrm{grad}_j\,\widehat{\varphi}_{\mathrm{CDL}}(\boldsymbol{q}) - \mathrm{grad}\,\varphi_{\mathrm{T}}(\boldsymbol{q})\right\|\right),$$

where for each $j$ we can apply concentration results in Proposition F.3 for controlling each individual quantity $\left\|\mathrm{grad}_j\,\widehat{\varphi}_{\mathrm{CDL}}(\boldsymbol{q}) - \mathrm{grad}\,\varphi_{\mathrm{T}}(\boldsymbol{q})\right\|$. Therefore, by using a union bound we can obtain the desired result. ∎

Table 1: Gradient for each different loss function

| Problem | Overcomplete Tensor | ODL | CDL |
|---|---|---|---|
| Loss $\varphi(\boldsymbol{q})$ | $-\frac{1}{4}\left\|\boldsymbol{A}^\top\boldsymbol{q}\right\|_4^4$ | $-\frac{1}{4p}\left\|\boldsymbol{Y}^\top\boldsymbol{q}\right\|_4^4$ | $-\frac{1}{4np}\sum_{i=1}^p\left\|\widetilde{\boldsymbol{y}_i^p}\circledast\boldsymbol{q}\right\|_4^4$ |
| Gradient $\nabla\varphi(\boldsymbol{q})$ | $-\boldsymbol{A}\left(\boldsymbol{A}^\top\boldsymbol{q}\right)^{\odot 3}$ | $-\frac{1}{p}\boldsymbol{Y}\left(\boldsymbol{Y}^\top\boldsymbol{q}\right)^{\odot 3}$ | $-\frac{1}{np}\sum_{i=1}^p\boldsymbol{y}_i^p\circledast\left(\widetilde{\boldsymbol{y}_i^p}\circledast\boldsymbol{q}\right)^{\odot 3}$ |

### F.3.2 Concentration for Hess $\widehat{\varphi}_{\mathrm{CDL}}(\cdot)$

**Corollary F.10 (Concentration of** Hess $\widehat{\varphi}_{\mathrm{CDL}}(\cdot)$**)** *Suppose $\boldsymbol{A}$ satisfies Equation (F.9) and $\boldsymbol{X}\in\mathbb{R}^{m\times np}$ is generated as in Equation (F.25) with $\boldsymbol{x}_{ij}\sim_{i.i.d.}\mathcal{BG}(\theta)$ $(1\leqslant i\leqslant p, 1\leqslant j\leqslant K)$ and $\theta\in\left(\frac{1}{m},\frac{1}{2}\right)$. For any given $\delta\in\left(0, cK^2/(m\log^2 p\log^2 np)\right)$, whenever*

$$p \;\geqslant\; C\delta^{-2}\theta K^6 n^3\log\left(\theta Kn/\delta\right),$$

*we have*

$$\sup_{\boldsymbol{q}\in\mathbb{S}^{n-1}}\|\mathrm{Hess}\,\varphi_{\mathrm{DL}}(\boldsymbol{q})-\mathrm{Hess}\,\varphi_{\mathrm{T}}(\boldsymbol{q})\| \;<\; \delta$$

*holds with probability at least $1-c'np^{-2}$. Here, $c, c', C > 0$ are some numerical constants.*

**Proof** Similar to the proof of Corollary F.9, the Riemannian Hessian of $\widehat{\varphi}_{\mathrm{CDL}}(\boldsymbol{q})$ can be written as

$$\begin{aligned}
&\mathrm{Hess}\,\widehat{\varphi}_{\mathrm{CDL}}(\boldsymbol{q})\\
&= -\frac{1}{3\theta(1-\theta)np}\sum_{i=1}^p\sum_{j=1}^n\boldsymbol{P}_{\boldsymbol{q}^\perp}\left[3\left(\boldsymbol{q}^\top\boldsymbol{A}\widetilde{\boldsymbol{x}}_{ij}\right)^2\boldsymbol{A}\boldsymbol{x}_k\left(\boldsymbol{A}\widetilde{\boldsymbol{x}}_{ij}\right)^\top-\left(\boldsymbol{q}^\top\boldsymbol{A}\widetilde{\boldsymbol{x}}_{ij}\right)^4\boldsymbol{I}\right]\boldsymbol{P}_{\boldsymbol{q}^\perp}\\
&= \frac{1}{n}\sum_{j=1}^n\Bigg\{\underbrace{-\frac{1}{3\theta(1-\theta)p}\sum_{i=1}^p\boldsymbol{P}_{\boldsymbol{q}^\perp}\left[3\left(\boldsymbol{q}^\top\boldsymbol{A}\widetilde{\boldsymbol{x}}_{ij}\right)^2\boldsymbol{A}\boldsymbol{x}_k\left(\boldsymbol{A}\widetilde{\boldsymbol{x}}_{ij}\right)^\top-\left(\boldsymbol{q}^\top\boldsymbol{A}\widetilde{\boldsymbol{x}}_{ij}\right)^4\boldsymbol{I}\right]\boldsymbol{P}_{\boldsymbol{q}^\perp}}_{\mathrm{Hess}_j\,\widehat{\varphi}_{\mathrm{CDL}}(\boldsymbol{q})}\Bigg\},
\end{aligned}$$

so that for each $j$ with $1\leqslant j\leqslant n$,

$$\mathrm{Hess}_j\,\widehat{\varphi}_{\mathrm{CDL}}(\boldsymbol{q}) \;=\; -\frac{1}{3\theta(1-\theta)p}\sum_{i=1}^p\boldsymbol{P}_{\boldsymbol{q}^\perp}\left[3\left(\boldsymbol{q}^\top\boldsymbol{A}\widetilde{\boldsymbol{x}}_{ij}\right)^2\boldsymbol{A}\boldsymbol{x}_k\left(\boldsymbol{A}\widetilde{\boldsymbol{x}}_{ij}\right)^\top-\left(\boldsymbol{q}^\top\boldsymbol{A}\widetilde{\boldsymbol{x}}_{ij}\right)^4\boldsymbol{I}\right]\boldsymbol{P}_{\boldsymbol{q}^\perp}$$

is a summation of independent random vectors across $p$. Hence, we have

$$\sup_{\boldsymbol{q}\in\mathbb{S}^{n-1}}\|\mathrm{Hess}\,\widehat{\varphi}_{\mathrm{CDL}}(\boldsymbol{q})-\mathrm{Hess}\,\varphi_{\mathrm{T}}(\boldsymbol{q})\| \;<\; \frac{1}{n}\sum_{j=1}^n\left(\sup_{\boldsymbol{q}\in\mathbb{S}^{n-1}}\|\mathrm{Hess}_j\,\widehat{\varphi}_{\mathrm{CDL}}(\boldsymbol{q})-\mathrm{Hess}\,\varphi_{\mathrm{T}}(\boldsymbol{q})\|\right),$$

where for each $j$ we can apply concentration results in Proposition F.6 for controlling each individual quantity $\|\mathrm{Hess}_j\,\widehat{\varphi}_{\mathrm{CDL}}(\boldsymbol{q})-\mathrm{Hess}\,\varphi_{\mathrm{T}}(\boldsymbol{q})\|$. Therefore, by using a union bound we can obtain the desired result. ∎

## G  Optimization Algorithms

### G.1  Optimization

In this part of the appendix, we introduce algorithmic details for optimizing the following problem

$$\min_{\boldsymbol{q}}\;\varphi(\boldsymbol{q}),\qquad \boldsymbol{q}\in\mathbb{S}^{n-1},$$

where the loss function $\varphi(\boldsymbol{q})$ and its gradient $\nabla\varphi(\boldsymbol{q})$ for different problems are listed in Table 1.

---

**Algorithm 2** Projected Riemannian Gradient Descent Algorithm

---

**Input:**   Data $\boldsymbol{Y} \in \mathbb{R}^{n \times p}$
**Output:**   the vector $\boldsymbol{q}_\star$
 1: Initialize the iterate $\boldsymbol{q}^{(0)}$ randomly, and set a stepsize $\tau^{(0)}$.
 2: **while** not converged **do**
 3:      Compute Riemannian gradient $\operatorname{grad} \varphi(\boldsymbol{q}^{(k)}) = \mathcal{P}_{(\boldsymbol{q}^{(k)})^\perp} \nabla \varphi(\boldsymbol{q}^{(k)})$.
 4:      Update the iterate by

$$\boldsymbol{q}^{(k+1)} \;=\; \mathcal{P}_{\mathbb{S}^{n-1}} \left( \boldsymbol{q}^{(k)} - \tau^{(k)} \operatorname{grad} \varphi(\boldsymbol{q}^{(k)}) \right).$$

 5:      Choose a new stepsize $\tau^{(k+1)}$, and set $k \leftarrow k + 1$.
 6: **end while**

---

**Algorithm 3** Power Method

---

**Input:**   Data $\boldsymbol{Y} \in \mathbb{R}^{n \times p}$
**Output:**   the vector $\boldsymbol{q}_\star$
 1: Randomly initialize the iterate $\boldsymbol{q}^{(0)}$.
 2: **while** not converged **do**
 3:      Compute the gradient $\nabla \varphi(\boldsymbol{q}^{(k)})$.
 4:      Update the iterate by

$$\boldsymbol{q}^{(k+1)} \;=\; \mathcal{P}_{\mathbb{S}^{n-1}} \left( -\nabla \varphi(\boldsymbol{q}^{(k)}) \right).$$

 5:      Set $k \leftarrow k + 1$.
 6: **end while**

---

**Riemannian gradient descent.**   To optimize the problem, the most natural idea is starting from a *random* initialization, and taking projected Riemannian gradient descent steps

$$\boldsymbol{q} \quad \leftarrow \quad \mathcal{P}_{\mathbb{S}^{n-1}} \left( \boldsymbol{q} - \tau \cdot \operatorname{grad} \varphi(\boldsymbol{q}) \right), \quad \operatorname{grad} \varphi(\boldsymbol{q}) \;=\; \mathcal{P}_{\boldsymbol{q}^\perp} \nabla \varphi(\boldsymbol{q}), \tag{G.1}$$

where $\tau$ is the stepsize that can be chosen via linesearch or set as a small constant. We summarize this simple method in Algorithm 2.

**Power method.**   In Algorithm 3 we also introduce a simple power method[17] Journée et al. (2010) by noting that the loss function $\varphi(\boldsymbol{q})$ is concave so that the problem is equivalent to maximizing a convex function. For each iteration, we simply update $\boldsymbol{q}$ by

$$\boldsymbol{q} \quad \leftarrow \quad \mathcal{P}_{\mathbb{S}^{n-1}} \left( -\nabla \varphi(\boldsymbol{q}) \right)$$

which is parameter-free and enjoys much faster convergence speed. We summarized the method in Algorithm 3. Notice that the power iteration can be interpreted as the Riemannian gradient descent with varied step sizes in the sense that

$$\mathcal{P}_{\mathbb{S}^{n-1}} \left( \boldsymbol{q} - \tau \cdot \operatorname{grad} \varphi(\boldsymbol{q}) \right) \;=\; \mathcal{P}_{\mathbb{S}^{n-1}} \bigg( -\tau \nabla \varphi(\boldsymbol{q}) + \underbrace{\left( 1 - \tau \cdot \boldsymbol{q}^\top \nabla \varphi(\boldsymbol{q}) \right) \boldsymbol{q}}_{=0} \bigg) \;=\; \mathcal{P}_{\mathbb{S}^{n-1}} \left( -\nabla \varphi(\boldsymbol{q}) \right)$$

by setting $\tau = \frac{1}{\boldsymbol{q}^\top \nabla \varphi(\boldsymbol{q})}$.

## G.2   FAST IMPLEMENTATION OF CDL VIA FFT

Given the problem setup of CDL in Section 3, in the following we describe more efficient implementation of solving CDL using convolution and FFTs. Namely, we show how to rewrite the gradient of $\varphi_{\text{CDL}}(\boldsymbol{q})$ in the convolutional form. Notice that the preconditioning matrix can be rewrite as a

---

[17]Similar approach also appears in (Zhai et al., 2019).

circulant matrix by

$$
\boldsymbol{P} \;=\; \left( \frac{1}{\theta n p} \sum_{i=1}^{p} \boldsymbol{C}_{\boldsymbol{y}_i} \boldsymbol{C}_{\boldsymbol{y}_i}^{\top} \right)^{-1/2} \;=\; \boldsymbol{F}^{*} \operatorname{diag}\left( \widehat{\boldsymbol{p}} \right) \boldsymbol{F} \;=\; \boldsymbol{C}_{\boldsymbol{p}}, \;\; \boldsymbol{p} \;=\; \boldsymbol{F}^{-1} \left( \frac{1}{\theta n p} \sum_{i=1}^{p} |\widehat{\boldsymbol{y}}_i|^{\odot 2} \right)^{-1/2},
$$

where $\widehat{\boldsymbol{y}}_i = \boldsymbol{F} \boldsymbol{y}_i$. Thus, we have

$$
\boldsymbol{P} \boldsymbol{C}_{\boldsymbol{y}_i} \;=\; \boldsymbol{C}_{\boldsymbol{p}} \boldsymbol{C}_{\boldsymbol{y}_i} \;=\; \boldsymbol{C}_{\boldsymbol{p} \circledast \boldsymbol{y}_i} \;=\; \boldsymbol{C}_{\boldsymbol{y}_i^p}, \qquad \boldsymbol{y}_i^p \;=\; \boldsymbol{p} \circledast \boldsymbol{y}_i,
$$

so that

$$
\min_{\boldsymbol{q}} \; \varphi_{\mathrm{CDL}}(\boldsymbol{q}) \;=\; -\frac{1}{4np} \sum_{i=1}^{p} \left\| \boldsymbol{C}_{\boldsymbol{p} \circledast \boldsymbol{y}_i}^{\top} \boldsymbol{q} \right\|_4^4 \;=\; -\frac{1}{4np} \sum_{i=1}^{p} \left\| \widetilde{\boldsymbol{y}_i^p} \circledast \boldsymbol{q} \right\|_4^4, \qquad \text{s.t.} \quad \boldsymbol{q} \in \mathbb{S}^{n-1},
$$

Thus, we have the gradient

$$
\nabla \varphi_{\mathrm{CDL}}(\boldsymbol{q}) \;=\; -\frac{1}{np} \sum_{i=1}^{p} \boldsymbol{y}_i^p \circledast \left( \widetilde{\boldsymbol{y}_i^p} \circledast \boldsymbol{q} \right)^{\odot 3},
$$

where $\breve{\boldsymbol{v}}$ denote a *cyclic reversal* of any $\boldsymbol{v} \in \mathbb{R}^n$, i.e., $\breve{\boldsymbol{v}} = [v_1, v_n, v_{n-1}, \cdots, v_2]^{\top}$.

