# OpenReview forum: "Geometric Analysis of Nonconvex Optimization Landscapes for Overcomplete Learning"
_ICLR.cc/2020/Conference — Accept (Talk)_

### Official Review · AnonReviewer2 · 2019-10-22
**Official Blind Review #2**

**Rating:** 8

**Review:**

[Summary]
This paper studies the problem of non-convex optimization for Dictionary Learning (DL) in the situation when the underlying dictionary is over-complete (more basis vectors m than the dimension n). The paper proves that the L4 maximization formulation has a nice global landscape and can be efficiently minimized by (Riemannian) gradient descent, when the over-complete ratio m/n is less than an absolute constant. A similar result is proved for convolutional dictionary learning.

[Pros]
The theoretical results in this paper provides a solid improvement over the prior understandings on overcomplete DL, a setting that is practically important yet theoretically more challenging than standard orthogonal/complete DL.

Specifically, the prior work of (Ge & Ma 2017) shows only a nice local optimization landscape when m > n^{1+\eps} and hypothesizes that the global landscape is bad in the same setting (there exists bad local minima out of a certain sub-level set). In comparison, this work proves that at least for m/n <= 3 (roughly), the landscape is globally benign (has the strict saddle property), therefore providing a new understanding that the benign landscape is still preserved from “the other side” where m/n grows mildly above 1. The analysis contains novel technicalities and can be of general interest for understanding the landscape of non-convex problems.

The paper also provides experimental evidence that gradient descent converges globally up until m = O(n^2), a broader regime than suggested by the theory (m <= 3n). (Though when m >= n^{1+\eps}, the reason of global convergence from random init may be far from the present theory, in that there can be potentially exponentially many bad local min yet gradient descent won’t get trapped.)

[Cons]
It is still a bit disturbing to see that m/n needs to be bounded by a fixed absolute constant, rather than *any* constant, for the theory to work. From the proofs it seems like this constant (3) may have the potential to be improved, but it is not quite easy to completely get rid of it?


**Experience Assessment:**

I have published one or two papers in this area.

**Review Assessment: Checking Correctness Of Derivations And Theory:**

I assessed the sensibility of the derivations and theory.

**Review Assessment: Checking Correctness Of Experiments:**

I assessed the sensibility of the experiments.

**Review Assessment: Thoroughness In Paper Reading:**

I read the paper at least twice and used my best judgement in assessing the paper.

---

> ### Author Response · Authors · 2019-11-13
> **Reply to Reviewer #2**
>
> We thank the reviewer for the accurate interpretations of our results and appreciation of our work. In the revision, to make our paper more accessible to the readers, we have carefully revised the main draft, correcting typos and inaccurate statements.
>
> We agree with the reviewer that the absolute constant for the overcompleteness of ODL is disturbing, and we also believe this benign geometry should hold for a larger overcompleteness than we proved here. The major bottleneck of showing this is due to our very loose analysis for negative curvature in Region $\mathcal R_{\mathrm N}$. The authors have tried many ways to improve this bound, but have not yet managed to succeed so far.
>
> One idea might be to consider i.i.d. Gaussian dictionary instead of the deterministic incoherent dictionary, and use probabilistic analysis instead of the worst-case deterministic analysis. However, our preliminary analysis suggests that elementary concentration tools for Gaussian empirical processes are not sufficient to achieve this goal. More advanced probabilistic tools might be needed here.
>
> Another idea that might be promising is to leverage more advanced tools such as the sum of squares (SoS) techniques. Previous results (e.g., Barak et al., 2015; Ma et al., 2016; Schramm & Steurer, 2017) used SoS as a computational tool for solving this type of problems, while the computational complexity is often quasi-polynomial and hence cannot handle problems of large-scale. In contrast, our idea here is to use SoS to verify the geometric structure of the optimizing landscape instead of computation, to have a uniform control of the negative curvature in $\mathcal R_{\mathrm N}$. If we succeeded, this might lead to a tighter bound on the overcompleteness. Moreover, this can also serve as a more general method for verifying the benign optimization landscapes of other nonconvex problems. We will include a discussion section for elaborating these ideas in the future if the reviewer thinks it would be beneficial for the audience.

---

### Official Review · AnonReviewer1 · 2019-10-23
**Official Blind Review #1**

**Rating:** 8

**Review:**

This paper studies the dictionary learning problem for two popular settings involving sparsely used over-complete dictionaries and convolutional dictionaries.

For the over-complete dictionary setting, given the measurements of the form $Y = A X$, where $A$ and $X$ denote the over-complete dictionary and the sparse coefficients, respectively, the paper explores an $\ell^4$-norm maximization approach to recover the dictionary $A$. This corresponds to maximizing $\|q^TY\|^4_4$ over $q \in \mathbb{S}^{n-1}$. Interestingly, the paper shows that when $A$ is unit norm tight frame and incoherent the optimization landscape of the aforementioned non-convex objective has strict saddle points that can be escaped by along negative curvature. Furthermore, all local minimizers are globally optimal which are close to one of the columns of $A$. This shows that any descent method that can escape the saddle points will (approximately) recover one of the columns of $A$.

For convolution dictionaries, the paper shows that when the underlying filters are incoherent a suitably modified $\ell^4$-norms based objective has only strict saddles over a sub-level set. Furthermore, all local optimizers within this sub-level set are close to one of the convolution filters.

The reviewer believes that this paper presents many interesting and novel results that extend our understanding of provable methods for dictionary learning. As claimed in the paper, this the first global characterization for the non-convex optimization landscape for over-complete dictionary learning. Besides, the paper provides the first provable guarantees for convolution dictionary learning. Overall, the paper is very well written and the key ideas used in the paper are nicely explained in the main body of the paper. The experimental results in the paper also corroborate the theoretical findings of the paper.

Minor comments:

1. In page 2, "....can be simply summarized by the following slogan." ---> "....can be simply summarized by the following statement."?

2.  In page 7, replace "cook up" with "propose"?

------------------------------
After rebuttal

Thank you for the response. Releasing the code for reproducibility purposes is certainly a great idea.

**Experience Assessment:**

I have published one or two papers in this area.

**Review Assessment: Checking Correctness Of Derivations And Theory:**

I assessed the sensibility of the derivations and theory.

**Review Assessment: Checking Correctness Of Experiments:**

I assessed the sensibility of the experiments.

**Review Assessment: Thoroughness In Paper Reading:**

I read the paper at least twice and used my best judgement in assessing the paper.

---

> ### Author Response · Authors · 2019-11-13
> **Reply to Reviewer #1**
>
> We thank the reviewer for the appreciation of our work. In the revision, we have carefully corrected all the minor issues raised by the reviewers. In addition, we have carefully revised the main draft, correcting other typos and inaccurate statements. We plan to release the code in the very near future as well for reproducible purposes.

---

### Official Review · AnonReviewer3 · 2019-10-25
**Official Blind Review #3**

**Rating:** 8

**Review:**

The authors consider two problems: Overcomplete dictionary learning (ODL)
and convolution dictionary learning (CDL).
Dictionary learning learns a matrix factorization of the data
Y = A X
where A is the dictionary and X is the (known to be sparse) code.
Y consists of n rows (sample size) and p columns (dimension).
In the overcomplete version A is n x m where m > n, i.e. the number of learned
features is larger than the sample size.
The CDL problem is a special case of the ODL problem where the dictionary
matrix is known to consist of convolution filters instead of being unstructured.

The authors show that under a given set of assumptions local nonconvex
optimization can be used to find globally relevant solutions.
The basic assumptions are:
(i) unit norm tight frame
(ii) mu-incoherence
	(relates the angles of the columns of a, e.g.\ if columns are orthogonal,
	they are incoherent / have small mu)
(ii) stochastic model of the code X that says entries are Gaussian and sparse
	according to a Bernoulli random variable
The authors present the idea of maximizing the l^4 norm of A^T q in order to
find q as rows of A.
Apparently l^4 norm maximization leads to "spikiness" which is exactly
desirable under mu-incoherence.

The authors show (assuming p \to \infty) that the optimization nonconvex
landscape (constrained to the sphere) does not contain any stationary points
without negative curvature.
A saddle avoiding optimizer therefore converges to local minimizers from
random initialization.

The authors also show that the analysis extends to CDL via a preconditioned
initializer.
Finally, they go on to briefly show some experiments that further validate
the theory presented in the paper.

Overall, the authors present a rigorous technical analysis using powerful
mathematical tools for nonconvex optimization (which is relevant to many
machine learning problems).

I am recommending to accept based on the high quality of the work.
But I am not confident as to the accessibility of the paper to the wide
audience of ICLR as it is rather technical.
Perhaps, the complete contribution would be better suited as a journal article.

Notes:
It would have been useful to give some more intuition about what "spikiness"
of A^T q is, why spikiness exists under mu-incoherence and why l^4 norm
maximization improves spikiness.

I am not sure that the inclusion of the CDL problem is beneficial for a
converence paper and would rather have more space allocated to the intuition on
why the method works for ODL.


**Experience Assessment:**

I do not know much about this area.

**Review Assessment: Checking Correctness Of Derivations And Theory:**

I did not assess the derivations or theory.

**Review Assessment: Checking Correctness Of Experiments:**

I did not assess the experiments.

**Review Assessment: Thoroughness In Paper Reading:**

I made a quick assessment of this paper.

---

> ### Author Response · Authors · 2019-11-13
> **Reply to Reviewer #3**
>
> We thank the reviewer for the comprehensive summary of our results and invaluable suggestions. We have double checked our paper and revised accordingly.
>
> On Page 4 of the revised draft, we have introduced a notion of the spikiness and used simulations in Figure 2 to provide a better explanation of why maximizing $\ell^4$ promotes spikiness (we are not aware of any formal definition of spikiness in the literature). Intuitively, we characterize the spikiness of a vector by the ratio between the largest and the second largest entries in magnitude. In Figure 2, we added simulations to demonstrate that the $\ell^4$-norm tends to be larger when the spikiness of the vector increases. If $\mathbf q$ is close to one of the columns of $\mathbf A$ (e.g., $\mathbf q = \mathbf a_1$), around Equation (2.5) we explained why the vector $\mathbf A^\top \mathbf q$ should be spiky, given the small incoherence $\mu$ of $\mathbf A$ (i.e., $\mu \ll 1$). In theory, we rigorously proved that $\mathbf q = \mathbf a_1$ is close to one of the global optimizers due to the spikiness of $\mathbf \zeta =\mathbf A^\top \mathbf q$.
>
> We agree with the reviewer that the inclusion of CDL might overexpose the readers. Indeed, the authors had several discussions over this before the submission. That being said, we had included CDL because we believe the inclusion of CDL is very beneficial to the audience in the ICLR community. The CDL problem can be reviewed as a more structured ODL problem such that it can be analyzed in a similar fashion with a few new ingredients (e.g., initialization, preconditioning, new concentration ideas). Building on the intuition and theory for ODL, it could make our introduction of CDL more accessible to the audience and save us the effort for another repetitive work. Moreover, the CDL can be reviewed as a very simple one-layer convolutional neural network (CNN) (Papyan et al., 2017a; 2018). The theory developed here has the potential to serve as a building block for developing more interpretable deep CNN, which closely relates to the core interest of the ICLR community. If the reviewer thinks it would be beneficial to address this issue, we will release a much-extended version of this work on arxiv in the future and provide a link in the final version of this paper.

---

> > ### Comment · AnonReviewer3 · 2019-11-15
> > **Reply to Authors**
> >
> > The authors' addressed my remarks and I changed my recommendation from weak accept to accept.

---

> > > ### Author Response · Authors · 2019-11-15
> > > **Reply to Reviewer #3**
> > >
> > > We thank the reviewer for the appreciations of our efforts in the revision.

---

### Decision · Program_Chairs · 2019-12-19

**Decision:**

Accept (Talk)

**Comment:**

This paper investigates the use non-convex optimization for two dictionary learning problems, i.e., over-complete dictionary learning and convolutional dictionary learning. The paper provides theoretical results, associated with empirical experiments, about the fact that, that when formulating the problem as an l4 optimization, gives rise to a landscape with strict saddle points and as such, they can be escaped with negative curvature. As a result, descent methods can be used for learning with provable guarantees. All reviews found the work extremely interesting, highlighting the importance of the results that constitute "a solid improvement over the prior understandings on over-complete DL" and "extends our understanding of provable methods for dictionary learning". This is an interesting submission on non-convex optimization, and as such of interest to the ML community of ICLR . I'm recommending this work for acceptance.